# Doubly Regularized Markov Decision Processes for Robust Reinforcement Learning

Yiting He [* 1]   Zhishuai Liu [* 2]   Pan Xu [2]

## Abstract

Empirical successes show that regularization improves the stability and efficiency of reinforcement learning (RL), with applications in robotics and post-training of large language models. Yet, theoretical analyses of regularized Markov decision processes (MDPs) have mostly been confined to the standard RL setting. In this work, we investigate regularized MDPs through the lens of robust RL. We introduce a doubly regularized MDP framework that combines policy and dynamics regularizations, enabling robust policy learning while naturally accommodating continuous action spaces. Within this framework, we develop an optimism-based online algorithm and provide the first finite-sample regret guarantees in both tabular and linear settings. Our results show that algorithms for doubly regularized MDPs are as sample-efficient as well-studied robust MDP algorithms, while additionally benefiting from the flexibility of soft policies. We further design practical algorithmic variants for both settings and demonstrate empirically that our approach efficiently and effectively handles function approximation and exploration in large state-action spaces, achieving robust performance.

## 1. Introduction

Regularizations have been widely studied in reinforcement learning (RL) literature. Practically, existing works leveraging policy regularization have demonstrated tremendous successes in real world applications such as game playing (Mnih et al., 2016), robotics control (Schulman et al., 2015; 2017) and post-training of large language models (Bai et al., 2022; Ouyang et al., 2022). Theoretically, extensive studies (Geist et al., 2019; Yang et al., 2019; Vieillard et al., 2020; Zhao et al., 2025) have investigated the policy regularized MDP in the standard RL setting. Moreover, recent works (Derman et al., 2021; Eysenbach & Levine, 2021; Husain et al., 2021) also point out its relation with reward-robust MDPs. Another line of works (Yang et al., 2023; Zhang et al., 2024; He et al., 2025) propose the regularized robust MDP (RRMDP) framework, where a dynamics regularization is added to the objective function to account for robustness against potential dynamics shift. They have shown that the dynamics regularization leads to more efficient estimation procedures of value functions compared with the conventional distributionally robust MDP framework (Iyengar, 2005; Nilim & El Ghaoui, 2005).

To combine the strength of regularization, in this work, we unify the policy and dynamics regularization in a doubly regularized MDP framework, and investigate it through the lens of robust reinforcement learning. In particular, we first establish the framework in the tabular setting where the state and action spaces are finite. Unlike prior works (Yang et al., 2023; Zhang et al., 2024; Panaganti et al., 2024) assuming access to a simulator or a fixed dataset, we focus on the more realistic online setting, where the agent can only gather data through interaction with the environment. In this setting, the agent faces the unique challenge of exploration, that is, efficiently acquiring information about the environment through strategically selecting the policy. Existing results under this online setting often rely on restrictive structural assumptions, such as the fail-state condition in Lu et al. (2024); Liu & Xu (2024a); Liu et al. (2024); Gu et al. (2025), which applies only to uncertainty sets defined by Total Variation (TV) distance and does not extend to more general $f$-divergences. Recently, He et al. (2025) identified the information deficit issue as the central obstacle in robust RL, arising from the need to generalize experience under distribution shifts to perturbed environments while only interacting with the source environment. Building on this, we adopt the bounded visitation measure ratio assumption proposed by He et al. (2025), extending the analysis to handle the double regularization of our framework.

[*]Equal contribution  [1]Georgia Institute of Technology [2]Duke University. Correspondence to: Pan Xu <pan.xu@duke.edu>.

*Proceedings of the $43^{rd}$ International Conference on Machine Learning*, Seoul, South Korea. PMLR 306, 2026. Copyright 2026 by the author(s).

To account for large state and action spaces in practical scenarios, we place particular emphasis on TV and KL divergences, where the resulting robust Bellman operators admit tractable forms that directly inform algorithm design. Building on this, we develop a set of algorithmic components as well as several stability-oriented design choices tailored to the proposed doubly regularized framework.

We summarize our main contribution as follows:

- We establish a novel doubly regularized MDP framework that incorporates both transition regularization and policy regularization, and show that this framework achieves robustness with respect to both transitions and rewards.

- Under this framework, we propose an online learning algorithm Robust Soft Policy Value Iteration (RSPVI) with general $f$-divergence based transition regularization terms and several policy regularization formulations. We establish the first finite-sample regret bounds for doubly regularized MDPs.

- We further extend RSPVI to the setting with linear function approximation. With fewer structural assumptions than prior work, we provide the first finite-sample regret guarantees applicable to broader settings defined by general $f$-divergences.

- In stark contrast to the conventional robust MDP framework, the optimal policies under doubly regularized MDPs are stochastic. We develop a practical version of RSPVI and empirically evaluate it in various tasks, showing that the stochastic policies learned by RSPVI facilitate exploration and possess robustness performances against environment perturbation.

**Notations** For $H \in \mathbb{Z}_+$, we denote $[H] = \{1, \cdots, H\}$. For any set $\mathcal{S}$, define $\Delta(\mathcal{S})$ as the set of probability distributions over $\mathcal{S}$. For a convex function $f : [0, +\infty) \to (-\infty, +\infty]$ with $f(x)$ finite for $x > 0$, $f(1) = 0$ and $f(0) = \lim_{t \to 0^+} f(t)$. For $P, Q \in \Delta(\mathcal{S})$ and $P \ll Q$, the $f$-divergence of $P$ from $Q$ is defined as $D_f(P\|Q) = \int_\Omega f(\frac{dP}{dQ}) \, dQ$. In this paper, we focus on total variation (TV) distance with $f(t) = \frac{1}{2}|t - 1|$, Kullback-Leibler (KL) divergence with $f(t) = t \ln t$, and $\chi^2$-divergence with $f(t) = (t - 1)^2$. We use $\mathcal{O}(\cdot)$ to hide absolute constant factors and $\widetilde{\mathcal{O}}(\cdot)$ to further hide logarithmic factors. We denote $\text{proj}_C(x) = \arg \inf_{c \in C} \|c - x\|^2$ as the projection of $x$ onto the set $C$. We denote the support function of a set $\mathcal{Z} \subseteq \mathbb{R}^n$ as $\sigma_{\mathcal{Z}}(\boldsymbol{y}) = \max_{\boldsymbol{a} \in \mathcal{Z}} \langle \boldsymbol{a}, \boldsymbol{y} \rangle$.

## 2. Most Related Work

There are two lines of work that are most related to ours.

**Robust MDPs** Robust MDPs were first introduced as a control problem with a known nominal transition model (Iyengar, 2005; Nilim & El Ghaoui, 2005; Xu & Mannor, 2006; Wiesemann et al., 2013; Mannor et al., 2016), where robust policies are derived by solving a constrained max-min optimization problem. Subsequent work extended robust MDPs to the learning setting, both with access to a generative model allowing the agent to sample transitions for an arbitrary state-action pair (Zhou et al., 2021; Yang et al., 2022; Panaganti & Kalathil, 2022; Shi et al., 2024) and with pre-collected datasets that provide adequate coverage of the optimal policy under the perturbed environment (Shi & Chi, 2024; Panaganti et al., 2022; Blanchet et al., 2023; Wang et al., 2024a; Liu & Xu, 2024b; 2025). More recently, Lu et al. (2024); Liu & Xu (2024a) studied robust MDPs in the online setting under the fail-state or vanishing minimal-value structure assumption. He et al. (2025) identified the key challenge of online robust learning as the information deficit issue, and addressed this by introducing a maximum visitation ratio assumption, extending the analysis to the general $f$-divergence setting. Ghosh et al. (2025) employed a variance-aware proof technique to refine the results of He et al. (2025). Recently, Gu et al. (2025) proposed incorporating the linear DRMDP framework with policy regularization. Our work differs substantially in several respects. First, our results also apply to the tabular setting. Second, we adopt the bounded visitation measure ratio assumption, whereas Gu et al. (2025) relies on the fail-state assumption. Finally, we propose a method to employ general function approximation in the tabular setting and to learn linear feature mappings via a discrete VAE, improving practicality and scalability to large state and action spaces.

**Regularized Robust MDPs** Unlike the above approaches formulating robust MDPs as a constrained optimization problem, regularized robust MDPs were first introduced by Yang et al. (2023) and Zhang et al. (2024) as an efficient alternative. Yang et al. (2023) focused on the generative model setting, providing a sample complexity analysis and proving the statistical equivalence of the two formulations. Zhang et al. (2024) established the equivalence with risk-sensitive MDPs, analyzed the convergence rate of policy gradient methods, and proposed a value iteration algorithm with general function approximation for the offline setting. Subsequently, Panaganti et al. (2024) extended the framework to general $f$-divergences with function approximation in the offline setting, and to the TV distance under a fail-state assumption in the hybrid offline-online setting. More recently, Tang et al. (2025) studied linear function approxi-

mation for general $f$-divergences using offline datasets.

## 3. Preliminaries

**Markov Decision Process**  A finite-horizon Markov Decision Process is denoted as the tuple $(\mathcal{S}, \mathcal{A}, P, r, H)$. Here, $\mathcal{S}$ and $\mathcal{A}$ represent the state and action spaces. We also denote $|\mathcal{A}|$ as the volume of $\mathcal{A}$, where $|\mathcal{A}|$ is the number of actions for discrete action space and $|\mathcal{A}| = \int_{\mathcal{A}} 1 \, da$ for continuous space. The transition dynamics are given by $P = \{P_h\}_{h=1}^H$, where each $P_h(\cdot|s, a) : \mathcal{S} \to \Delta(\mathcal{S})$ specifies the probability distribution over the next state when action $a$ is taken at state $s$ and step $h$. The reward functions are $r = \{r_h\}_{h=1}^H$, with each $r_h : \mathcal{S} \times \mathcal{A} \to [0, 1]$ assumed to be deterministic. We define the value function and $Q$-function as $V_h^\pi(s) = \mathbb{E}_{\pi, P}\big[\sum_{t=h}^H r_t(s_t, a_t) \mid s_h = s\big]$ and $Q_h^\pi(s, a) = \mathbb{E}_{\pi, P}\big[\sum_{t=h}^H r_t(s_t, a_t) \mid s_h = s, a_h = a\big]$.

**Doubly Regularized MDP**  We now introduce the doubly regularized framework. A finite-horizon doubly regularized MDP can be denoted as the tuple $(\mathcal{S}, \mathcal{A}, P^o, r, D, \beta, \Omega^\eta, H)$. Here, $P^o$ and $r$ denote the transition dynamics and the reward function, $D$ is a probability divergence that quantifies the distribution shift (later we will instantiate D as TV, KL or $\chi^2$-divergence), and $\beta$ is the corresponding regularization parameter. $\Omega_s^\eta(\pi) \triangleq \Omega^\eta(s, \pi(\cdot|s))$ denotes a penalty function on the chosen policy $\pi$ at state $s$, with the $\eta$ in $\Omega^\eta$ controlling the extent of this penalty. We define the doubly regularized value function and $Q$-function as

$$V_h^{\pi,\beta,\eta}(s) = \inf_{P_t \in \Delta(\mathcal{S})} \mathbb{E}_{\pi, \{P_t\}_{t=h}^H}\Bigg[ \sum_{t=h}^H \Big( r_t(s_t, a_t) $$
$$+ \beta \cdot \mathrm{D}(P_t(\cdot|s_t, a_t)\|P_t^o(\cdot|s_t, a_t)) - \Omega_{s_t}^\eta(\pi_t) \Big) \mid s_h = s\Bigg],$$

$$Q_h^{\pi,\beta,\eta}(s, a) = \inf_{P_t \in \Delta(\mathcal{S})} \mathbb{E}_{\pi, \{P_t\}_{t=h}^H}\Bigg[ -\sum_{t=h+1}^H \Omega_{s_t}^\eta(\pi_t) + \sum_{t=h}^H $$
$$\Big( r_t(s_t, a_t) + \beta \cdot \mathrm{D}(P_t(\cdot|s_t, a_t)\|P_t^o(\cdot|s_t, a_t)) \Big) \mid s_h = s, a_h = a\Bigg].$$

We then define the optimal value function $V_h^{*,\beta,\eta}(s) = \sup_{\pi \in \Pi} V_h^{\pi,\beta,\eta}(s)$ and the optimal $Q$-function $Q_h^{*,\beta,\eta}(s, a) = \sup_{\pi \in \Pi} Q_h^{\pi,\beta,\eta}(s, a)$ for all $(h, s, a) \in [H] \times \mathcal{S} \times \mathcal{A}$, where $\Pi$ is the set of all possible policies. Correspondingly, the optimal policy $\pi^* = \{\pi_h^*\}_{h=1}^H$ is defined as the policy that achieves the optimal value function for all $(h, s) \in [H] \times \mathcal{S}$, that is, $\pi_h^*(s) = \arg\sup_{\pi \in \Pi} V_h^{\pi,\beta,\eta}(s)$.

**Dynamic Programming Principle**  Under the doubly regularized framework, we establish the dynamic programming principle as follows.

**Proposition 3.1.** For doubly regularized tabular MDPs, it holds that for any policy $\pi$ and any $(h, s, a) \in [H] \times \mathcal{S} \times \mathcal{A}$,

$$Q_h^{\pi,\beta,\eta}(s, a) = r_h(s, a) + \inf_{P_h \in \Delta(\mathcal{S})} \Big\{ \mathbb{E}_{s' \sim P_h(\cdot|s,a)}\big[V_{h+1}^{\pi,\beta,\eta}(s')\big]$$
$$+ \beta \cdot \mathrm{D}\big(P_h(\cdot|s, a)\big\|P_h^o(\cdot|s, a)\big) \Big\}, \quad (3.1)$$
$$V_h^{\pi,\beta,\eta}(s) = -\Omega_s^\eta(\pi_h) + \mathbb{E}_{a \sim \pi_h(\cdot|s)}\big[Q_h^{\pi,\beta,\eta}(s, a)\big]. \quad (3.2)$$

**Double Robustness of the Proposed Framework**  The effectiveness of incorporating regularization on transition deviations and accounting for the worst case in transition robustness has been extensively studied in the literature (Yang et al., 2023; Zhang et al., 2024; Panaganti et al., 2024; He et al., 2025; Tang et al., 2025). Remark 3.2 further demonstrates that introducing a policy regularization term is equivalent to considering the worst-case reward within a suitably defined uncertainty set (Derman et al., 2021). Therefore, our proposed doubly regularized framework can achieve double robustness.

**Remark 3.2.** When setting the policy regularization term as $\Omega_s^\eta(\pi) \triangleq \Omega^\eta(s, \pi(\cdot|s)) = \sigma_{\mathcal{R}_s^\eta(\pi)}(-\pi(\cdot|s))$ for all $(\pi, s) \in \Pi \times \mathcal{S}$, where $\mathcal{R}^\eta : \mathcal{S} \times \mathcal{A} \times \Pi \to I \subseteq \mathbb{R}$ denotes the uncertainty set within which the reward deviation $\Delta r = \widetilde{r} - r$ may lie, and $\sigma_{\mathcal{R}_s^\eta(\pi)}(-\pi(\cdot|s)) = \sup_{\Delta r \in \mathcal{R}^\eta(\pi)} \langle -\pi(\cdot|s), \Delta r(s, \cdot) \rangle$. Then for any state $s \in \mathcal{S}$ and any function $V : \mathcal{S} \to \mathbb{R}$, we have

$$\inf_{P \in \Delta(\mathcal{S})} \mathbb{E}_{a \sim \pi(\cdot|s)}\big[ r(s, a) + \mathbb{E}_{s' \sim P(\cdot|s,a)}[V(s')]$$
$$+ \beta \cdot \mathrm{D}\big(P(s, a)\big\|P^o(s, a)\big) - \Omega_s^\eta(\pi)\big]$$
$$= \inf_{\substack{\widetilde{r} \in r + \mathcal{R}^\eta(\pi) \\ P \in \Delta(\mathcal{S})}} \mathbb{E}_{a \sim \pi(\cdot|s)}\big[ \widetilde{r}(s, a) + \mathbb{E}_{s' \sim P(\cdot|s,a)}[V(s')]$$
$$+ \beta \cdot \mathrm{D}\big(P(s, a)\big\|P^o(s, a)\big)\big].$$

**Learning Goal**  We consider the online setting, where an agent aims to learn the optimal robust policy by interacting with the environment over $K$ episodes. At the start of each episode $k$, the agent is given a fixed initial state $s_1^k$. For the policy $\pi^k = \{\pi_h^k\}_{h=1}^H$ selected by the agent based on the history at episode $k$, we measure its sub-optimality by $V_1^{*,\beta,\eta}(s_1) - V_1^{\pi^k,\beta,\eta}(s_1)$. The cumulative sub-optimality after $K$ episodes, which is commonly referred to as $\mathrm{Regret}(K)$, is defined as $\mathrm{Regret}(K) = \sum_{k=1}^K \big(V_1^{*,\beta,\eta}(s_1) - V_1^{\pi^k,\beta,\eta}(s_1)\big)$.

## 4. Algorithm

We first present a meta-algorithm Algorithm 1 for online doubly regularized MDPs with general $f$-divergence based regularization and various types of policy regularization terms. We then instantiate Algorithm 1 in each case. Similar to Lu et al. (2024); Liu & Xu (2024a); He et al. (2025), our algorithm Robust Soft Policy Value Iteration (RSPVI)

adopts a value iteration framework. We leverage the robust Bellman optimality equation and incorporate the optimism principle (Abbasi-Yadkori et al., 2011) to estimate the robust $Q$-functions. For the explicit computation expression, we employ the dual formulation corresponding to each $f$-divergence setting.

---

**Algorithm 1** Robust Soft Policy Value Iteration (RSPVI) for tabular setting

---

**Require:** transition regularizer $\beta$, policy regularizer $\Omega^\eta$.
1: **for** $k = 1, \cdots, K$ **do**
2:     $V_{H+1}^{k,\beta,\eta}(\cdot) \leftarrow 0$.
3:     **for** $h = H, \cdots, 1$ **do**
4:        **for** $\forall (s, a) \in \mathcal{S} \times \mathcal{A}$ **do**
5:           Update $Q$-function estimation $Q_h^{k,\beta,\eta}(s, a)$ according to (4.2).
6:        **end for**
7:        **for** $\forall s \in \mathcal{S}$ **do**
8:           Update policy $\pi_h^k(\cdot|s)$ according to (4.3).
9:           Update value function estimation $V_h^{k,\beta,\eta}(s) = \left\langle Q_h^{k,\beta,\eta}(s, \cdot), \pi_h^k(\cdot|s) \right\rangle - \Omega_s^\eta(\pi_h^k)$.
10:        **end for**
11:     **end for**
12:     Collect a trajectory $\tau^k$ by executing $\pi^k$.
13:     Update model estimation as (4.1).
14: **end for**

---

## 4.1. Model Estimation

We use a model based manner to estimate the empirical reward and transition in the tabular setting. In each episode $k$, after executing policy $\pi^k$ and collecting a trajectory $\tau^k = (s_1^k, a_1^k, r_1^k, \cdots, s_H^k, a_H^k, r_H^k)$, RSPVI updates the empirical estimations as follows

$$n_h^k(s, a) = \sum_{i=1}^k \mathbb{1}\left\{s_h^i = s, a_h^i = a\right\},$$

$$\widehat{r}_h^{k+1}(s, a) = \frac{\sum\limits_{i=1}^k r_h^i(s, a) \cdot \mathbb{1}\{s_h^i = s, a_h^i = a\}}{n_h^k(s, a) \vee 1}, \quad (4.1)$$

$$\widehat{P}_h^{k+1}(s'|s, a) = \frac{\sum\limits_{i=1}^k \mathbb{1}\{s_h^i = s, a_h^i = a, s_{h+1}^i = s'\}}{n_h^k(s, a) \vee 1},$$

where the indicator function $\mathbb{1}\{s_h^i = s, a_h^i = a\} = 1$ if the agent visits the state-action pair $(s, a)$ at step $h$ in episode $i$. $\mathbb{1}\{s_h^i = s, a_h^i = a, s_{h+1}^i = s'\}$ is defined similarly.

## 4.2. $Q$-function Estimation

The estimation of optimal robust $Q$-function $Q_h^{k,\beta,\eta}$, $(h, k) \in [H] \times [K]$ is constructed as follows

$$Q_h^{k,\beta,\eta}(s, a) = \min\left\{\mathrm{RB}_h^k(s, a) + b_h^k(s, a), H - h + 1\right\}, \quad (4.2)$$

which consists of a robust Bellman estimator $\mathrm{RB}_h^k(s, a)$ and a bonus term $b_h^k(s, a)$. In the following, we instantiate this generic form under different $f$-divergence settings, providing explicit formulations for robust Bellman estimation as well as bonus design.

**Tabular-TV Setting** According to the dual formulation for regularized TV setting in Lemma D.3, given the estimated value function $V_{h+1}^{k,\beta,\eta}$ and empirical reward $\widehat{r}_h^k$ and transition $\widehat{P}_h^k$, we choose the robust Bellman estimator $\mathrm{RB}_h^k(s, a)$ and the bonus $b_h^k(s, a)$ as

$$\mathrm{RB}_h^k(s, a) = \widehat{r}_h^k(s, a) - \mathbb{E}_{\widehat{P}_h^k}\left[\left(\min_{s' \in \mathcal{S}} V_{h+1}^{k,\beta,\eta}(s') + \beta \right. \right.$$
$$\left. \left. - V_{h+1}^{k,\beta,\eta}(s)\right)_+\right](s, a) + \left(\min_{s' \in \mathcal{S}} V_{h+1}^{k,\beta,\eta}(s') + \beta\right),$$

$$b_h^k(s, a) = 2H\sqrt{\frac{2S \ln(2SAHK/\delta)}{n_h^{k-1}(s, a) \vee 1}}.$$

**Tabular-KL Setting** According to the dual formulation for regularized KL setting in Lemma D.6, given the estimated value function $V_{h+1}^{k,\beta,\eta}$ and empirical reward $\widehat{r}_h^k$ and transition $\widehat{P}_h^k$, we choose $\mathrm{RB}_h^k(s, a)$ and $b_h^k(s, a)$ as

$$\mathrm{RB}_h^k(s, a) = \widehat{r}_h^k(s, a) - \beta \ln \mathbb{E}_{\widehat{P}_h^k}\left[\exp\left(-\beta^{-1} V_{h+1}^{k,\beta,\eta}\right)\right](s, a),$$

$$b_h^k(s, a) = \left(1 + \beta e^{\beta^{-1}H}\sqrt{S}\right)\sqrt{\frac{2 \ln(2SAHK/\delta)}{n_h^{k-1}(s, a) \vee 1}}.$$

**Tabular-$\chi^2$ Setting** According to the dual formulation for regularized $\chi^2$ setting in Lemma D.9, given the estimated value function $V_{h+1}^{k,\beta,\eta}$ and empirical reward $\widehat{r}_h^k$ and transition $\widehat{P}_h^k$, we choose $\mathrm{RB}_h^k(s, a)$ and $b_h^k(s, a)$ as

$$\mathrm{RB}_h^k(s, a) = \widehat{r}_h^k(s, a) + \sup_{\boldsymbol{\lambda} \in [0,H]}\left(\mathbb{E}_{\widehat{P}_h^k}\left[V_{h+1}^{k,\beta,\eta} - \boldsymbol{\lambda}\right](s, a)\right.$$
$$\left. - 1/(4\beta)\mathrm{Var}_{\widehat{P}_h^k}\left[V_{h+1}^{k,\beta,\eta} - \boldsymbol{\lambda}\right](s, a)\right),$$

$$b_h^k(s, a) = \left(2 + \frac{3H}{4\beta}\right)H\sqrt{\frac{2S^2 \ln(48SAH^3K^2/\delta)}{n_h^{k-1}(s, a) \vee 1}} + \frac{1 + 4\beta}{4\beta K}.$$

## 4.3. Policy Update

For the selection of policy $\pi_h^k$, $(h, k) \in [H] \times [K]$, we first present a generic form (4.3), which maximizes the estimated value function $V_h^k(s)$.

$$\pi_h^k(\cdot|s) = \underset{\pi \in \Pi}{\mathrm{argmax}}\left\{\mathbb{E}_{\pi(\cdot|s)}\left[Q_h^{k,\beta,\eta}(s, \cdot)\right] - \Omega_s^\eta(\pi)\right\}$$
$$= \underset{\pi \in \Pi}{\mathrm{argmax}}\left\{\mathbb{E}_{\pi(\cdot|s)}\left[Q_h^{k,\beta,\eta}(s, \cdot)\right] - \sigma_{\mathcal{R}_s^\eta(\pi)}(-\pi(\cdot|s))\right\}, \quad (4.3)$$

where we specify the choice of policy regularization according to Remark 3.2 in the second equality.

Next, we specify the policy update formulation for different types of policy regularization. The exact form of the reward-uncertainty interval for the associated support function is established in Derman et al. (2021, Section 3).

**Negative Shannon Entropy** Let $\mathcal{R}^{\mathrm{NS},\eta}_{s,a}(\pi) = [\eta \ln(1/\pi(a|s)), +\infty)$ for all $(\pi, s, a) \in \Pi \times \mathcal{S} \times \mathcal{A}$, and the associated support function is $\sigma_{\mathcal{R}^{\mathrm{NS},\eta}_s(\pi)}(-\pi(\cdot|s)) = \eta \cdot \sum_{a \in \mathcal{A}} \pi(a|s) \ln(\pi(a|s))$. We update the policy as

$$\pi^k_h(\cdot|s) \propto \exp(Q^{k,\beta,\eta}_h(s,\cdot)/\eta). \tag{4.4}$$

**Kullback-Leibler Divergence** Fix a reference policy $\widetilde{\pi}$, let $\mathcal{R}^{\mathrm{KL},\eta}_{s,a}(\pi) = \eta \ln(\widetilde{\pi}(a|s)) + \mathcal{R}^{\mathrm{NS},\eta}_{s,a}(\pi)$ for all $(\pi, s, a) \in \Pi \times \mathcal{S} \times \mathcal{A}$, and the associated support function is $\sigma_{\mathcal{R}^{\mathrm{KL},\eta}_s(\pi)}(-\pi(\cdot|s)) = \eta \cdot \sum_{a \in \mathcal{A}} \pi(a|s) \ln(\pi(a|s)/\widetilde{\pi}(a|s))$. For this setting, we update the policy as

$$\pi^k_h(\cdot|s) \propto \widetilde{\pi}^k_h(\cdot|s) \exp(Q^{k,\beta,\eta}_h(s,\cdot)/\eta). \tag{4.5}$$

**Negative Tsallis Entropy** Let $\mathcal{R}^{\mathrm{NT},\eta}_{s,a}(\pi) = \eta[(1 - \pi(a|s))/2, +\infty)$ for all $(\pi, s, a) \in \Pi \times \mathcal{S} \times \mathcal{A}$, and the associated support function is $\sigma_{\mathcal{R}^{\mathrm{NT},\eta}_s(\pi)}(-\pi(\cdot|s)) = \eta \cdot (\|\pi(\cdot|s)\|^2 - 1)/2$. For this setting, we update the policy

$$\pi^k_h(\cdot|s) = \mathrm{proj}_\Delta(Q^{k,\beta,\eta}_h(s,\cdot)/\eta). \tag{4.6}$$

In Lu et al. (2024); Liu & Xu (2024a); He et al. (2025), the updated policy is chosen as the greedy policy with respect to the estimated $Q$-function $Q^{k,\beta}_h$. Unlike prior work, since our formulation incorporates a policy regularization term, the corresponding robust Bellman equations Proposition 6.2 differ from theirs, leading to a different policy update rule. Notably, the formulations in (4.4), (4.5), and (4.6) allow us to employ a soft policy update.

# 5. Theoretical Results

In this section, we present the regret bounds for our proposed algorithm RSPVI. We begin by introducing the assumptions required for online robust learning. Then we discuss our results under different $f$-divergence settings.

He et al. (2025) show that online robust learning can be exponentially hard without appropriate assumptions. They introduce the bounded visitation measure ratio assumption and justify its necessity by deriving the matching lower bounds. Following their approach, we introduce several notations that will be useful throughout the discussion.

**Definition 5.1** (Worst-case transition). In the tabular setting, for any policy $\pi$, we define the worst-case transition corresponding to any $(h, s, a) \in [H] \times \mathcal{S} \times \mathcal{A}$ as

$$P^{w,\pi}_h(\cdot|s,a) = \underset{P_h \in \Delta(S)}{\mathrm{argmin}}\, \mathbb{E}_{P_h}[V^{\pi,\beta,\eta}_{h+1}](s,a)$$
$$+ \beta \cdot \mathrm{D}(P_h(\cdot|s,a)\|P^o_h(\cdot|s,a)).$$

We now define the corresponding visitation measure and make the same bounded visitation measure ratio assumption as in He et al. (2025)

**Definition 5.2** (Visitation measure). At timestep $h \in [H]$, we denote $\mathrm{d}^\pi_h(\cdot)$ as the visitation measure on $\mathcal{S}$ induced by policy $\pi$ under $P^o$, and $\mathrm{q}^\pi_h(\cdot)$ as the visitation measure on $\mathcal{S}$ induced by policy $\pi$ under $P^{w,\pi}$.

**Assumption 5.3** (Bounded visitation measure ratio). Under Definition 5.2, we define $C_{vr} := \sup_{\pi,h,s} \frac{\mathrm{q}^\pi_h(s)}{\mathrm{d}^\pi_h(s)}$ as the supremal ratio between the nominal visitation measure and the worst-case visitation measure. We assume that $C_{vr}$ is polynomial in $H$, $S$ and $A$.

**Theorem 5.4** (Tabular RRMDP regret bounds). Assume that Assumption 5.3 holds for each $f$-divergence-based transition regularization, and consider the policy regularizations in (4.4), (4.5), and (4.6) Then for any $\delta \in (0, 1/3)$, with probability at least $1 - 3\delta$, the regret of Algorithm 1 satisfies

$\mathrm{Regret}(K) =$

$$\begin{cases} \widetilde{\mathcal{O}}(C_{vr}S^{\frac{3}{2}}AH^2 + C_{vr}^{\frac{1}{2}}SA^{\frac{1}{2}}H^2\sqrt{K}) & \text{(TV)} \\ \widetilde{\mathcal{O}}((1 + \beta e^{\beta^{-1}H}\sqrt{S})(C_{vr}SAH + C_{vr}^{\frac{1}{2}}S^{\frac{1}{2}}A^{\frac{1}{2}}H\sqrt{K})) & \text{(KL)} \\ \widetilde{\mathcal{O}}((1 + \frac{H}{\beta})(C_{vr}S^2AH^2 + C_{vr}^{\frac{1}{2}}S^{\frac{3}{2}}A^{\frac{1}{2}}H^2\sqrt{K})) & (\chi^2) \end{cases} \cdot$$

Theorem 5.4 presents the first regret bound for online MDPs that simultaneously accounts for both transition and reward uncertainties, demonstrating that sample-efficient online robust learning is achievable. Unlike transition dynamics, we make no assumptions on reward uncertainty, since reward variations do not affect the difficulty of reaching specific states across the source and target environments. A key distinction of our setting is that the optimal policy is soft-max, whereas in He et al. (2025) it must be greedy. This advantage arises from policy regularization, and often leads to the learned policy more exploratory and reliable.

Our regret bound aligns with that in He et al. (2025), demonstrating that incorporating an additional mechanism for reward robustness does not increase the overall regret. This underscores the sample efficiency of RSPVI and the advantages of our proposed doubly regularized framework. The key insight lies in the fact that in the online setting, we introduce a bonus to the estimated function to ensure optimism, and the estimation error can be bounded by this bonus term. Due to our algorithmic design, the optimism property persists for all estimated value functions and $Q$-functions when

the bonus terms are chosen consistently. Consequently, by applying Assumption 5.3 in the same manner as He et al. (2025), the order of the regret remains unchanged.

# 6. Linear MDP Extension

**Linear Function Approximation** $d$-rectangular linear regularized robust MDP was first proposed in Tang et al. (2025), which changes the uncertainty set into a penalty term while persisting the linear structure of the transitions and reward functions. We make the following assumption the same as Liu & Xu (2024a); Tang et al. (2025).

**Assumption 6.1.** (Jin et al., 2020) Given a known state-action feature mapping $\phi : \mathcal{S} \times \mathcal{A} \to \mathbb{R}^d$ satisfying $\sum_{i=1}^d \phi_i(s, a) = 1, \phi_i(s, a) \geq 0$, we further assume the reward function $\{r_h\}_{h=1}^H$ and the nominal transition kernels $\{P_h^o\}_{h=1}^H$ admit linear structures. Specifically, we have for all $(h, s, a) \in [H] \times \mathcal{S} \times \mathcal{A}$, $r_h(s, a) = \langle \phi(s, a), \theta_h \rangle$, $P_h^o(\cdot|s, a) = \langle \phi(s, a), \mu_h^o(\cdot) \rangle$, where $\{\theta_h\}_{h=1}^H$ are unknown vectors with bounded norm $\|\theta_h\|_2 \leq \sqrt{d}$ and $\{\mu_h^o\}_{h=1}^H$ are unknown probability measure vectors over $\mathcal{S}$, i.e., $\mu_h^o = (\mu_{h,1}^o, \mu_{h,2}^o, \cdots, \mu_{h,d}^o)$, $\mu_{h,i}^o \in \Delta(\mathcal{S}), \forall i \in [d]$.

Based on such linear structure, the robust value function and $Q$-function are defined as

$$V_h^{\pi,\beta,\eta}(s) = \inf_{\substack{\mu_t \in \Delta(S)^d \\ P_t = \langle \phi, \mu_t \rangle}} \mathbb{E}_{\pi, \{P_t\}_{t=h}^H} \left[ \sum_{t=h}^H \left( r_t(s_t, a_t) \right. \right.$$
$$\left. \left. + \beta \cdot \left\langle \phi(s_t, a_t), \mathbf{D}(\mu_t \| \mu_t^o) \right\rangle - \Omega_{s_t}^\eta(\pi_t) \right) \mid s_h = s \right],$$

$$Q_h^{\pi,\beta,\eta}(s, a) = \inf_{\substack{\mu_t \in \Delta(S)^d \\ P_t = \langle \phi, \mu_t \rangle}} \mathbb{E}_{\pi, \{P_t\}_{t=h}^H} \left[ -\sum_{t=h+1}^H \Omega_{s_t}^\eta(\pi_t) + \sum_{t=h}^H \right.$$
$$\left. \left( r_t(s_t, a_t) + \beta \cdot \left\langle \phi(s_t, a_t), \mathbf{D}(\mu_t \| \mu_t^o) \right\rangle \right) \mid s_h = s, a_h = a \right],$$

where $\mathbf{D}(\mu \| \mu^o) = (\mathrm{D}(\mu_1 \| \mu_1^o), \cdots, \mathrm{D}(\mu_d \| \mu_d^o))^\top$.

It is worth noting that there has been no prior framework addressing reward robustness in the linear setting, largely due to the constraints of its linear structure. In contrast, under our doubly regularized framework, reward robustness naturally emerges from the policy regularization, making such robustness much easier to achieve.

**Dynamic Programming Principle** Similar to the tabular setting, we have the dynamic programming principle for doubly regularized linear MDPs as follows

**Proposition 6.2.** For doubly regularized linear MDPs, it holds that for any policy $\pi$ and any $(h, s, a) \in [H] \times \mathcal{S} \times \mathcal{A}$,

$$Q_h^{\pi,\beta,\eta}(s, a) = \inf_{\mu_h \in \Delta(S)^d, P_h = \langle \phi, \mu_h \rangle} \left\{ \mathbb{E}_{s' \sim P_h(\cdot|s,a)}[V_{h+1}^{\pi,\beta,\eta}(s')] \right.$$

$$+ \beta \cdot \left\langle \phi(s, a), \mathbf{D}(\mu_h \| \mu_h^o) \right\rangle \right\} + r_h(s, a), \quad (6.1)$$

$$V_h^{\pi,\beta,\eta}(s) = -\Omega_s^\eta(\pi_h) + \mathbb{E}_{a \sim \pi_h(\cdot|s)}\left[ Q_h^{\pi,\beta,\eta}(s, a) \right]. \quad (6.2)$$

## 6.1. Algorithm

The linear version of RSPVI is displayed in Algorithm 2.

---
**Algorithm 2** RSPVI for linear setting

---
**Require:** transition regularizer $\beta$, policy regularizer $\Omega^\eta$, ridge regression regularizer $\lambda$.
1: **for** $k = 1, \cdots, K$ **do**
2:     $V_{H+1}^{k,\beta,\eta}(\cdot) \leftarrow 0$.
3:     **for** $h = H, \cdots, 1$ **do**
4:        **for** $\forall (s, a) \in \mathcal{S} \times \mathcal{A}$ **do**
5:           Update $Q$-function estimation $Q_h^{k,\beta,\eta}(s, a)$ according to (6.3).
6:        **end for**
7:        **for** $\forall s \in \mathcal{S}$ **do**
8:           Update policy $\pi_h^k(\cdot|s)$ according to (4.3).
9:           Update value function estimation $V_h^{k,\beta,\eta}(s) = \langle Q_h^{k,\beta,\eta}(s, \cdot), \pi_h^k(\cdot|s) \rangle - \Omega_s^\eta(\pi_h^k)$.
10:        **end for**
11:     **end for**
12:     Collect a trajectory $\tau^k$ by executing $\pi^k$.
13: **end for**

---

**$Q$-function Estimation** For the estimation of robust $Q$-function $Q_h^{k,\beta,\eta}$, $(h, k) \in [H] \times [K]$, we first present a generic form (6.3). Since the feature mapping is known, it is sufficient to estimate the weight vectors $\theta_h^k$ and $w_h^k$ to recover the robust $Q$-function. Here, $\theta_h^k$ corresponds to the reward component, and $w_h^k$ corresponds to the estimated value function $V_{h+1}^{k,\beta,\eta}$. Instead of a model-based method in the tabular setting, we use ridge regression to estimate $\theta_h^k$ and $w_h^k$. $\Gamma_h^k(s, a)$ denotes the reward term, where $c$ needs to be determined according to specific $f$-divergence setting. In the following, we instantiate this generic form under different $f$-divergence settings and provide explicit formulations for robust Bellman estimation as well as the coefficient $c$ for the bonus term.

$$Q_h^k(s, a) = \min \left\{ \langle \phi(s, a), \theta_h^k + w_h^k \rangle + \Gamma_h^k(s, a), H - h + 1 \right\}, \quad (6.3)$$

$$\Gamma_h^k(s, a) = c \sum_{i=1}^d \|\phi_i(s, a) \mathbb{1}_i\|_{(\Lambda_h^k)^{-1}},$$

where $\Lambda_h^k = \sum_{\tau=1}^{k-1} \phi(s_h^\tau, a_h^\tau)\phi(s_h^\tau, a_h^\tau)^\top + \lambda I$, $\lambda > 0$ is the regularizer in the ridge regression.

**Linear-TV Setting** According to the linear structure assumption in Assumption 6.1 and the dual formulation for regularized TV setting in Lemma E.3, given the estimated

robust value function $V_{h+1}^{k,\beta,\eta}$, we estimate the parameters $\boldsymbol{\theta}_h^k$ and $\boldsymbol{w}_h^k$ as follows

$$\boldsymbol{\theta}_h^k = \underset{\boldsymbol{\theta} \in \mathbb{R}^d}{\operatorname{argmin}} \sum_{\tau=1}^k \left(r_h^\tau - \boldsymbol{\phi}(s_h^\tau, a_h^\tau)^\top \boldsymbol{\theta}\right)^2 + \lambda \|\boldsymbol{\theta}\|_2^2,$$

$$\boldsymbol{w}_h^k = \underset{\boldsymbol{w} \in \mathbb{R}^d}{\operatorname{argmin}} \sum_{\tau=1}^k \left([V_{h+1}^{k,\beta,\eta}(s_{h+1}^\tau)]_{\min_{s' \in \mathcal{S}} V_{h+1}^{k,\beta,\eta}(s')+\beta} \right.$$
$$\left. - \boldsymbol{\phi}(s_h^\tau, a_h^\tau)^\top \boldsymbol{w}\right)^2 + \lambda \|\boldsymbol{w}\|_2^2.$$

The bonus coefficient is given by $c = Hd \cdot \xi_{\mathrm{TV}}$, where $\xi_{\mathrm{TV}} = 720 + 3\sqrt{40 \log\left(96 K^{13/2} H |\mathcal{A}|^3/\delta\right)}$.

**Linear-KL Setting** According to the linear structure assumption in Assumption 6.1 and the dual formulation for regularized KL setting in Lemma E.6, we estimate $\boldsymbol{w}_h^k$ through two steps. First, given the estimated robust value function $V_{h+1}^{k,\beta,\eta}$, we estimate $\widehat{\mathbb{E}}_{s \sim \boldsymbol{\mu}^o}[e^{-V_{h+1}^{k,\beta,\eta}(s)/\beta}]$ as

$$\widehat{\mathbb{E}}_{s \sim \boldsymbol{\mu}^o}[e^{-V_{h+1}^{k,\beta,\eta}(s)/\beta}] = \underset{\boldsymbol{w} \in \mathbb{R}^d}{\operatorname{argmin}} \sum_{\tau=1}^k \left(e^{-V_{h+1}^{k,\beta,\eta}(s_{h+1}^\tau)/\beta} \right.$$
$$\left. - \boldsymbol{\phi}(s_h^\tau, a_h^\tau)^\top \boldsymbol{w}\right)^2 + \lambda \|\boldsymbol{w}\|_2^2.$$

And then we estimate the parameters $\boldsymbol{\theta}_h^k$ and $\boldsymbol{w}_h^k$ as

$$\boldsymbol{\theta}_h^k = \underset{\boldsymbol{\theta} \in \mathbb{R}^d}{\operatorname{argmin}} \sum_{\tau=1}^k \left(r_h^\tau - \boldsymbol{\phi}(s_h^\tau, a_h^\tau)^\top \boldsymbol{\theta}\right)^2 + \lambda \|\boldsymbol{\theta}\|_2^2,$$

$$\boldsymbol{w}_h^k = -\beta \log \max\left\{\widehat{\mathbb{E}}_{s \sim \boldsymbol{\mu}^o}[e^{-V_{h+1}^{k,\beta,\eta}(s)/\beta}], e^{-H/\beta}\right\}.$$

The truncation here is to make $V_h^k$ lower bounded and therefore the ridge regression operation is well-defined. And the bonus coefficient is given by $c = (1 + 2\beta e^{\beta^{-1}H})Hd \cdot \xi_{\mathrm{KL}}$, where $\xi_{\mathrm{KL}} = 80 + \sqrt{40 \log\left(64\beta K^{11/2} H d^{1/2}(1 + 2\beta e^{\beta^{-1}H})^2 |\mathcal{A}|^3/\delta\right)}$.

**Linear-$\chi^2$ Setting** According to the linear structure assumption in Assumption 6.1 and the dual formulation for regularized $\chi^2$ setting in Lemma E.9, we estimate $\boldsymbol{w}_h^k$ through two steps. First, given the estimated robust value function $V_{h+1}^{k,\beta,\eta}$ and an arbitrary constant $\alpha$, we estimate $\widehat{\mathbb{E}}_{s \sim \boldsymbol{\mu}^o}[V_{h+1}^{k,\beta,\eta}(s)]_\alpha$ and $\widehat{\mathbb{E}}_{s \sim \boldsymbol{\mu}^o}[V_{h+1}^{k,\beta,\eta}(s)]_\alpha^2$ as

$$\widehat{\mathbb{E}}_{s \sim \boldsymbol{\mu}^o}[V_{h+1}^{k,\beta,\eta}(s)]_\alpha = \left[ \underset{\boldsymbol{w} \in \mathbb{R}^d}{\operatorname{argmin}} \sum_{\tau=1}^k \left([V_{h+1}^{k,\beta,\eta}(s_{h+1}^\tau)]_\alpha \right.\right.$$
$$\left.\left. - \boldsymbol{\phi}(s_h^\tau, a_h^\tau)^\top \boldsymbol{w}\right)^2 + \lambda \|\boldsymbol{w}\|_2^2 \right]_{[0,H]},$$

$$\widehat{\mathbb{E}}_{s \sim \boldsymbol{\mu}^o}[V_{h+1}^{k,\beta,\eta}(s)]_\alpha^2 = \left[ \underset{\boldsymbol{w} \in \mathbb{R}^d}{\operatorname{argmin}} \sum_{\tau=1}^k \left([V_{h+1}^{k,\beta,\eta}(s_{h+1}^\tau)]_\alpha^2 \right.\right.$$
$$\left.\left. - \boldsymbol{\phi}(s_h^\tau, a_h^\tau)^\top \boldsymbol{w}\right)^2 + \lambda \|\boldsymbol{w}\|_2^2 \right]_{[0,H]}.$$

Based on the estimated expectations and the dual variable $\alpha$, we estimate the parameters $\boldsymbol{\theta}_h^k$ and $\boldsymbol{w}_h^k$ as

$$\boldsymbol{\theta}_h^k = \underset{\boldsymbol{\theta} \in \mathbb{R}^d}{\operatorname{argmin}} \sum_{\tau=1}^k \left(r_h^\tau - \boldsymbol{\phi}(s_h^\tau, a_h^\tau)^\top \boldsymbol{\theta}\right)^2 + \lambda \|\boldsymbol{\theta}\|_2^2,$$

$$\boldsymbol{w}_h^k = \sup_{\alpha \in [0,H]} \left\{\widehat{\mathbb{E}}_{s \sim \boldsymbol{\mu}^o}[V_{h+1}^{k,\beta,\eta}(s)]_\alpha - \frac{1}{4\beta}\widehat{\mathrm{Var}}_{s \sim \boldsymbol{\mu}^o}[V_{h+1}^{k,\beta,\eta}(s)]_\alpha\right\}$$

$$= \sup_{\alpha \in [0,H]} \left\{\widehat{\mathbb{E}}_{s \sim \boldsymbol{\mu}^o}[V_{h+1}^{k,\beta,\eta}(s)]_\alpha + \frac{1}{4\beta}\left(\widehat{\mathbb{E}}_{s \sim \boldsymbol{\mu}^o}[V_{h+1}^{k,\beta,\eta}(s)]_\alpha\right)^2 \right.$$
$$\left. - \frac{1}{4\beta}\widehat{\mathbb{E}}_{s \sim \boldsymbol{\mu}^o}[V_{h+1}^{k,\beta,\eta}(s)]_\alpha^2\right\}.$$

The bonus coefficient is given by $c = \left(1 + H/(2\beta)\right)Hd \cdot \xi_{\chi^2}$, where $\xi_{\chi^2} = 720 + 3\sqrt{40 \log\left(96 K^6 H^5 (1 + H/(2\beta))^3 |\mathcal{A}|^3/\delta\right)}$.

## 6.2. Theoretical Results

We first introduce the necessary definitions and assumptions.

**Definition 6.3** (Worst-case transition). In the linear setting, since the transition model $P_h^o(\cdot|s,a) = \langle \boldsymbol{\phi}(s,a), \boldsymbol{\mu}_h^o(\cdot)\rangle$ is a factor distribution here, for any policy $\pi$, we first define $\boldsymbol{\mu}_h^{w,\pi} = \left(\mu_{h,1}^{w,\pi}, \cdots, \mu_{h,d}^{w,\pi}\right)^\top$, where for any $i \in [d]$,

$$\mu_{h,i}^{w,\pi}(\cdot) = \underset{\mu_h \in \Delta(S)}{\operatorname{argmin}} \mathbb{E}_{\mu_h}[V_{h+1}^{\pi,\beta,\eta}](s,a) + \beta \cdot \mathrm{D}(\mu_h(\cdot)\|\mu_{h,i}^o(\cdot)).$$

Then we can directly obtain the worst-case transition from the linear structure assumption Assumption 6.1, written as $P_h^{w,\pi}(\cdot|s,a) = \langle \boldsymbol{\phi}(s,a), \boldsymbol{\mu}_h^{w,\pi}(\cdot)\rangle$.

We define the worst-case transition differently in the tabular and linear settings, since in the linear case the transitions additionally admit a linear structure. With such definition in place, we can now naturally define the corresponding visitation measures as follows

**Definition 6.4** (Visitation measure). At timestep $h \in [H]$, we denote $\mathrm{d}_h^\pi(\cdot)$ as the visitation measure on $\mathcal{S}$ induced by policy $\pi$ under $P^o$, and $\mathrm{q}_h^\pi(\cdot)$ as the visitation measure on $\mathcal{S}$ induced by policy $\pi$ under $P^{w,\pi}$.

We now introduce the linear version of the bounded visitation measure ratio assumption as follows

**Assumption 6.5** (Bounded visitation measure ratio). Under Definition 6.4, we define $C_{vr} := \sup_{\pi,h,s} \frac{\mathrm{q}_h^\pi(s)}{\mathrm{d}_h^\pi(s)}$ as the supremal ratio between the nominal visitation measure and the worst-case visitation measure. We assume that $C_{vr}$ is polynomial in $H$, $S$ and $A$.

Note that the formulations in Definition 6.4 and Assumption 6.5 are the same as those in Definition 5.2 and Assumption 5.3 in the tabular setting. This showcases that $C_{vr}$

is an intrinsic quantity characterizing the difficulty of the online robust learning problem, independent of the linear structure assumption. Next, we present our theoretical results in the linear setting. As in Liu & Xu (2024a); Tang et al. (2025), we begin by presenting the instance-dependent upper bounds.

**Theorem 6.6.** Assume Assumption 6.1 and Assumption 6.5 hold for each $f$-divergence-based transition regularization, and consider the policy regularizations in (4.4), (4.5), and (4.6). Then by choosing $\lambda = 1$, for any $\delta \in (0, 1/3)$, with probability at least $1 - 3\delta$, the regret of Algorithm 2 satisfies

$$
\mathrm{Regret}(K) \leq 2C_{vr}\left( \sqrt{2KH^3 \ln(2/\delta)} \right.
$$
$$
\left. + c\sum_{k=1}^{K}\sum_{h=1}^{H}\sum_{i=1}^{d} \left\| \phi_i(s_h^k, a_h^k)\mathbb{1}_i \right\|_{(\Lambda_h^k)^{-1}} \right), \qquad (6.4)
$$

where $c$ is the parameter in bonus coefficients defined in Section 6.1 for each $f$-divergence setting.

Theorem 6.6 provides the first finite-sample regret bound in both online distributionally robust learning under general $f$-divergences with linear function approximation and reward-robust learning. This result demonstrates that when Assumption 6.5 is satisfied, efficient online robust learning is attainable. The formulation of (6.4) is similar to that of Liu & Xu (2024a). However, since we impose the bounded visitation measure assumption rather than the fail-state assumption (Liu & Xu, 2024a, Assumption 4.1), the bound includes a dependence on $C_{vr}$.

**Theorem 6.7.** Assume Assumption 6.1 holds for each $f$-divergence-based transition regularization, and consider the policy regularizations in (4.4), (4.5), and (4.6). Instead of assuming Assumption 6.5, we now assume that

$$
\mathbb{E}_\pi\left[ \phi(s_h, a_h)\phi(s_h, a_h)^\top \right] \geq \alpha I \qquad (6.5)
$$

for all $(\pi, h) \in \Pi \times [H]$, where $\alpha > 0$ is a constant, and $A \succeq B$ means that the matrix $A - B$ is positive semidefinite. Then by choosing $\lambda = 1$, for any $\delta \in (0, 1/3)$, with probability at least $1 - 3\delta$, the regret of Algorithm 2 satisfies

$$
\mathrm{Regret}(K) \leq 2cH\sqrt{K}\sqrt{\frac{128}{\alpha^2}\log\left( \frac{dHK}{\delta} \right) + \frac{2}{\alpha}\log(K)},
$$

where $c$ is the parameter in bonus coefficients defined in Section 6.1 for each $f$-divergence setting.

As discussed in Liu & Xu (2024a), the $d$-rectangular estimation error in (6.4), which arises from the structure of the $d$-rectangular uncertainty set, cannot be bounded directly using the elliptical potential lemma (Abbasi-Yadkori et al., 2011). To address this difficulty, Liu & Xu (2024a, Corollary 5.3) introduce an additional (6.5) assumption. When specialized to the tabular setting, $\phi(s_h, a_h)$ reduces to a one-hot vector. In this case, the diagonal entry of the matrix

$\mathbb{E}_\pi[\phi(s_h, a_h)\phi(s_h, a_h)^\top]$ corresponds to the probability of visiting $(s, a) \in \mathcal{S} \times \mathcal{A}$ at step $h$ under policy $\pi$. Hence, we have $\mathrm{d}_h^\pi(s, a) \geq \alpha$ for all $(\pi, h, s, a) \in \Pi \times [H] \times \mathcal{S} \times \mathcal{A}$, which directly implies Assumption 5.3. In the linear setting, (6.5) can be interpreted as a uniform coverage condition, requiring the nominal environment to be sufficiently exploratory. Theorem 6.7 shows that (6.5) can also serve as a sufficient condition for online robust learning.

Compared with Liu & Xu (2024a, Corollary 5.3), their result additionally relies on a fail-state assumption (Liu & Xu, 2024a, Assumption 4.1) in addition to (6.5). Moreover, their result is restricted to transition uncertainty characterized by Total Variation, whereas Theorem 6.7 extends to more general $f$-divergence-based transition robustness.

# 7. Experiments

In this section, we demonstrate the robust performance of RSPVI by evaluating its tabular variant (Algorithm 1) in the Inverted Pendulum and Inverted Double Pendulum environments, and its linear variant (Algorithm 2) in the CartPole environment from OpenAI Gym. We adopt TV and KL divergence as transition regularizers, and negative Shannon entropy as the policy regularizer. These policy regularizers are chosen because they admit closed-form dual formulations (see Lemmas D.3 and D.6), thereby eliminating the need to solve additional optimization problems within the robust Bellman operator. All reported results are averaged over 10 random seeds, with shaded regions indicating the standard deviation. Additional experimental details and results are provided in Appendix B. To the best of our knowledge, this work represents the first attempt to make the DRMDP framework practical.

**Tabular Setting** We evaluate the tabular version of RSPVI in the Inverted Pendulum and Inverted Double Pendulum environments. To handle large state and action spaces, we employ general function approximation techniques, using neural networks to approximate both the $Q$-function and the policy. Our implementation follows the Soft Actor-Critic (SAC) framework (Haarnoja et al., 2018), in which the $Q$-function and policy networks are updated alternately via stochastic gradient descent.

For $(s, a)$-rectangular KL-RRMDPs, we introduce a new neural parameterization for value estimation (B.3) based on the robust Bellman operator. Instead of directly estimating the robust value function, we target an exponential transformation. This parameterization is accompanied by several stability-oriented design choices, including (1) a clipping to avoid numerical issues when computing $\log f_\theta(s, a)$; (2)

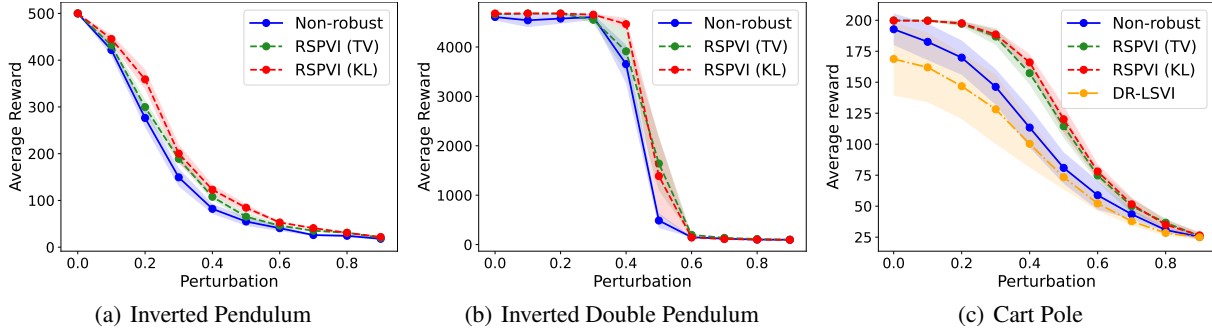

| (a) Inverted Pendulum | (b) Inverted Double Pendulum | (c) Cart Pole |

*Figure 1.* Experimental results for RSPVI, with shaded regions indicating the standard deviation. Figures 1(a) and 1(b) correspond to the tabular version of RSPVI, while Figure 1(c) corresponds to the linear version. The $x$-axis represents the perturbation level during evaluation. The $y$-axis represents the average cumulative reward achieved over 50 episodes.

maintaining two $Q$-networks as in SAC, but taking their maximum rather than minimum, since the network estimates $\exp(-Q(s,a)/\beta)$; (3) introducing a separate value-network $g_\eta$ to reduce estimation variance and bias. These design choices arise naturally from the structure of the robust objective.

We compare RSPVI against its non-robust counterpart. Existing algorithms in the literature (Lu et al., 2024; He et al., 2025) are not included because they rely on deterministic greedy policies and therefore cannot accommodate continuous action spaces with infinitely many actions. To evaluate policy robustness, we introduce disturbances during evaluation. For the Inverted Pendulum environment, action perturbations are applied by replacing the agent's chosen action with a random action with a fixed probability. For the Inverted Double Pendulum environment, robustness is assessed under both actuator degradation and external disturbances. Specifically, the executed action is scaled as $a' = a * (1 - \text{perturbation})$, simulating reduced actuator effectiveness, and a constant leftward acceleration of $3\text{m/s}^2$ is applied to the system. From Figures 1(a) and 1(b), we observe that our robust algorithm outperforms the non-robust baseline.

**Linear Setting**   We further evaluate the linear version of RSPVI in the CartPole environment. In practice, the linear feature mapping $\phi(s,a)$ in Assumption 6.1, which is constrained to lie on a $d$-dimensional probability simplex, is unknown. Despite existing papers (Agarwal et al., 2020; Uehara et al., 2021; Ren et al., 2022; Zhang et al., 2022; Shribak et al., 2024) study representation learning for linear MDPs, their approaches do not directly apply to our setting. To accommodate this structure, we propose to learn $\phi$ via a discrete Variational Autoencoder (VAE). We further incorporate entropy regularization to prevent representation collapse, and separate representation learning from policy learning through a warm-up phase to improve training sta-

bility. Given the learned feature, we estimate the robust $Q$-function via $d$ ridge regressions. With this high-dimensional feature representation, computing the exploration bonus in Algorithm 2 becomes computationally expensive and constitutes the main runtime bottleneck. As a result, we omit the bonus term in our implementation, at the cost of potentially mild performance degradation. During evaluation, we again apply action perturbations.

As shown in Figure 1(c), RSPVI exhibits significantly lower sensitivity to perturbations compared to the non-robust baseline. We also compare RSPVI with DR-LSVI-UCB (Liu & Xu, 2024a), and observe that DR-LSVI-UCB fails to learn a robust policy without its exploration bonus. In contrast, the soft policies learned by RSPVI exhibit inherent exploratory behavior. These results further highlight the advantages of soft policies and our doubly regularized framework in large state-action spaces.

# 8. Conclusion

We propose a novel doubly regularized framework that achieves both reward robustness and transition robustness. Within this framework, we introduce an online learning algorithm, RSPVI, and prove that when using general $f$-divergence-based transition regularization together with various policy regularization formulations, the sample complexity remains no worse than that of traditional RMDPs. We further extend RSPVI to the linear setting, establishing the first finite-sample regret guarantee under general $f$-divergence. Compared with prior work, our results require fewer structural assumptions. Finally, we develop practical algorithms within the proposed framework and demonstrate its effectiveness and robustness through extensive experiments.

## Impact Statement

This paper presents work whose goal is to advance the field of Machine Learning. There are many potential societal consequences of our work, none which we feel must be specifically highlighted here.

## Acknowledgments

ZL and PX are supported in part by the National Science Foundation (DMS-2323112) and the Whitehead Scholars Program at the Duke University School of Medicine. We would like to thank the anonymous reviewers for their constructive feedback on this document. The views and conclusions contained in this paper are those of the authors and should not be interpreted as representing any funding agencies.

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

## A. Additional Related Works

**Reward Robust MDPs** Most existing works focus solely on transition robustness. For studies addressing reward robustness, notable examples include Zhou et al. (2021), Wang et al. (2023), and Wang et al. (2024b). All of these studies formulate reward robustness through an uncertainty set. Specifically, Zhou et al. (2021) proposed a model-based value iteration algorithm using a pre-collected dataset, while Wang et al. (2023) introduced a model-free $Q$-learning algorithm under the assumption of access to a simulator. Building on this, Wang et al. (2024b) incorporated a variance-reduction technique into $Q$-learning, also assuming access to a generative model. However, none of these works address the more practical online setting. Eysenbach & Levine (2021) and Derman et al. (2021) study reward robustness through the lens of policy regularization, which we provide a detailed comparison in the following.

**MDPs with Policy Regularization** Eysenbach & Levine (2021) and Derman et al. (2021) also consider the policy regularization, but our work differs from theirs in several key respects. First, Eysenbach & Levine (2021) interpret policy regularization through the lens of reward or transition robustness with a transformed reward, while Derman et al. (2021) impose regularization on both the policy and the value function to achieve transition robustness. In contrast, we explicitly model transition uncertainty through transition regularization. Second, our approach admits closed-form updates under both TV and KL-divergence regularization (Lemmas D.3 and D.6), leading to a more computationally efficient algorithm that is straightforward to implement. Third, Derman et al. (2021) establish the existence of a unique optimal policy and show that their algorithm converges at the same rate as in standard MDPs, yielding an asymptotic convergence guarantee. By contrast, Eysenbach & Levine (2021) focus on policy-regularized algorithms that optimize a bound on a robust RL objective but do not provide a convergence analysis. Our work instead derives a finite-sample regret bound, providing a non-asymptotic convergence guarantee, which is substantially more challenging to obtain.

## B. Additional Details on Experiments

Here, we provide more experimental results and details in addition to Section 7.

### B.1. Practical Algorithm for RSPVI in the Tabular Setting

The hyper-parameters used in the experiments are reported in Tables 1 and 2. Next, we provide additional details on the function approximation setting and the corresponding update rules.

**TV distance setting.** We consider a parameterized soft $Q$-function $Q_\theta(s_t, a_t)$ and a policy $\pi_\phi(a_t \mid s_t)$ with parameters $\theta$ and $\phi$ respectively. From Lemma D.3, the robust Bellman equation under TV distance uncertainty is given by

$$Q(s,a) = r(s,a) + \gamma \cdot \inf_{P \in \Delta(S)} \left( \mathbb{E}_P[V'] + \beta \mathrm{TV}(P \| P^o) \right)(s,a)$$

$$= r(s,a) + \gamma \cdot \left\{ \mathbb{E}_{s' \sim P^o(\cdot|s,a)} \left[ V'(s') \right]_{V_{\min} + \beta} \right\}.$$

For comparison, the non-robust Bellman update is

$$Q(s,a) = r(s,a) + \gamma \cdot \mathbb{E}_{s' \sim P^o(\cdot|s,a)} \left[ V'(s') \right].$$

The robust update differs from the non-robust case only by an additional truncation applied to the value function, which avoids the need to solve an additional optimization problem and makes the update more computationally efficient.

The soft $Q$-function is trained by minimizing the robust Bellman residual,

$$J_Q(\theta) = \mathbb{E}_{(s_t,a_t) \sim \mathcal{D}} \left[ \frac{1}{2} \left( Q_\theta(s_t, a_t) - r(s_t, a_t) - \gamma \, \mathbb{E}_{s_{t+1} \sim p} \left[ V_{\bar\theta}(s_{t+1}) \right]_{V_{\min} + \beta} \right)^2 \right],$$

where the value function $V_{\bar\theta}$ is implicitly defined via the soft $Q$-function according to (3.2). A target network with parameters $\bar\theta$ is maintained as an exponential moving average of $\theta$ to stabilize training (Haarnoja et al., 2018).

The policy parameters $\phi$ are updated according to (4.3). Under negative Shannon entropy policy regularization, the policy update coincides with that of Soft Actor-Critic (Haarnoja et al., 2018). In particular, the policy is obtained by solving

$$\pi_{\text{new}} = \arg\min_{\pi' \in \Pi} D_{\text{KL}}\left(\pi'(\cdot|s_t) \middle\| \frac{\exp\left(Q^{\pi_{\text{old}}}(s_t, \cdot)/\alpha\right)}{Z^{\pi_{\text{old}}}(s_t)}\right),$$

which in practice is implemented by minimizing the following objective

$$J_\pi(\phi) = \mathbb{E}_{s_t \sim \mathcal{D}}\left[\mathbb{E}_{a_t \sim \pi_\phi}\left[\alpha \log\left(\pi_\phi(a_t|s_t)\right) - Q_\theta(s_t, a_t)\right]\right].$$

**KL divergence setting.** From Lemma D.6, the robust Bellman equation under KL divergence uncertainty is given by

$$Q(s, a) = r(s, a) + \gamma \cdot \inf_{P \in \Delta(S)}\left(\mathbb{E}_P[V'] + \beta \text{KL}\left(P\|P^o\right)\right)(s, a)$$

$$= r(s, a) - \gamma \cdot \beta \ln \mathbb{E}_{P^o}\left[e^{-\beta^{-1}V'(s')}\right]. \tag{B.1}$$

We see that (B.1) also admits a closed-form solution. However, in contrast to the TV distance setting, the right-hand side of (B.1) is nonlinear in the value function. Direct minimization of the corresponding Bellman residual would therefore lead to biased Monte Carlo estimates when sampling transitions from $P^o$.

To obtain an unbiased estimator, we exponentiate both sides of (B.1) and rewrite the update as

$$e^{-\gamma^{-1}\beta^{-1}Q(s,a)} = e^{-\gamma^{-1}\beta^{-1}r(s,a)} \cdot \mathbb{E}_{P^o}\left[e^{-\beta^{-1}V'(s')}\right]. \tag{B.2}$$

Due to the nonlinearity in (B.2), we introduce a separate value network $V_\eta(s)$ with parameters $\eta$. The resulting function approximation is defined as

$$\begin{cases} f_{\theta'}(s_t, a_t) = e^{-\beta^{-1}\gamma^{-1}Q_\theta(s_t, a_t)}, \\ g_{\eta'}(s_t) = -V_\eta(s_t), \end{cases} \tag{B.3}$$

which transforms the robust Bellman equation into a linear expectation with respect to the transition dynamics.

The $Q$-function and value networks are trained by minimizing the corresponding robust Bellman residuals. Substituting (B.3) into (B.2), the loss functions are given by

$$J_g(\eta') = \mathbb{E}_{s_t}\left[\frac{1}{2}\left(g_{\eta'}(s_t) - \mathbb{E}_{a_t}\left(\beta\gamma \log f_{\theta'}(s_t, a_t) + \alpha \log \pi_\phi(a_t|s_t)\right)\right)^2\right],$$

$$J_f(\theta') = \mathbb{E}_{(s_t, a_t) \sim \mathcal{D}}\left[\frac{1}{2}\left(f_{\theta'}(s_t, a_t) - e^{-\beta^{-1}\gamma^{-1}r(s_t, a_t)} \cdot \mathbb{E}_{s_{t+1} \sim p}\left[e^{\beta^{-1}g_{\eta'}(s_{t+1})}\right]\right)^2\right],$$

$$J_\pi(\phi) = \mathbb{E}_{s_t \sim \mathcal{D}}\left[\mathbb{E}_{a_t \sim \pi_\phi}\left[\alpha \log\left(\pi_\phi(a_t|s_t)\right) + \beta\gamma \log f_{\theta'}(s_t, a_t)\right]\right].$$

### B.2. Practical Algorithm for RSPVI in the Linear Setting

The hyper-parameters used in the experiments are reported in Table 3. In practice, we find that the ridge regression coefficient $\lambda$ plays a crucial role in training and is tuned from $\{1, 0.1, 0.01, 0.001, 0.0001\}$.

We begin by providing additional details on how the linear feature mapping $\phi$ is learned via a discrete VAE.

**More Details on the Discrete VAE** The VAE consists of an encoder $q(z|s, a, s')$, a decoder $p(s'|z)$, and an auxiliary distribution $p(z|s, a)$, where $z$ denotes a latent variable. Noting that the transition model $P_h^o(\cdot|s, a) = \langle\phi(s, a), \mu_h^o(\cdot)\rangle$ can be interpreted as a mixture distribution, we associate $p(z|s, a)$ with $\phi(s, a)$ and $p(s'|z)$ with $\mu_h^o(z)$. Under this formulation, the VAE is able to approximately capture the underlying linear structure.

We also conduct ablation study to show the impact of $\eta$ on the algorithm's performance with negative Shannon entropy policy regularization.

**Negative Shannon Entropy**  As shown in Figures 2(a) and 2(d), it is clear that selecting an appropriate value for $\eta$ is crucial for algorithm training. When $\eta$ is too large, the excessive penalty on policy regularization can drive the policy toward near-random behavior, preventing the agent from gathering sufficient useful information during exploration. Conversely, when $\eta$ is too small, the policy becomes less stochastic, which reduces exploration and may lead to the agent learning a suboptimal policy.

Since our theoretical framework also covers Kullback-Leibler divergence and negative Tsallis entropy policy regularization, we conduct additional experiments to evaluate the performance of RSPVI under these two settings.

**Kullback-Leibler Divergence**  In Figures 2(b) and 2(e), we present the results of RSPVI with KL divergence policy regularization for different values of $\eta$. The reference policy is learned with NS policy regularization but without transition regularization, making it non-robust to environmental perturbations. As $\eta$ increases, the stronger policy regularization brings the learned policy closer to the reference policy. However, the overall performance of the learned policy still surpasses that of the reference policy. Further, the KL-divergence defined transition regularization leads to better robustness performance compared to the TV case.

**Negative Tsallis Entropy**  In Figures 2(c) and 2(f), we present the results of RSPVI with negative Tsallis entropy policy regularization for different values of $\eta$. With TV-distance defined transition regularization, similar to the behavior observed with Negative Shannon Entropy, the policy can be too random for large $\eta$ and becomes too deterministic for small $\eta$, while a moderate $\eta$ lead to the optimal performance. While for the case with KL-divergence defined transition regularization, our algorithm is not sensitive to the choice of $\eta$.

In general, this CartPole experiment suggests that our robust algorithm RSPVI outperforms the non-robust baseline.

*Table 1.* Hyperparameters used in experiments on the Inverted Pendulum task.

| Symbol | Description | Value |
|---|---|---|
| $K$ | total training steps | $1 \times 10^5$ |
| lr | learning rate | $3 \times 10^{-4}$ |
| | number of hidden layers | 2 |
| | hidden layer dimension | 256 |
| | batch size | 256 |
| | nonlinearity | ReLU |
| $\tau$ | target smoothing coefficient | 0.005 |
| $\gamma$ | discount factor | 0.99 |
| $\beta_{\text{TV}}$ | transition regularization parameter | 85 |
| $\beta_{\text{KL}}$ | transition regularization parameter | 90 |

*Table 2.* Hyperparameters used in experiments on the Inverted Double Pendulum task.

| Symbol | Description | Value |
|---|---|---|
| $K$ | total training steps | $1 \times 10^5$ |
| lr | learning rate | $3 \times 10^{-4}$ |
| | number of hidden layers | 2 |
| | hidden layer dimension | 256 |
| | batch size | 256 |
| | nonlinearity | ReLU |
| $\tau$ | target smoothing coefficient | 0.005 |
| $\gamma$ | discount factor | 0.99 |
| $\beta_{\text{TV}}$ | transition regularization parameter | 80 |
| $\beta_{\text{KL}}$ | transition regularization parameter | 100 |

*Table 3.* Hyperparameters used in experiments on the CartPole task.

| Symbol | Description | Value |
|:---:|:---:|:---:|
| $d$ | latent space dimension | 80 |
| $K$ | total episodes | 2000 |
| $\lambda$ | ridge regression parameter | 0.001 |
| $\gamma$ | discount factor | 0.99 |
| $\beta_{\text{TV}}$ | transition regularization parameter | 70 |
| $\beta_{\text{KL}}$ | transition regularization parameter | 120 |

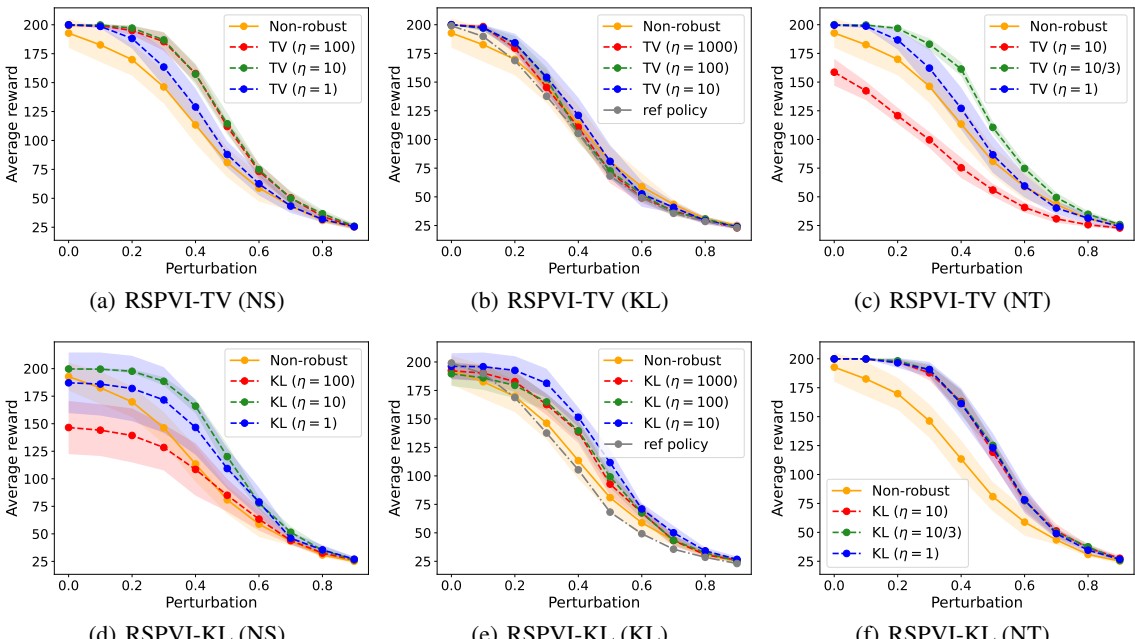

*Figure 2.* Performance of RSPVI with different types of policy regularization and values of $\eta$. Figures 2(a) to 2(c) use TV transition regularization, while Figures 2(d) to 2(f) use KL transition regularization. Additionally, Figures 2(a) and 2(d) apply negative Shannon entropy (NS) policy regularization, Figures 2(b) and 2(e) apply Kullback-Leibler divergence (KL) policy regularization, and Figures 2(c) and 2(f) apply negative Tsallis entropy (NT) policy regularization. The $x$-axis represents the perturbation level during evaluation, and the $y$-axis represents the average cumulative reward achieved over 50 episodes.

## C. Proof of the Policy Update Formulation

In particular, we prove that the policy update formulation in (4.4), (4.5), (4.6) satisfies (4.3).

*Proof of the Kullback-Leibler Divergence case.* Fixing a reference policy $\widetilde{\pi}$, we need to prove that policy

$$\pi(\cdot|s) \propto \widetilde{\pi}_h^k(\cdot|s) \exp\left(Q_h^{k,\beta,\eta}(s,\cdot)/\eta\right)$$

maximizes the objective $J(\pi)$

$$J(\pi) = \left\langle Q_h^{k,\beta,\eta}(s,\cdot), \pi(\cdot|s)\right\rangle - \eta \cdot \sum_{a\in\mathcal{A}} \pi(a|s)\ln\left(\frac{\pi(a|s)}{\widetilde{\pi}_h^k(a|s)}\right).$$

In order to prove this, we define an auxiliary policy $\pi'$ as

$$\pi'(a|s) = \frac{\widetilde{\pi}_h^k(a|s)\exp\left(Q_h^{k,\beta,\eta}(s,a)/\eta\right)}{Z}, \quad \text{where} \quad Z = \sum_a \widetilde{\pi}_h^k(a|s)\exp\left(Q_h^{k,\beta,\eta}(s,a)/\eta\right).$$

From the non-negativity of KL divergence, we have

$$0 \leq \sum_a \pi(a|s) \log\left(\frac{\pi(a|s)}{\pi'(a|s)}\right)$$

$$= \sum_a \pi(a|s) \log\left(\frac{\pi(a|s)}{\widetilde{\pi}_h^k(a|s)}\right) - \frac{1}{\eta}\sum_a \pi(a|s) Q_h^{k,\beta,\eta}(s,a) + \log Z.$$

Rearrange this and it follows that

$$\sum_a \pi(a|s) Q_h^{k,\beta,\eta}(s,a) - \eta\sum_a \pi(a|s) \log\left(\frac{\pi(a|s)}{\widetilde{\pi}_h^k(a|s)}\right) \leq \eta \log Z.$$

The equality is achieved if and only if $\mathrm{KL}(\pi \| \pi') = 0$, or $\pi = \pi'$. Since $J(\pi)$ is strictly concave with respect to $\pi$, $\pi'$ is the unique maximizer of $J(\pi)$. $\qquad\square$

*Proof of the Negative Shannon Entropy case.* We need to prove that policy

$$\pi(\cdot|s) \propto \exp\left(Q_h^{k,\beta,\eta}(s,\cdot)/\eta\right)$$

maximizes the objective $J(\pi)$

$$J(\pi) = \left\langle Q_h^{k,\beta,\eta}(s,\cdot), \pi(\cdot|s)\right\rangle - \eta \cdot \sum_{a\in\mathcal{A}} \pi(a|s) \ln\left(\pi(a|s)\right).$$

The result is directly followed by the Kullback-Leibler Divergence case by choosing $\widetilde{\pi}(\cdot|s)$ as the uniform distribution. $\qquad\square$

*Proof of the Negative Tsallis Entropy case.* We need to prove that policy

$$\pi(\cdot|s) = \mathrm{proj}_\Delta\left(Q_h^{k,\beta,\eta}(s,\cdot)/\eta\right)$$

maximizes the objective $J(\pi)$

$$J(\pi) = \left\langle Q_h^{k,\beta,\eta}(s,\cdot), \pi(\cdot|s)\right\rangle - \frac{\eta}{2}\|\pi(\cdot|s)\|^2,$$

In order to prove this, we rewrite the objective as

$$J(\pi) = \left\langle Q_h^{k,\beta,\eta}(s,\cdot), \pi(\cdot|s)\right\rangle - \frac{\eta}{2}\|\pi(\cdot|s)\|^2$$

$$= -\frac{\eta}{2}\left\|\pi(\cdot|s) - \frac{1}{\eta}Q_h^{k,\beta,\eta}(s,\cdot)\right\|^2 + \frac{1}{2\eta}\|Q_h^{k,\beta,\eta}(s,\cdot)\|^2.$$

Since $\|Q_h^{k,\beta,\eta}(s,\cdot)\|^2/(2\eta)$ is independent of $\pi$, it is equivalent to minimize $\|\pi(\cdot|s) - Q_h^{k,\beta,\eta}(s,\cdot)/\eta\|^2$, subject to $\pi(\cdot|s)$ being a distribution. As the projection operator $\mathrm{proj}$ minimizes the distance, its solution is

$$\pi(\cdot|s) = \mathrm{proj}_\Delta\left(Q_h^{k,\beta,\eta}(s,\cdot)/\eta\right).$$

This finishes the proof. $\qquad\square$

# D. Proofs of the Results in Tabular Setting

## D.1. Proof of the Dynamic Programming Principle

*Proof of Proposition 3.1.* We prove a stronger version of the proposition. Specifically, there exists transition kernels $\{\widetilde{P}_t\}_{t=1}^H$, such that for all $(h,s) \in [H] \times \mathcal{S}$,

$$V_h^{\pi,\beta,\eta}(s) = \mathbb{E}_{\{\widetilde{P}_t\}_{t=h}^H}\left[\sum_{t=h}^H \left(r_t(s_t,a_t) - \Omega_{s_t}^\eta(\pi_t) + \beta \cdot \mathrm{D}\big(P_t(\cdot|s_t,a_t)\|P_t^o(\cdot|s_t,a_t)\big)\right)\Big| s_h = s, \pi_{h:H}\right]. \tag{D.1}$$

We prove this statement by induction. For the base case $h = H$, the claim holds trivially since no transition kernels are involved. For the inductive step, assume the conclusion holds for step $h + 1$, meaning that there exists $\{\widetilde{r}_t\}_{t=1}^H$ and $\{\widetilde{P}_t\}_{t=h+1}^H$ such that

$$V_{h+1}^{\pi,\beta,\eta}(s) = \mathbb{E}_{\{\widetilde{P}_t\}_{t=h+1}^H}\left[\sum_{t=h+1}^H \left(r_t(s_t,a_t) - \Omega_{s_t}^\eta(\pi_t) + \beta \cdot \mathrm{D}\big(P_t(\cdot|s_t,a_t)\|P_t^o(\cdot|s_t,a_t)\big)\right)\Big| s_{h+1} = s, \pi_{h+1:H}\right]. \tag{D.2}$$

We now prove (D.1) for the case of $h$. For $Q_h^{\pi,\beta,\eta}(s,a)$, on the one hand, we upper bound it as

$$Q_h^{\pi,\beta,\eta}(s,a)$$

$$= \inf_{P_t \in \Delta(S)} \mathbb{E}_{\{P_t\}_{t=h}^H}\left[\sum_{t=h}^H \left(r_t(s_t,a_t) + \beta \cdot \mathrm{D}\big(P_t(\cdot|s_t,a_t)\|P_t^o(\cdot|s_t,a_t)\big)\right) - \sum_{t=h+1}^H \Omega_{s_t}^\eta(\pi_t)\Big| s_h = s, a_h = a, \pi_{h+1:H}\right]$$

$$= r_h(s,a) + \inf_{P_t \in \Delta(S)}\left\{\beta \cdot \mathrm{D}\big(P_h(\cdot|s_h,a_h)\|P_h^o(\cdot|s_h,a_h)\big) + \int_S P_h(\mathrm{d}s'|s,a)\mathbb{E}_{\{P_t\}_{t=h+1}^H}\left[\right.\right.$$

$$\sum_{t=h+1}^H \left(r_t(s_t,a_t) - \Omega_{s_t}^\eta(\pi_t) + \beta \cdot \mathrm{D}\big(P_t(\cdot|s_t,a_t)\|P_t^o(\cdot|s_t,a_t)\big)\right)\Big| s_{h+1} = s', \pi_{h+1:H}\left.\right]\right\}$$

$$\le r_h(s,a) + \inf_{P_h \in \Delta(S)}\left\{\beta \cdot \mathrm{D}\big(P_h(\cdot|s_h,a_h)\|P_h^o(\cdot|s_h,a_h)\big) + \int_S P_h(\mathrm{d}s'|s,a)\mathbb{E}_{\{\widetilde{P}_t\}_{t=h+1}^H}\left[\right.\right.$$

$$\sum_{t=h+1}^H \left(r_t(s_t,a_t) - \Omega_{s_t}^\eta(\pi_t) + \beta \cdot \mathrm{D}\big(\widetilde{P}_t(\cdot|s_t,a_t)\|P_t^o(\cdot|s_t,a_t)\big)\right)\Big| s_{h+1} = s', \pi_{h+1:H}\left.\right]\right\} \tag{D.3}$$

$$= r_h(s,a) + \inf_{P_h \in \Delta(S)}\left\{\beta \cdot \mathrm{D}\big(P_h(\cdot|s_h,a_h)\|P_h^o(\cdot|s_h,a_h)\big) + \mathbb{E}_{s'\sim P_h^o(\cdot|s,a)}\big[V_{h+1}^{\pi,\beta,\eta}(s')\big]\right\}, \tag{D.4}$$

where (D.3) is because the definition of the infimum operator and (D.4) is from (D.2).

On the other hand, we can lower bound $Q_h^{\pi,\beta,\eta}(s,a)$ as

$$Q_h^{\pi,\beta,\eta}(s,a)$$

$$= \inf_{P_t \in \Delta(S)} \mathbb{E}_{\{P_t\}_{t=h}^H}\left[\sum_{t=h}^H \left(r_t(s_t,a_t) + \beta \cdot \mathrm{D}\big(P_t(\cdot|s_t,a_t)\|P_t^o(\cdot|s_t,a_t)\big)\right) - \sum_{t=h+1}^H \Omega_{s_t}^\eta(\pi_t)\Big| s_h = s, a_h = a, \pi_{h+1:H}\right]$$

$$= r_h(s,a) + \inf_{P_t \in \Delta(S)}\left\{\beta \cdot \mathrm{D}\big(P_h(\cdot|s_h,a_h)\|P_h^o(\cdot|s_h,a_h)\big) + \int_S P_h(\mathrm{d}s'|s,a)\mathbb{E}_{\{P_t\}_{t=h+1}^H}\left[\right.\right.$$

$$\sum_{t=h+1}^H \left(r_t(s_t,a_t) - \Omega_{s_t}^\eta(\pi_t) + \beta \cdot \mathrm{D}\big(P_t(\cdot|s_t,a_t)\|P_t^o(\cdot|s_t,a_t)\big)\right)\Big| s_{h+1} = s', \pi_{h+1:H}\left.\right]\right\}$$

$$\ge r_h(s,a) + \inf_{P_h \in \Delta(S)}\left\{\beta \cdot \mathrm{D}\big(P_h(\cdot|s_h,a_h)\|P_h^o(\cdot|s_h,a_h)\big) + \int_S P_h(\mathrm{d}s'|s,a)\inf_{P_t \in \Delta(S)}\mathbb{E}_{\{P_t\}_{t=h+1}^H}\left[\right.\right.$$

$$\sum_{t=h+1}^H \left(r_t(s_t,a_t) - \Omega_{s_t}^\eta(\pi_t) + \beta \cdot \mathrm{D}\big(P_t(\cdot|s_t,a_t)\|P_t^o(\cdot|s_t,a_t)\big)\right)\Big| s_{h+1} = s', \pi_{h+1:H}\left.\right]\right\} \tag{D.5}$$

$$= r_h(s,a) + \inf_{P_h \in \Delta(S)}\left\{\beta \cdot \mathrm{D}\big(P_h(\cdot|s_h,a_h)\|P_h^o(\cdot|s_h,a_h)\big) + \mathbb{E}_{s'\sim P_h^o(\cdot|s,a)}\big[V_{h+1}^{\pi,\beta,\eta}(s')\big]\right\}, \tag{D.6}$$

where (D.5) is because $\inf \int f \ge \int \inf f$ and (D.6) is from the definition of $V_{h+1}^{\pi,\beta,\eta}(s')$.

By combining (D.4) and (D.6), we have

$$Q_h^{\pi,\beta,\eta}(s,a) = r_h(s,a) + \inf_{\substack{\mu_h \in \Delta(S)^d \\ P_h = \langle \phi, \mu_h \rangle}}\left\{\mathbb{E}_{s'\sim P_h^o(\cdot|s,a)}\big[V_{h+1}^{\pi,\beta,\eta}(s')\big] + \beta \cdot \mathrm{D}\big(P_h(\cdot|s_h,a_h)\|P_h^o(\cdot|s_h,a_h)\big)\right\},$$

which implies (3.1).

Since $\Delta(\mathcal{S})$ is compact, there exists $\widetilde{P}_h$, such that

$$Q_h^{\pi,\beta,\eta}(s,a) = r_h(s,a) + \mathbb{E}_{s' \sim \widetilde{P}_h(\cdot|s,a)}\big[V_{h+1}^{\pi,\beta,\eta}(s')\big] + \beta \cdot \mathrm{D}\big(\widetilde{P}_h(\cdot|s,a)\big\|P_h^o(\cdot|s,a)\big). \tag{D.7}$$

This finishes the proof of (D.1).

For $V_h^{\pi,\beta,\eta}(s)$, on the one hand, we upper bound it as

$$V_h^{\pi,\beta,\eta}(s)$$

$$= \inf_{P_t \in \Delta(S)} \mathbb{E}_{\{P_t\}_{t=h}^H}\left[ \sum_{t=h}^H \Big( r_t(s_t,a_t) - \Omega_{s_t}^{\eta}(\pi_t) + \beta \cdot \mathrm{D}\big(P_t(\cdot|s_t,a_t)\big\|P_t^o(\cdot|s_t,a_t)\big)\Big) \Big| s_h = s, \pi_{h:H}\right]$$

$$= \inf_{P_t \in \Delta(S)} \bigg\{ -\Omega_s^{\eta}(\pi_h) + \sum_{a \in \mathcal{A}} \pi_h(a|s)\mathbb{E}_{\{P_t\}_{t=h}^H}\bigg[$$

$$\sum_{t=h}^H \Big( r_t(s_t,a_t) + \beta \cdot \mathrm{D}\big(P_t(\cdot|s_t,a_t)\big\|P_t^o(\cdot|s_t,a_t)\big)\Big) - \sum_{t=h+1}^H \Omega_{s_t}^{\eta}(\pi_t)\Big| s_h = s, a_h = a, \pi_{h+1:H}\bigg]\bigg\}$$

$$\leq -\Omega_s^{\eta}(\pi_h) + \sum_{a \in \mathcal{A}} \pi_h(a|s)\mathbb{E}_{\{\widetilde{P}_t\}_{t=h}^H}\bigg[$$

$$\sum_{t=h}^H \Big( r_t(s_t,a_t) + \beta \cdot \mathrm{D}\big(\widetilde{P}_t(\cdot|s_t,a_t)\big\|P_t^o(\cdot|s_t,a_t)\big)\Big) - \sum_{t=h+1}^H \Omega_{s_t}^{\eta}(\pi_t)\Big| s_h = s, a_h = a, \pi_{h+1:H}\bigg] \tag{D.8}$$

$$= -\Omega_s^{\eta}(\pi_h) + \sum_{a \in \mathcal{A}} \pi_h(a|s)Q_h^{\pi,\beta,\eta}(s,a), \tag{D.9}$$

where (D.8) is because the definition of the infimum operator and (D.9) is from (D.7) and (D.2).

On the other hand, we can lower bound $V_h^{\pi,\beta,\eta}(s)$ as

$$V_h^{\pi,\beta,\eta}(s)$$

$$= \inf_{P_t \in \Delta(S)} \mathbb{E}_{\{P_t\}_{t=h}^H}\left[ \sum_{t=h}^H \Big( r_t(s_t,a_t) - \Omega_{s_t}^{\eta}(\pi_t) + \beta \cdot \mathrm{D}\big(P_t(\cdot|s_t,a_t)\big\|P_t^o(\cdot|s_t,a_t)\big)\Big) \Big| s_h = s, \pi_{h:H}\right]$$

$$= \inf_{P_t \in \Delta(S)} \bigg\{ -\Omega_s^{\eta}(\pi_h) + \sum_{a \in \mathcal{A}} \pi_h(a|s)\mathbb{E}_{\{P_t\}_{t=h}^H}\bigg[$$

$$\sum_{t=h}^H \Big( r_t(s_t,a_t) + \beta \cdot \mathrm{D}\big(P_t(\cdot|s_t,a_t)\big\|P_t^o(\cdot|s_t,a_t)\big)\Big) - \sum_{t=h+1}^H \Omega_{s_t}^{\eta}(\pi_t)\Big| s_h = s, a_h = a, \pi_{h+1:H}\bigg]\bigg\}$$

$$\geq -\Omega_s^{\eta}(\pi_h) + \sum_{a \in \mathcal{A}} \pi_h(a|s) \inf_{P_t \in \Delta(S)} \mathbb{E}_{\{P_t\}_{t=h}^H}\bigg[$$

$$\sum_{t=h}^H \Big( r_t(s_t,a_t) + \beta \cdot \mathrm{D}\big(P_t(\cdot|s_t,a_t)\big\|P_t^o(\cdot|s_t,a_t)\big)\Big) - \sum_{t=h+1}^H \Omega_{s_t}^{\eta}(\pi_t)\Big| s_h = s, a_h = a, \pi_{h+1:H}\bigg] \tag{D.10}$$

$$= -\Omega_s^{\eta}(\pi_h) + \sum_{a \in \mathcal{A}} \pi_h(a|s)Q_h^{\pi,\beta,\eta}(s,a), \tag{D.11}$$

where (D.10) is because $\inf \sum f \geq \sum \inf f$ and (D.11) is from the definition of $Q_h^{\pi,\beta,\eta}(s,a)$.

By combining (D.9) and (D.11), we have

$$V_h^{\pi,\beta,\eta}(s) = -\Omega_s^{\eta}(\pi_h) + \sum_{a \in \mathcal{A}} \pi_h(a|s)Q_h^{\pi,\beta,\eta}(s,a),$$

which implies (3.2).  $\qquad \square$

## D.2. Proof of the Double Robustness

*Proof of Remark 3.2.* We have

$$\inf_{\substack{\widetilde{r} \in r + \mathcal{R}^\eta(\pi) \\ P \in \Delta(\mathcal{S})}} \mathbb{E}_{a \sim \pi(\cdot|s)}\big[\widetilde{r}(s,a) + \mathbb{E}_{s' \sim P(\cdot|s,a)}[V(s')] + \beta \cdot \mathrm{D}\big(P(s,a)\|P^o(s,a)\big)\big]$$

$$= \inf_{\substack{\widetilde{r} \in r + \mathcal{R}^\eta(\pi) \\ P \in \Delta(\mathcal{S})}} \Big(\mathbb{E}_{a \sim \pi(\cdot|s)}\big[\widetilde{r}(s,a)\big] + \mathbb{E}_{a \sim \pi(\cdot|s)}\Big[\mathbb{E}_{s' \sim P(\cdot|s,a)}[V(s')] + \beta \cdot \mathrm{D}\big(P(s,a)\|P^o(s,a)\big)\big]\Big]\Big)$$

$$= \inf_{\widetilde{r} \in r + \mathcal{R}^\eta(\pi)} \sum_{a \in \mathcal{A}} \pi(a|s)\widetilde{r}(s,a) + \inf_{P \in \Delta(\mathcal{S})} \mathbb{E}_{a \sim \pi(\cdot|s)}\Big[\mathbb{E}_{s' \sim P(\cdot|s,a)}[V(s')] + \beta \cdot \mathrm{D}\big(P(s,a)\|P^o(s,a)\big)\big]\Big]$$

$$= \inf_{\Delta r \in \mathcal{R}^\eta(\pi)} \sum_{a \in \mathcal{A}} \pi(a|s)\Delta r(s,a) + \inf_{P \in \Delta(\mathcal{S})} \mathbb{E}_{a \sim \pi(\cdot|s)}\Big[r(s,a) + \mathbb{E}_{s' \sim P(\cdot|s,a)}[V(s')] + \beta \cdot \mathrm{D}\big(P(s,a)\|P^o(s,a)\big)\big]\Big]$$

$$= \inf_{\Delta r \in \mathcal{R}^\eta(\pi)} \big\langle \pi(\cdot|s), \Delta r(s,\cdot) \big\rangle + \inf_{P \in \Delta(\mathcal{S})} \mathbb{E}_{a \sim \pi(\cdot|s)}\Big[r(s,a) + \mathbb{E}_{s' \sim P(\cdot|s,a)}[V(s')] + \beta \cdot \mathrm{D}\big(P(s,a)\|P^o(s,a)\big)\big]\Big]$$

$$= -\sup_{\Delta r \in \mathcal{R}^\eta(\pi)} \big\langle -\pi(\cdot|s), \Delta r(s,\cdot) \big\rangle + \inf_{P \in \Delta(\mathcal{S})} \mathbb{E}_{a \sim \pi(\cdot|s)}\Big[r(s,a) + \mathbb{E}_{s' \sim P(\cdot|s,a)}[V(s')] + \beta \cdot \mathrm{D}\big(P(s,a)\|P^o(s,a)\big)\big]\Big]$$

$$= -\sigma_{\mathcal{R}_s^\eta(\pi)}\big(-\pi(\cdot|s)\big) + \inf_{P \in \Delta(\mathcal{S})} \mathbb{E}_{a \sim \pi(\cdot|s)}\Big[r(s,a) + \mathbb{E}_{s' \sim P(\cdot|s,a)}[V(s')] + \beta \cdot \mathrm{D}\big(P(s,a)\|P^o(s,a)\big)\big]\Big] \tag{D.12}$$

$$= \inf_{P \in \Delta(\mathcal{S})} \mathbb{E}_{a \sim \pi(\cdot|s)}\Big[r(s,a) + \mathbb{E}_{s' \sim P(\cdot|s,a)}[V(s')] + \beta \cdot \mathrm{D}\big(P(s,a)\|P^o(s,a)\big)\big]\Big] - \Omega_s^\eta(\pi) \tag{D.13}$$

$$= \inf_{P \in \Delta(\mathcal{S})} \mathbb{E}_{a \sim \pi(\cdot|s)}\Big[r(s,a) + \mathbb{E}_{s' \sim P(\cdot|s,a)}[V(s')] + \beta \cdot \mathrm{D}\big(P(s,a)\|P^o(s,a)\big) - \Omega_s^\eta(\pi)\Big],$$

where (D.12) is from the definition that $\sigma_{\mathcal{Z}}(\boldsymbol{y}) = \max_{\boldsymbol{a} \in \mathcal{Z}}\langle \boldsymbol{a}, \boldsymbol{y} \rangle$, (D.13) is because we choose $\Omega_s^\eta(\pi) = \sigma_{\mathcal{R}_s^\eta(\pi)}(-\pi(\cdot|s))$. This finishes the proof. $\square$

## D.3. Proofs of the Main Results

To establish Theorem 5.4, we first introduce Lemma D.1, which bounds the regret by the sum of bonus terms under the expectation defined by the worst-case transition.

**Lemma D.1.** Under Assumption 5.3, for each $f$-divergence setting and any $\delta \in (0, 1/2)$, with probability at least $1 - 2\delta$, the regret of Algorithm 1 satisfies

$$\mathrm{Regret}(K) \leq 2 \sum_{k=1}^{K} \sum_{h=1}^{H} \mathbb{E}_{P^{w,k}, \pi^k}\big[b_h^k(s,a)\big], \tag{D.14}$$

where $b_h^k(s,a)$ is the bonus term defined in Section 4.2 for each $f$-divergence setting.

To upper bound the sum of the expectations of $\sqrt{\frac{1}{n_h^{k-1} \vee 1}}$ under the worst-case environment, He et al. (2025) proved the following lemma.

**Lemma D.2.** (He et al., 2025, Lemma C.4, Lemma C.9) If the algorithm RSPVI selects $\pi^k$ for the $k$-th episode, with probability at least $1 - \delta$, it holds that

$$\sum_{k=1}^{K} \sum_{h=1}^{H} \mathbb{E}_{P^{w,k}, \pi^k}\left[\sqrt{\frac{1}{n_h^{k-1}(s,a) \vee 1}}\right] = \widetilde{\mathcal{O}}\big(\sqrt{C_{vr}SAH^2K} + C_{vr}SAH\big).$$

*Proof of Theorem 5.4.* The result follows immediately from combining Lemma D.1 and Lemma D.2, and substituting the specific bonus formulation defined in Section 4.2. $\square$

## D.4. Proofs of the Technical Lemmas

Here, we present the proof of Lemma D.1 is various $f$-divergence settings. For convenience, we also write $P^{w,k} := P^{w,\pi^k}$, $\mathrm{d}^k := \mathrm{d}^{\pi^k}$ and $\mathrm{q}^k := \mathrm{q}^{\pi^k}$.

### D.4.1. PROOF OF LEMMA D.1 IN TV SETTING

**Lemma D.3** (Dual formulation). (He et al., 2025, Lemma D.1) For the optimization problem $Q_h(s,a) = r_h(s,a) + \inf_{P \in \Delta(S)} \left( \mathbb{E}_P[V_{h+1}] + \beta \mathrm{TV}(P\|P_h^o) \right)(s,a)$, we have its dual formulation as follows

$$Q_h = r_h - \mathbb{E}_{P_h^o}\left[ \left( \min_{s \in \mathcal{S}} V_{h+1}(s) + \beta - V_{h+1}(s) \right)_+ \right] + \left( \min_{s \in \mathcal{S}} V_{h+1}(s) + \beta \right). \tag{D.15}$$

**Lemma D.4** (Optimism). If we set the bonus term as follows

$$\mathrm{bonus}_h^k(s,a) = 2H\sqrt{\frac{2S\ln(2SAHK/\delta)}{n_h^{k-1}(s,a) \vee 1}}, \tag{D.16}$$

then for any policy $\pi$ and any $(k,h,s,a) \in [K] \times [H] \times \mathcal{S} \times \mathcal{A}$, with probability at least $1 - 2\delta$, we have $Q_h^{k,\beta,\eta}(s,a) \geq Q_h^{\pi,\beta,\eta}(s,a)$. Specially, by setting $\pi = \pi^*$, we have $Q_h^{k,\beta,\eta}(s,a) \geq Q_h^{\pi^*,\beta,\eta}(s,a)$.

*Proof.* We prove this by induction. First, when $h = H + 1$, $Q_{H+1}^{k,\beta,\eta}(s,a) = 0 = Q_{H+1}^{\pi,\beta,\eta}(s,a)$ holds trivially.

Assume $Q_{h+1}^{k,\beta,\eta}(s,a) \geq Q_{h+1}^{\pi,\beta,\eta}(s,a)$ holds, due to the choice of $\pi_{h+1}^k$ in (4.3), we have

$$\begin{aligned}
V_{h+1}^{k,\beta,\eta}(s) &= \langle Q_{h+1}^{k,\beta,\eta}(s,\cdot), \pi_{h+1}^k(\cdot|s) \rangle - \Omega_s^\eta(\pi_{h+1}^k) \\
&\geq \langle Q_{h+1}^{k,\beta,\eta}(s,\cdot), \pi_{h+1}(\cdot|s) \rangle - \Omega_s^\eta(\pi_{h+1}) \\
&\geq \langle Q_{h+1}^{\pi,\beta,\eta}(s,\cdot), \pi_{h+1}(\cdot|s) \rangle - \Omega_s^\eta(\pi_{h+1}) = V_{h+1}^{\pi,\beta,\eta}(s).
\end{aligned}$$

Recall that we denote $Q_h^{k,\beta,\eta}$ as the optimistic estimation in $k$-th episode, that is,

$$Q_h^{k,\beta,\eta}(s,a) = \min\left\{ \mathrm{bonus}_h^k(s,a) + \widehat{r}_h^k(s,a) + \inf_{P \in \Delta(S)} \left( \mathbb{E}_P\left[V_{h+1}^{k,\beta,\eta}\right] + \beta \mathrm{TV}(P\|\widehat{P}_h^k) \right)(s,a), H - h + 1 \right\}.$$

If $Q_h^{k,\beta,\eta}(s,a) = H - h + 1$, then it follows immediately that

$$Q_h^{k,\beta,\eta}(s,a) = H - h + 1 \geq Q_h^{\pi,\beta,\eta}(s,a)$$

by the definition of $Q_h^{\pi,\beta,\eta}(s,a)$. Otherwise, we can infer that

$$\begin{aligned}
&Q_h^{k,\beta,\eta} - Q_h^{\pi,\beta,\eta} \\
&= \mathrm{bonus}_h^k + \widehat{r}_h^k + \inf_{P \in \Delta(S)} \left( \mathbb{E}_P\left[V_{h+1}^{k,\beta,\eta}\right] + \beta \mathrm{TV}(P\|\widehat{P}_h^k) \right) - r_h - \inf_{P \in \Delta(S)} \left( \mathbb{E}_P\left[V_{h+1}^{\pi,\beta,\eta}\right] + \beta \mathrm{TV}(P\|P_h^o) \right) \\
&= \mathrm{bonus}_h^k + \widehat{r}_h^k - r_h + \inf_{P \in \Delta(S)} \left( \mathbb{E}_P\left[V_{h+1}^{k,\beta,\eta}\right] + \beta \mathrm{TV}(P\|\widehat{P}_h^k) \right) - \inf_{P \in \Delta(S)} \left( \mathbb{E}_P\left[V_{h+1}^{k,\beta,\eta}\right] + \beta \mathrm{TV}(P\|P_h^o) \right) \\
&\quad + \inf_{P \in \Delta(S)} \left( \mathbb{E}_P\left[V_{h+1}^{k,\beta,\eta}\right] + \beta \mathrm{TV}(P\|P_h^o) \right) - \inf_{P \in \Delta(S)} \left( \mathbb{E}_P\left[V_{h+1}^{\pi,\beta,\eta}\right] + \beta \mathrm{TV}(P\|P_h^o) \right) \\
&\geq \mathrm{bonus}_h^k + \widehat{r}_h^k - r_h + \inf_{P \in \Delta(S)} \left( \mathbb{E}_P\left[V_{h+1}^{k,\beta,\eta}\right] - \mathbb{E}_P\left[V_{h+1}^{\pi,\beta,\eta}\right] \right) \\
&\quad + \inf_{P \in \Delta(S)} \left( \mathbb{E}_P\left[V_{h+1}^{k,\beta,\eta}\right] + \beta \mathrm{TV}(P\|\widehat{P}_h^k) \right) - \inf_{P \in \Delta(S)} \left( \mathbb{E}_P\left[V_{h+1}^{k,\beta,\eta}\right] + \beta \mathrm{TV}(P\|P_h^o) \right) \tag{D.17} \\
&\geq \mathrm{bonus}_h^k + \widehat{r}_h^k - r_h + \inf_{P \in \Delta(S)} \left( \mathbb{E}_P\left[V_{h+1}^{k,\beta,\eta}\right] + \beta \mathrm{TV}(P\|\widehat{P}_h^k) \right) - \inf_{P \in \Delta(S)} \left( \mathbb{E}_P\left[V_{h+1}^{k,\beta,\eta}\right] + \beta \mathrm{TV}(P\|P_h^o) \right) \tag{D.18}
\end{aligned}$$

$$
\begin{aligned}
&= \text{bonus}_h^k + \widehat{r}_h^k - r_h - \mathbb{E}_{P_h^o}\left[\left(\min_{s\in\mathcal{S}} V_{h+1}^{k,\beta,\eta}(s) + \beta - V_{h+1}^{k,\beta,\eta}(s)\right)_+\right] + \left(\min_{s\in\mathcal{S}} V_{h+1}^{k,\beta,\eta}(s) + \beta\right) \\
&\quad + \mathbb{E}_{\widehat{P}_h^k}\left[\left(\min_{s\in\mathcal{S}} V_{h+1}^{k,\beta,\eta}(s) + \beta - V_{h+1}^{k,\beta,\eta}(s)\right)_+\right] - \left(\min_{s\in\mathcal{S}} V_{h+1}^{k,\beta,\eta}(s) + \beta\right) \quad\quad\text{(D.19)} \\
&= \text{bonus}_h^k + \widehat{r}_h^k - r_h \\
&\quad - \mathbb{E}_{P_h^o}\left[\left(\min_{s\in\mathcal{S}} V_{h+1}^{k,\beta,\eta}(s) + \beta - V_{h+1}^{k,\beta,\eta}(s)\right)_+\right] + \mathbb{E}_{\widehat{P}_h^k}\left[\left(\min_{s\in\mathcal{S}} V_{h+1}^{k,\beta,\eta}(s) + \beta - V_{h+1}^{k,\beta,\eta}(s)\right)_+\right] \\
&\geq \text{bonus}_h^k - \underbrace{|\widehat{r}_h^k - r_h|}_{(i)} \\
&\quad - \underbrace{\left|\mathbb{E}_{P_h^o}\left[\left(\min_{s\in\mathcal{S}} V_{h+1}^{k,\beta,\eta}(s) + \beta - V_{h+1}^{k,\beta,\eta}(s)\right)_+\right] - \mathbb{E}_{\widehat{P}_h^k}\left[\left(\min_{s\in\mathcal{S}} V_{h+1}^{k,\beta,\eta}(s) + \beta - V_{h+1}^{k,\beta,\eta}(s)\right)_+\right]\right|}_{(ii)}, \quad\text{(D.20)}
\end{aligned}
$$

where (D.17) is from $\inf f(x) - \inf g(x) \geq \inf(f-g)(x)$, (D.18) is from the induction assumption, we plug in the dual formulation (D.15) in (D.19).

For term (i) in (D.20), from Lemma F.1 and a union bound, with probability at least $1-\delta$, we have

$$
\left|\widehat{r}_h^k(s,a) - r_h(s,a)\right| \leq \sqrt{\frac{\ln(2SAHK/\delta)}{2n_h^{k-1}(s,a) \vee 1}} \quad\quad\text{(D.21)}
$$

for any $(k,h,s,a) \in [K] \times [H] \times \mathcal{S} \times \mathcal{A}$.

For any fixed $V$, we apply Lemma F.3 and have

$$
\left|\mathbb{E}_{P_h^o}[V] - \mathbb{E}_{\widehat{P}_h^k}[V]\right| \leq \left\|P_h^o - \widehat{P}_h^k\right\|_1 \cdot \left\|V\right\|_\infty \leq H\sqrt{\frac{2S\ln(2/\delta)}{n_h^{k-1} \vee 1}}, \quad\quad\text{(D.22)}
$$

with probability at least $1-\delta$.

For term (ii) in (D.20), by applying (D.22) and a union bound, with probability at least $1-\delta$, we have

$$
\begin{aligned}
&\left|\mathbb{E}_{P_h^o}\left[\left(\min_{s\in\mathcal{S}} V_{h+1}^{k,\beta,\eta}(s) + \beta - V_{h+1}^{k,\beta,\eta}(s)\right)_+\right] - \mathbb{E}_{\widehat{P}_h^k}\left[\left(\min_{s\in\mathcal{S}} V_{h+1}^{k,\beta,\eta}(s) + \beta - V_{h+1}^{k,\beta,\eta}(s)\right)_+\right]\right| \\
&\leq H\sqrt{\frac{2S\ln(2SAHK/\delta)}{n_h^{k-1} \vee 1}} \quad\quad\text{(D.23)}
\end{aligned}
$$

for any $(k,h,s,a) \in [K] \times [H] \times \mathcal{S} \times \mathcal{A}$.

Apply the union bound again and combine (D.20) with (D.21), (D.23), the definition of bonus and induction assumption. With probability at least $1-2\delta$, we have $Q_h^{k,\rho}(s,a) \geq Q_h^{\pi,\rho}(s,a)$ for any $(k,h,s,a) \in [K] \times [H] \times \mathcal{S} \times \mathcal{A}$. This completes the proof. $\qquad\square$

**Lemma D.5.** If we set the bonus term to be the same as in Lemma D.4, then for any $(k,s,a) \in [K] \times \mathcal{S} \times \mathcal{A}$, with probability at least $1-\delta$, the sum of estimation errors can be bounded as

$$
Q_1^{k,\beta,\eta}(s,a) - Q_1^{\pi^k,\beta,\eta}(s,a) \leq 2 \cdot \mathbb{E}_{P^{w,k},\pi^k}\left[\text{bonus}_h^k\right],
$$

*Proof.* From the proof of Lemma D.4, we see that with probability at least $1-\delta$, for any $(k,s,a) \in [K] \times \mathcal{S} \times \mathcal{A}$,

$$
\begin{aligned}
&\left|\widehat{r}_h^k(s,a) - r_h(s,a)\right| + \Big|\inf_{P\in\Delta(S)}\left(\mathbb{E}_P\left[V_{h+1}^{k,\beta,\eta}\right] + \beta\text{TV}(P\|\widehat{P}_h^k)\right)(s,a) \\
&\quad - \inf_{P\in\Delta(S)}\left(\mathbb{E}_P\left[V_{h+1}^{k,\beta,\eta}\right] + \beta\text{TV}(P\|P_h^o)\right)(s,a)\Big| \leq \text{bonus}_h^k(s,a). \quad\quad\text{(D.24)}
\end{aligned}
$$

Recall that we define $P_h^{w,k} = \underset{P \in \Delta(S)}{\operatorname{argmin}} \left( \mathbb{E}_P\left[V_{h+1}^{\pi^k,\beta,\eta}\right] + \beta \mathrm{TV}(P\|P_h^o) \right)$ as the worst-case transition in Definition 5.1, we have

$$Q_h^{k,\beta,\eta} - Q_h^{\pi^k,\beta,\eta}$$

$$\leq \mathrm{bonus}_h^k + \widehat{r}_h^k + \inf_{P \in \Delta(S)} \left( \mathbb{E}_P\left[V_{h+1}^{k,\beta,\eta}\right] + \beta \mathrm{TV}(P\|\widehat{P}_h^k) \right) - r_h - \inf_{P \in \Delta(S)} \left( \mathbb{E}_P\left[V_{h+1}^{\pi^k,\beta,\eta}\right] + \beta \mathrm{TV}(P\|P_h^o) \right)$$

$$\leq 2\mathrm{bonus}_h^k + \inf_{P \in \Delta(S)} \left( \mathbb{E}_P\left[V_{h+1}^{k,\beta,\eta}\right] + \beta \mathrm{TV}(P\|P_h^o) \right) - \inf_{P \in \Delta(S)} \left( \mathbb{E}_P\left[V_{h+1}^{\pi^k,\beta,\eta}\right] + \beta \mathrm{TV}(P\|P_h^o) \right) \tag{D.25}$$

$$= 2\mathrm{bonus}_h^k + \mathbb{E}_{P_h^{w,k},\pi_h^k}\left[Q_{h+1}^{k,\beta,\eta} - Q_{h+1}^{\pi^k,\beta,\eta}\right]. \tag{D.26}$$

where (D.25) uses (D.24). Apply (D.26) recursively, we can obtain the result. $\qquad\square$

*Proof of Lemma D.1 in TV Setting.* The result directly follows from combining Lemma D.4 and Lemma D.5, along with applying a union bound. $\qquad\square$

### D.4.2. PROOF OF LEMMA D.1 IN KL SETTING

**Lemma D.6** (Dual formulation). (He et al., 2025, Lemma D.5) For the optimization problem $Q_h(s,a) = r_h(s,a) + \inf_{P \in \Delta(S)} \left( \mathbb{E}_P[V_{h+1}] + \beta \mathrm{KL}(P\|P_h^o) \right)(s,a)$, we have its dual formulation as follows

$$Q_h = r_h - \beta \ln \mathbb{E}_{P_h^o}\left[e^{-\beta^{-1}V_{h+1}}\right]. \tag{D.27}$$

**Lemma D.7** (Optimism). If we set the bonus term as follows

$$\mathrm{bonus}_h^k(s,a) = \left(1 + \beta e^{\beta^{-1}H}\sqrt{S}\right)\sqrt{\frac{2\ln(2SAHK/\delta)}{n_h^{k-1}(s,a) \vee 1}}, \tag{D.28}$$

then for any policy $\pi$ and any $(k,h,s,a) \in [K] \times [H] \times \mathcal{S} \times \mathcal{A}$, with probability at least $1 - 2\delta$, we have $Q_h^{k,\beta,\eta}(s,a) \geq Q_h^{\pi,\beta,\eta}(s,a)$. Specially, by setting $\pi = \pi^*$, we have $Q_h^{k,\beta,\eta}(s,a) \geq Q_h^{\pi^*,\beta,\eta}(s,a)$.

*Proof.* We prove this by induction. First, when $h = H+1$, $Q_{H+1}^{k,\beta,\eta}(s,a) = 0 = Q_{H+1}^{\pi,\beta,\eta}(s,a)$ holds trivially.

Assume $Q_{h+1}^{k,\beta,\eta}(s,a) \geq Q_{h+1}^{\pi,\beta,\eta}(s,a)$ holds, due to the choice of $\pi_{h+1}^k$ in (4.3), we have

$$\begin{aligned} V_{h+1}^{k,\beta,\eta}(s) &= \left\langle Q_{h+1}^{k,\beta,\eta}(s,\cdot), \pi_{h+1}^k(\cdot|s) \right\rangle - \Omega_s^\eta(\pi_{h+1}^k) \\ &\geq \left\langle Q_{h+1}^{k,\beta,\eta}(s,\cdot), \pi_{h+1}(\cdot|s) \right\rangle - \Omega_s^\eta(\pi_{h+1}) \\ &\geq \left\langle Q_{h+1}^{\pi,\beta,\eta}(s,\cdot), \pi_{h+1}(\cdot|s) \right\rangle - \Omega_s^\eta(\pi_{h+1}) = V_{h+1}^{\pi,\beta,\eta}(s). \end{aligned}$$

Recall that we denote $Q_h^{k,\beta,\eta}$ as the optimistic estimation in $k$-th episode, that is,

$$Q_h^{k,\beta,\eta}(s,a) = \min\left\{ \mathrm{bonus}_h^k(s,a) + \widehat{r}_h^k(s,a) + \inf_{P \in \Delta(S)} \left( \mathbb{E}_P\left[V_{h+1}^{k,\beta,\eta}\right] + \beta \mathrm{KL}(P\|\widehat{P}_h^k) \right)(s,a), H - h + 1 \right\}.$$

If $Q_h^{k,\beta,\eta}(s,a) = H - h + 1$, then it follows immediately that

$$Q_h^{k,\beta,\eta}(s,a) = H - h + 1 \geq Q_h^{\pi,\beta,\eta}(s,a)$$

by the definition of $Q_h^{\pi,\beta,\eta}(s,a)$. Otherwise, we can infer that

$$Q_h^{k,\beta,\eta} - Q_h^{\pi,\beta,\eta}$$

$$= \text{bonus}_h^k + \widehat{r}_h^k + \inf_{P \in \Delta(S)} \left( \mathbb{E}_P\left[V_{h+1}^{k,\beta,\eta}\right] + \beta \text{KL}(P\|\widehat{P}_h^k) \right) - r_h - \inf_{P \in \Delta(S)} \left( \mathbb{E}_P\left[V_{h+1}^{\pi,\beta,\eta}\right] + \beta \text{KL}(P\|P_h^o) \right)$$

$$= \text{bonus}_h^k + \widehat{r}_h^k - r_h + \inf_{P \in \Delta(S)} \left( \mathbb{E}_P\left[V_{h+1}^{k,\beta,\eta}\right] + \beta \text{KL}(P\|\widehat{P}_h^k) \right) - \inf_{P \in \Delta(S)} \left( \mathbb{E}_P\left[V_{h+1}^{k,\beta,\eta}\right] + \beta \text{KL}(P\|P_h^o) \right)$$

$$+ \inf_{P \in \Delta(S)} \left( \mathbb{E}_P\left[V_{h+1}^{k,\beta,\eta}\right] + \beta \text{KL}(P\|P_h^o) \right) - \inf_{P \in \Delta(S)} \left( \mathbb{E}_P\left[V_{h+1}^{\pi,\beta,\eta}\right] + \beta \text{KL}(P\|P_h^o) \right)$$

$$\geq \text{bonus}_h^k + \widehat{r}_h^k - r_h + \inf_{P \in \Delta(S)} \left( \mathbb{E}_P\left[V_{h+1}^{k,\beta,\eta}\right] - \mathbb{E}_P\left[V_{h+1}^{\pi,\beta,\eta}\right] \right)$$

$$+ \inf_{P \in \Delta(S)} \left( \mathbb{E}_P\left[V_{h+1}^{k,\beta,\eta}\right] + \beta \text{KL}(P\|\widehat{P}_h^k) \right) - \inf_{P \in \Delta(S)} \left( \mathbb{E}_P\left[V_{h+1}^{k,\beta,\eta}\right] + \beta \text{KL}(P\|P_h^o) \right) \tag{D.29}$$

$$\geq \text{bonus}_h^k + \widehat{r}_h^k - r_h + \inf_{P \in \Delta(S)} \left( \mathbb{E}_P\left[V_{h+1}^{k,\beta,\eta}\right] + \beta \text{KL}(P\|\widehat{P}_h^k) \right) - \inf_{P \in \Delta(S)} \left( \mathbb{E}_P\left[V_{h+1}^{k,\beta,\eta}\right] + \beta \text{KL}(P\|P_h^o) \right) \tag{D.30}$$

$$= \text{bonus}_h^k + \widehat{r}_h^k - r_h + \beta \ln \mathbb{E}_{P_h^o}\left[e^{-\beta^{-1} V_{h+1}^{k,\beta,\eta}}\right] - \beta \ln \mathbb{E}_{\widehat{P}_h^k}\left[e^{-\beta^{-1} V_{h+1}^{k,\beta,\eta}}\right] \tag{D.31}$$

$$\geq \text{bonus}_h^k - \underbrace{\left|\widehat{r}_h^k - r_h\right|}_{(i)} - \underbrace{\beta\left|\ln \mathbb{E}_{\widehat{P}_h^k}\left[e^{-\beta^{-1} V_{h+1}^{k,\beta,\eta}}\right] - \ln \mathbb{E}_{P_h^o}\left[e^{-\beta^{-1} V_{h+1}^{k,\beta,\eta}}\right]\right|}_{(ii)}, \tag{D.32}$$

where (D.29) is from $\inf f(x) - \inf g(x) \geq \inf(f - g)(x)$, (D.30) is from the induction assumption, we plug in the dual formulation (D.27) in (D.31).

For term (i) in (D.32), from Lemma F.1 and a union bound, with probability at least $1 - \delta$, we have

$$\left|\widehat{r}_h^k(s,a) - r_h(s,a)\right| \leq \sqrt{\frac{\ln(2SAHK/\delta)}{2n_h^{k-1}(s,a) \vee 1}} \tag{D.33}$$

for any $(k,h,s,a) \in [K] \times [H] \times \mathcal{S} \times \mathcal{A}$.

For term (ii) in (D.32), by applying (D.22) and a union bound, with probability at least $1 - \delta$, we have

$$\left|\ln \mathbb{E}_{\widehat{P}_h^k}\left[e^{-\beta^{-1} V_{h+1}^{k,\beta,\eta}}\right] - \ln \mathbb{E}_{P_h^o}\left[e^{-\beta^{-1} V_{h+1}^{k,\beta,\eta}}\right]\right| \leq e^{\beta^{-1} H} \sqrt{\frac{2S \ln(2SAHK/\delta)}{n_h^{k-1} \vee 1}} \tag{D.34}$$

for any $(k,h,s,a) \in [K] \times [H] \times \mathcal{S} \times \mathcal{A}$.

Apply the union bound again and combine (D.32) with (D.33), (D.34), the definition of bonus and induction assumption. With probability at least $1 - 2\delta$, we have $Q_h^{k,\rho}(s,a) \geq Q_h^{\pi,\rho}(s,a)$ for any $(k,h,s,a) \in [K] \times [H] \times \mathcal{S} \times \mathcal{A}$. This completes the proof. $\square$

**Lemma D.8.** If we set the bonus term to be the same as in Lemma D.7, then for any $(k,s,a) \in [K] \times \mathcal{S} \times \mathcal{A}$, with probability at least $1 - \delta$, the sum of estimation errors can be bounded as

$$Q_1^{k,\beta,\eta}(s,a) - Q_1^{\pi^k,\beta,\eta}(s,a) \leq 2 \cdot \mathbb{E}_{P^{w,k},\pi^k}\left[\text{bonus}_h^k\right],$$

*Proof.* From the proof of Lemma D.7, we see that with probability at least $1 - \delta$, for any $(k,s,a) \in [K] \times \mathcal{S} \times \mathcal{A}$,

$$\left|\widehat{r}_h^k(s,a) - r_h(s,a)\right| + \left|\inf_{P \in \Delta(S)} \left(\mathbb{E}_P\left[V_{h+1}^{k,\beta,\eta}\right] + \beta \text{KL}(P\|\widehat{P}_h^k)\right)(s,a)\right.$$

$$\left. - \inf_{P \in \Delta(S)} \left(\mathbb{E}_P\left[V_{h+1}^{k,\beta,\eta}\right] + \beta \text{KL}(P\|P_h^o)\right)(s,a)\right| \leq \text{bonus}_h^k(s,a). \tag{D.35}$$

Recall that we define $P_h^{w,k} = \underset{P \in \Delta(S)}{\text{argmin}} \left(\mathbb{E}_P\left[V_{h+1}^{\pi^k,\beta,\eta}\right] + \beta \text{KL}(P\|P_h^o)\right)$ as the worst-case transition in Definition 5.1, we have

$$Q_h^{k,\beta,\eta} - Q_h^{\pi^k,\beta,\eta}$$

$$\leq \text{bonus}_h^k + \widehat{r}_h^k + \inf_{P \in \Delta(S)} \left( \mathbb{E}_P\left[V_{h+1}^{k,\beta,\eta}\right] + \beta \text{KL}(P\|\widehat{P}_h^k) \right) - r_h - \inf_{P \in \Delta(S)} \left( \mathbb{E}_P\left[V_{h+1}^{\pi^k,\beta,\eta}\right] + \beta \text{KL}(P\|P_h^o) \right)$$

$$\leq 2\text{bonus}_h^k + \inf_{P \in \Delta(S)} \left( \mathbb{E}_P\left[V_{h+1}^{k,\beta,\eta}\right] + \beta \text{KL}(P\|P_h^o) \right) - \inf_{P \in \Delta(S)} \left( \mathbb{E}_P\left[V_{h+1}^{\pi^k,\beta,\eta}\right] + \beta \text{KL}(P\|P_h^o) \right) \tag{D.36}$$

$$= 2\text{bonus}_h^k + \mathbb{E}_{P_h^{w,k}, \pi_h^k}\left[Q_{h+1}^{k,\beta,\eta} - Q_{h+1}^{\pi^k,\beta,\eta}\right]. \tag{D.37}$$

where (D.36) uses (D.35). Apply (D.37) recursively, we can obtain the result. $\qquad \square$

*Proof of Lemma D.1 in KL Setting.* The result directly follows from combining Lemma D.7 and Lemma D.8, along with applying a union bound. $\qquad \square$

### D.4.3. PROOF OF LEMMA D.1 IN $\chi^2$ SETTING

**Lemma D.9** (Dual formulation). (He et al., 2025, Lemma D.9) For the optimization problem $Q_h(s,a) = r_h(s,a) + \inf_{P \in \Delta(S)} \left( \mathbb{E}_P[V_{h+1}] + \beta \chi^2(P\|P_h^o) \right)(s,a)$, we have its dual formulation as follows

$$Q_h = r_h + \sup_{\boldsymbol{\lambda} \in [0,H]} \left( \mathbb{E}_{P_h^o}\left[V_{h+1} - \boldsymbol{\lambda}\right] - \frac{1}{4\beta} \text{Var}_{P_h^o}(V_{h+1} - \boldsymbol{\lambda}) \right). \tag{D.38}$$

**Lemma D.10** (Optimism). If we set the bonus term as follows

$$\text{bonus}_h^k(s,a) = \left(2 + \frac{3H}{4\beta}\right) H \sqrt{\frac{2S^2 \ln(48SAH^3K^2/\delta)}{n_h^{k-1}(s,a) \vee 1}} + \left(1 + \frac{1}{4\beta}\right) \frac{1}{K}, \tag{D.39}$$

then for any policy $\pi$ and any $(k,h,s,a) \in [K] \times [H] \times \mathcal{S} \times \mathcal{A}$, with probability at least $1 - 3\delta$, we have $Q_h^{k,\beta,\eta}(s,a) \geq Q_h^{\pi,\beta,\eta}(s,a)$. Specially, by setting $\pi = \pi^*$, we have $Q_h^{k,\beta,\eta}(s,a) \geq Q_h^{\pi^*,\beta,\eta}(s,a)$.

*Proof.* We prove this by induction. First, when $h = H+1$, $Q_{H+1}^{k,\beta,\eta}(s,a) = 0 = Q_{H+1}^{\pi,\beta,\eta}(s,a)$ holds trivially.

Assume $Q_{h+1}^{k,\beta,\eta}(s,a) \geq Q_{h+1}^{\pi,\beta,\eta}(s,a)$ holds, due to the choice of $\pi_{h+1}^k$ in (4.3), we have

$$\begin{aligned} V_{h+1}^{k,\beta,\eta}(s) &= \left\langle Q_{h+1}^{k,\beta,\eta}(s,\cdot), \pi_{h+1}^k(\cdot|s) \right\rangle - \Omega_s^\eta(\pi_{h+1}^k) \\ &\geq \left\langle Q_{h+1}^{k,\beta,\eta}(s,\cdot), \pi_{h+1}(\cdot|s) \right\rangle - \Omega_s^\eta(\pi_{h+1}) \\ &\geq \left\langle Q_{h+1}^{\pi,\beta,\eta}(s,\cdot), \pi_{h+1}(\cdot|s) \right\rangle - \Omega_s^\eta(\pi_{h+1}) = V_{h+1}^{\pi,\beta,\eta}(s). \end{aligned}$$

Recall that we denote $Q_h^{k,\beta,\eta}$ as the optimistic estimation in $k$-th episode, that is,

$$Q_h^{k,\beta,\eta}(s,a) = \min \left\{ \text{bonus}_h^k(s,a) + \widehat{r}_h^k(s,a) + \inf_{P \in \Delta(S)} \left( \mathbb{E}_P\left[V_{h+1}^{k,\beta,\eta}\right] + \beta \chi^2(P\|\widehat{P}_h^k) \right)(s,a), H-h+1 \right\}.$$

If $Q_h^{k,\beta,\eta}(s,a) = H-h+1$, then it follows immediately that

$$Q_h^{k,\beta,\eta}(s,a) = H-h+1 \geq Q_h^{\pi,\beta,\eta}(s,a)$$

by the definition of $Q_h^{\pi,\beta,\eta}(s,a)$. Otherwise, we can infer that

$$\begin{aligned} &Q_h^{k,\beta,\eta} - Q_h^{\pi,\beta,\eta} \\ &= \text{bonus}_h^k + \widehat{r}_h^k + \inf_{P \in \Delta(S)} \left( \mathbb{E}_P\left[V_{h+1}^{k,\beta,\eta}\right] + \beta \chi^2(P\|\widehat{P}_h^k) \right) - r_h - \inf_{P \in \Delta(S)} \left( \mathbb{E}_P\left[V_{h+1}^{\pi,\beta,\eta}\right] + \beta \chi^2(P\|P_h^o) \right) \\ &= \text{bonus}_h^k + \widehat{r}_h^k - r_h + \inf_{P \in \Delta(S)} \left( \mathbb{E}_P\left[V_{h+1}^{k,\beta,\eta}\right] + \beta \chi^2(P\|\widehat{P}_h^k) \right) - \inf_{P \in \Delta(S)} \left( \mathbb{E}_P\left[V_{h+1}^{k,\beta,\eta}\right] + \beta \chi^2(P\|P_h^o) \right) \end{aligned}$$

$$+ \inf_{P\in\Delta(S)} \left(\mathbb{E}_P\left[V_{h+1}^{k,\beta,\eta}\right] + \beta\chi^2(P\|P_h^o)\right) - \inf_{P\in\Delta(S)} \left(\mathbb{E}_P\left[V_{h+1}^{\pi,\beta,\eta}\right] + \beta\chi^2(P\|P_h^o)\right)$$

$$\geq \mathrm{bonus}_h^k + \widehat{r}_h^k - r_h + \inf_{P\in\Delta(S)} \left(\mathbb{E}_P\left[V_{h+1}^{k,\beta,\eta}\right] - \mathbb{E}_P\left[V_{h+1}^{\pi,\beta,\eta}\right]\right)$$

$$+ \inf_{P\in\Delta(S)} \left(\mathbb{E}_P\left[V_{h+1}^{k,\beta,\eta}\right] + \beta\chi^2(P\|\widehat{P}_h^k)\right) - \inf_{P\in\Delta(S)} \left(\mathbb{E}_P\left[V_{h+1}^{k,\beta,\eta}\right] + \beta\chi^2(P\|P_h^o)\right) \tag{D.40}$$

$$\geq \mathrm{bonus}_h^k + \widehat{r}_h^k - r_h + \inf_{P\in\Delta(S)} \left(\mathbb{E}_P\left[V_{h+1}^{k,\beta,\eta}\right] + \beta\chi^2(P\|\widehat{P}_h^k)\right) - \inf_{P\in\Delta(S)} \left(\mathbb{E}_P\left[V_{h+1}^{k,\beta,\eta}\right] + \beta\chi^2(P\|P_h^o)\right) \tag{D.41}$$

$$= \mathrm{bonus}_h^k + \widehat{r}_h^k - r_h + \sup_{\boldsymbol{\lambda}\in[0,H]} \left(\mathbb{E}_{\widehat{P}_h^k}\left[V_{h+1}^{k,\beta,\eta} - \boldsymbol{\lambda}\right] - \frac{1}{4\beta}\mathrm{Var}_{\widehat{P}_h^k}\left(V_{h+1}^{k,\beta,\eta} - \boldsymbol{\lambda}\right)\right)$$

$$- \sup_{\boldsymbol{\lambda}\in[0,H]} \left(\mathbb{E}_{P_h^o}\left[V_{h+1}^{k,\beta,\eta} - \boldsymbol{\lambda}\right] - \frac{1}{4\beta}\mathrm{Var}_{P_h^o}\left(V_{h+1}^{k,\beta,\eta} - \boldsymbol{\lambda}\right)\right) \tag{D.42}$$

$$\geq \mathrm{bonus}_h^k + \widehat{r}_h^k - r_h + \inf_{\boldsymbol{\lambda}\in[0,H]} \left\{\left(\mathbb{E}_{\widehat{P}_h^k}\left[V_{h+1}^{k,\beta,\eta} - \boldsymbol{\lambda}\right] - \frac{1}{4\beta}\mathrm{Var}_{\widehat{P}_h^k}\left(V_{h+1}^{k,\beta,\eta} - \boldsymbol{\lambda}\right)\right)\right.$$

$$\left.- \left(\mathbb{E}_{P_h^o}\left[V_{h+1}^{k,\beta,\eta} - \boldsymbol{\lambda}\right] - \frac{1}{4\beta}\mathrm{Var}_{P_h^o}\left(V_{h+1}^{k,\beta,\eta} - \boldsymbol{\lambda}\right)\right)\right\}$$

$$\geq \mathrm{bonus}_h^k - \left|\widehat{r}_h^k - r_h\right| - \sup_{\boldsymbol{\lambda}\in[0,H]} \left|\left(\mathbb{E}_{\widehat{P}_h^k}\left[V_{h+1}^{k,\beta,\eta} - \boldsymbol{\lambda}\right] - \frac{1}{4\beta}\mathrm{Var}_{\widehat{P}_h^k}\left(V_{h+1}^{k,\beta,\eta} - \boldsymbol{\lambda}\right)\right.\right.$$

$$\left.\left.- \left(\mathbb{E}_{P_h^o}\left[V_{h+1}^{k,\beta,\eta} - \boldsymbol{\lambda}\right] - \frac{1}{4\beta}\mathrm{Var}_{P_h^o}\left(V_{h+1}^{k,\beta,\eta} - \boldsymbol{\lambda}\right)\right)\right|\right.$$

$$\geq \mathrm{bonus}_h^k - \underbrace{\left|\widehat{r}_h^k - r_h\right|}_{(i)} - \underbrace{\sup_{\boldsymbol{\lambda}\in[0,H]} \left|\mathbb{E}_{\widehat{P}_h^k}\left[V_{h+1}^{k,\beta,\eta} - \boldsymbol{\lambda}\right] - \mathbb{E}_{P_h^o}\left[V_{h+1}^{k,\beta,\eta} - \boldsymbol{\lambda}\right]\right|}_{(ii)}$$

$$- \underbrace{\frac{1}{4\beta}\sup_{\boldsymbol{\lambda}\in[0,H]} \left|\mathrm{Var}_{\widehat{P}_h^k}\left(V_{h+1}^{k,\beta,\eta} - \boldsymbol{\lambda}\right) - \mathrm{Var}_{P_h^o}\left(V_{h+1}^{k,\beta,\eta} - \boldsymbol{\lambda}\right)\right|}_{(iii)}, \tag{D.43}$$

where (D.40) is from $\inf f(x) - \inf g(x) \geq \inf(f - g)(x)$, (D.41) is from the induction assumption, we plug in the dual formulation (D.38) in (D.42).

For term (i) in (D.43), from Lemma F.1 and a union bound, with probability at least $1 - \delta$, we have

$$\left|\widehat{r}_h^k(s,a) - r_h(s,a)\right| \leq \sqrt{\frac{\ln(2SAHK/\delta)}{2n_h^{k-1}(s,a) \vee 1}}, \tag{D.44}$$

for any $(k,h,s,a) \in [K] \times [H] \times \mathcal{S} \times \mathcal{A}$.

We denote $V(\boldsymbol{\lambda}) = V_{h+1}^{k,\rho} - \boldsymbol{\lambda} \in [-H, H]$ and $\mathcal{V} = \left\{V \in \mathbb{R}^S : \|V\|_\infty \leq H\right\}$. To bound term (ii) in (D.43), we create a $\epsilon$-net $\mathcal{N}_\mathcal{V}(\epsilon)$ for $\mathcal{V}$. From Lemma F.4, it holds that $\ln|\mathcal{N}_\mathcal{V}(\epsilon)| \leq |S| \cdot \ln(3H/\epsilon)$.

Therefore, by the definition of $\mathcal{N}_\mathcal{V}(\epsilon)$, for any fixed $V$, there exists a $V' \in \mathcal{N}_\mathcal{V}(\epsilon)$ such that $\|V - V'\|_\infty \leq \epsilon$, that is

$$\left|\mathbb{E}_{P_h^o}[V] - \mathbb{E}_{\widehat{P}_h^k}[V]\right| \leq \left|\mathbb{E}_{P_h^o}[V] - \mathbb{E}_{P_h^o}[V']\right| + \left|\mathbb{E}_{P_h^o}[V'] - \mathbb{E}_{\widehat{P}_h^k}[V']\right| + \left|\mathbb{E}_{\widehat{P}_h^k}[V'] - \mathbb{E}_{\widehat{P}_h^k}[V]\right|$$

$$\leq \|P_h^o\|_1\|V - V'\|_\infty + \left|\mathbb{E}_{P_h^o}[V'] - \mathbb{E}_{\widehat{P}_h^k}[V']\right| + \left\|\widehat{P}_h^k\right\|_1\|V - V'\|_\infty$$

$$\leq \sup_{V'\in\mathcal{N}_\mathcal{V}(\epsilon)} \left|\mathbb{E}_{P_h^o}[V'] - \mathbb{E}_{\widehat{P}_h^k}[V']\right| + 2\epsilon, \tag{D.45}$$

where the second inequality follows from the Holder's inequality.

Then with probability at least $1 - \delta$, we have

$$\sup_{\boldsymbol{\lambda}\in[0,H]} \left|\mathbb{E}_{\widehat{P}_h^k}\left[V_{h+1}^{k,\rho} - \boldsymbol{\lambda}\right] - \mathbb{E}_{P_h^o}\left[V_{h+1}^{k,\rho} - \boldsymbol{\lambda}\right]\right| \leq \sup_{\eta\in[0,H]} \left|\mathbb{E}_{P_h^o}\left[V(\boldsymbol{\lambda})\right] - \mathbb{E}_{\widehat{P}_h^k}\left[V(\boldsymbol{\lambda})\right]\right|$$

$$\leq \sup_{V \in \mathcal{N}_{\mathcal{V}}(\epsilon)} \left| \mathbb{E}_{P_h^o}[V] - \mathbb{E}_{\widehat{P}_h^k}[V] \right| + 2\epsilon \tag{D.46}$$

$$\leq H \sqrt{\frac{2S \ln(2SAHK|\mathcal{N}_{\mathcal{V}}(\epsilon)|/\delta)}{n_h^{k-1} \vee 1}} + 2\epsilon \tag{D.47}$$

$$\leq H \sqrt{\frac{2S^2 \ln(6SAH^2K/\epsilon\delta)}{n_h^{k-1} \vee 1}} + 2\epsilon$$

$$= H \sqrt{\frac{2S^2 \ln(12SAH^2K^2/\delta)}{n_h^{k-1} \vee 1}} + \frac{1}{K}, \tag{D.48}$$

for any $(k, h, s, a) \in [K] \times [H] \times \mathcal{S} \times \mathcal{A}$, where (D.46) follows from (D.45), (D.47) is from (D.22) and a union bound, we set $\epsilon = 1/2K$ in (D.48).

To bound term (iii) in (D.43), we again invoke the definition of $\mathcal{N}_{\mathcal{V}}(\epsilon)$. For any fixed $V$, there exists a $V' \in \mathcal{N}_{\mathcal{V}}(\epsilon)$ such that $\|V - V'\|_\infty \leq \epsilon$, that is

$$\left| \mathrm{Var}_{P_h^o}(V) - \mathrm{Var}_{\widehat{P}_h^k}(V) \right|$$

$$\leq \left| \mathrm{Var}_{P_h^o}(V) - \mathrm{Var}_{P_h^o}(V') \right| + \left| \mathrm{Var}_{P_h^o}(V') - \mathrm{Var}_{\widehat{P}_h^k}(V') \right| + \left| \mathrm{Var}_{\widehat{P}_h^k}(V') - \mathrm{Var}_{\widehat{P}_h^k}(V) \right|$$

$$\leq \left| \mathbb{E}_{P_h^o}\left[V^2 - V'^2\right] \right| + \left| \mathbb{E}_{P_h^o}^2[V] - \mathbb{E}_{P_h^o}^2[V'] \right| + \left| \mathbb{E}_{\widehat{P}_h^k}\left[V'^2 - V^2\right] \right| + \left| \mathbb{E}_{\widehat{P}_h^k}^2[V'] - \mathbb{E}_{\widehat{P}_h^k}^2[V] \right|$$

$$+ \left| \mathrm{Var}_{P_h^o}(V') - \mathrm{Var}_{\widehat{P}_h^k}(V') \right|$$

$$\leq \left| \mathrm{Var}_{P_h^o}(V') - \mathrm{Var}_{\widehat{P}_h^k}(V') \right| + 8H\epsilon$$

$$\leq \sup_{V' \in N_{\mathcal{V}}} \left| \mathrm{Var}_{P_h^o}(V') - \mathrm{Var}_{\widehat{P}_h^k}(V') \right| + 8H\epsilon. \tag{D.49}$$

For any fixed $V$, following the same analysis as (D.22), we have

$$\left| \mathrm{Var}_{P_h^o}(V) - \mathrm{Var}_{\widehat{P}_h^k}(V) \right| = \left| \left(\mathbb{E}_{P_h^o}[V^2] - \mathbb{E}_{P_h^o}^2[V]\right) - \left(\mathbb{E}_{\widehat{P}_h^k}[V^2] - \mathbb{E}_{\widehat{P}_h^k}^2[V]\right) \right|$$

$$\leq \left| \mathbb{E}_{P_h^o}[V^2] - \mathbb{E}_{\widehat{P}_h^k}[V^2] \right| + \left| \mathbb{E}_{P_h^o}^2[V] - \mathbb{E}_{\widehat{P}_h^k}^2[V] \right|$$

$$\leq H^2 \sqrt{\frac{2S \ln(2/\delta)}{n_h^{k-1} \vee 1}} + \left(\mathbb{E}_{P_h^o}[V] + \mathbb{E}_{\widehat{P}_h^k}[V]\right) \cdot \left| \mathbb{E}_{P_h^o}[V] - \mathbb{E}_{\widehat{P}_h^k}[V] \right|$$

$$\leq H^2 \sqrt{\frac{2S \ln(2/\delta)}{n_h^{k-1} \vee 1}} + 2H^2 \sqrt{\frac{2S \ln(2/\delta)}{n_h^{k-1} \vee 1}}$$

$$\leq 3H^2 \sqrt{\frac{2S \ln(2/\delta)}{n_h^{k-1} \vee 1}} \tag{D.50}$$

with probability at least $1 - \delta$.

Then with probability at least $1 - \delta$, we have

$$\sup_{\boldsymbol{\lambda} \in [0,H]} \left| \mathrm{Var}_{\widehat{P}_h^k}\left(V_{h+1}^{k,\beta,\eta} - \boldsymbol{\lambda}\right) - \mathrm{Var}_{P_h^o}\left(V_{h+1}^{k,\beta,\eta} - \boldsymbol{\lambda}\right) \right| \leq \sup_{\boldsymbol{\lambda} \in [0,H]} \left| \mathrm{Var}_{P_h^o}(V(\boldsymbol{\lambda})) - \mathrm{Var}_{\widehat{P}_h^k}(V(\boldsymbol{\lambda})) \right|$$

$$\leq \sup_{V \in \mathcal{N}_{\mathcal{V}}(\epsilon)} \left| \mathrm{Var}_{P_h^o}(V) - \mathrm{Var}_{\widehat{P}_h^k}(V) \right| + 8H\epsilon \tag{D.51}$$

$$\leq 3H^2 \sqrt{\frac{2S \ln(2SAHK|\mathcal{N}_{\mathcal{V}}|/\delta)}{n_h^{k-1} \vee 1}} + 8H\epsilon \tag{D.52}$$

$$\leq 3H^2 \sqrt{\frac{2S^2 \ln(6SAH^2K/\epsilon\delta)}{n_h^{k-1} \vee 1}} + 8H\epsilon$$

$$= 3H^2\sqrt{\frac{2S^2\ln(48SAH^3K^2/\delta)}{n_h^{k-1}\vee 1}} + \frac{1}{K}, \tag{D.53}$$

for any $(k, h, s, a) \in [K] \times [H] \times \mathcal{S} \times \mathcal{A}$, where (D.51) follows from (D.49), (D.52) is from (D.50) and a union bound, we set $\epsilon = \frac{1}{8HK}$ in (D.53).

Apply the union bound again and combine (D.43) with (D.44), (D.48), (D.53), the definition of bonus and induction assumption. With probability at least $1 - 3\delta$, we have $Q_h^{k,\rho}(s,a) \geq Q_h^{\pi,\rho}(s,a)$ for any $(k, h, s, a) \in [K] \times [H] \times \mathcal{S} \times \mathcal{A}$. This completes the proof. $\qquad\square$

**Lemma D.11.** If we set the bonus term to be the same as in Lemma D.10, then for any $(k, s, a) \in [K] \times \mathcal{S} \times \mathcal{A}$, with probability at least $1 - \delta$, the sum of estimation errors can be bounded as

$$Q_1^{k,\beta,\eta}(s,a) - Q_1^{\pi^k,\beta,\eta}(s,a) \leq 2 \cdot \mathbb{E}_{P^{w,k},\pi^k}\left[\text{bonus}_h^k\right],$$

*Proof.* From the proof of Lemma D.10, we see that with probability at least $1 - \delta$, for any $(k, s, a) \in [K] \times \mathcal{S} \times \mathcal{A}$,

$$\left|\widehat{r}_h^k(s,a) - r_h(s,a)\right| + \left|\inf_{P\in\Delta(S)}\left(\mathbb{E}_P\left[V_{h+1}^{k,\beta,\eta}\right] + \beta\chi^2(P\|\widehat{P}_h^k)\right)(s,a)\right.$$

$$\left. - \inf_{P\in\Delta(S)}\left(\mathbb{E}_P\left[V_{h+1}^{k,\beta,\eta}\right] + \beta\chi^2(P\|P_h^o)\right)(s,a)\right| \leq \text{bonus}_h^k(s,a). \tag{D.54}$$

Recall that we define $P_h^{w,k} = \underset{P\in\Delta(S)}{\arg\min}\left(\mathbb{E}_P\left[V_{h+1}^{\pi^k,\beta,\eta}\right] + \beta\chi^2(P\|P_h^o)\right)$ as the worst-case transition in Definition 5.1, we have

$$Q_h^{k,\beta,\eta} - Q_h^{\pi^k,\beta,\eta}$$

$$\leq \text{bonus}_h^k + \widehat{r}_h^k + \inf_{P\in\Delta(S)}\left(\mathbb{E}_P\left[V_{h+1}^{k,\beta,\eta}\right] + \beta\chi^2(P\|\widehat{P}_h^k)\right) - r_h - \inf_{P\in\Delta(S)}\left(\mathbb{E}_P\left[V_{h+1}^{\pi^k,\beta,\eta}\right] + \beta\chi^2(P\|P_h^o)\right)$$

$$\leq 2\text{bonus}_h^k + \inf_{P\in\Delta(S)}\left(\mathbb{E}_P\left[V_{h+1}^{k,\beta,\eta}\right] + \beta\chi^2(P\|P_h^o)\right) - \inf_{P\in\Delta(S)}\left(\mathbb{E}_P\left[V_{h+1}^{\pi^k,\beta,\eta}\right] + \beta\chi^2(P\|P_h^o)\right) \tag{D.55}$$

$$= 2\text{bonus}_h^k + \mathbb{E}_{P_h^{w,k},\pi^k}\left[Q_{h+1}^{k,\beta,\eta} - Q_{h+1}^{\pi^k,\beta,\eta}\right]. \tag{D.56}$$

where (D.55) uses (D.54). Apply (D.56) recursively, we can obtain the result. $\qquad\square$

*Proof of Lemma D.1 in $\chi^2$ Setting.* The result directly follows from combining Lemma D.10 and Lemma D.11, along with applying a union bound. $\qquad\square$

# E. Proofs of the Results in Linear Setting

## E.1. Proof of the Dynamic Programming Principle

*Proof of Proposition 6.2.* We prove a stronger version of the proposition. Specifically, there exists transition weights $\{\widetilde{\boldsymbol{\mu}}_t\}_{t=1}^H$ together with $\widetilde{P}_t = \langle\boldsymbol{\phi}, \widetilde{\boldsymbol{\mu}}_t\rangle$, such that for all $(h, s) \in [H] \times \mathcal{S}$,

$$V_h^{\pi,\beta,\eta}(s) = \mathbb{E}_{\{\widetilde{P}_t\}_{t=h}^H}\left[\sum_{t=h}^H\left(r_t(s_t,a_t) - \Omega_{s_t}^\eta(\pi_t) + \beta\langle\boldsymbol{\phi}(s_t,a_t), \mathbf{D}(\widetilde{\boldsymbol{\mu}}_t\|\boldsymbol{\mu}_t^o)\rangle\right)\Big| s_h = s, \pi_{h:H}\right]. \tag{E.1}$$

We prove this statement by induction. For the base case $h = H$, the claim holds trivially since no transition kernels are involved. For the inductive step, assume the conclusion holds for step $h + 1$, meaning that there exists $\{\widetilde{r}_t\}_{t=1}^H$ and $\{\widetilde{P}_t\}_{t=h+1}^H$ such that

$$V_{h+1}^{\pi,\beta,\eta}(s) = \mathbb{E}_{\{\widetilde{P}_t\}_{t=h+1}^H}\left[\sum_{t=h+1}^H\left(r_t(s_t,a_t) - \Omega_{s_t}^\eta(\pi_t) + \beta\langle\boldsymbol{\phi}(s_t,a_t), \mathbf{D}(\widetilde{\boldsymbol{\mu}}_t\|\boldsymbol{\mu}_t^o)\rangle\right)\Big| s_{h+1} = s, \pi_{h+1:H}\right]. \tag{E.2}$$

We now prove (E.1) for the case of $h$. For $Q_h^{\pi,\beta,\eta}(s,a)$, on the one hand, we upper bound it as

$$Q_h^{\pi,\beta,\eta}(s,a)$$

$$= \inf_{\substack{\boldsymbol{\mu}_t \in \Delta(S)^d \\ P_t = \langle \boldsymbol{\phi}, \boldsymbol{\mu}_t \rangle}} \mathbb{E}_{\{P_t\}_{t=h}^H} \left[ \sum_{t=h}^H \left( r_t(s_t, a_t) + \beta \langle \boldsymbol{\phi}(s_t, a_t), \mathbf{D}(\boldsymbol{\mu}_t \| \boldsymbol{\mu}_t^o) \rangle \right) - \sum_{t=h+1}^H \Omega_{s_t}^\eta(\pi_t) \Big| s_h = s, a_h = a, \pi_{h+1:H} \right]$$

$$= r_h(s,a) + \inf_{\substack{\boldsymbol{\mu}_t \in \Delta(S)^d \\ P_t = \langle \boldsymbol{\phi}, \boldsymbol{\mu}_t \rangle}} \left\{ \beta \langle \boldsymbol{\phi}(s_h, a_h), \mathbf{D}(\boldsymbol{\mu}_h \| \boldsymbol{\mu}_h^o) \rangle + \int_{\mathcal{S}} P_h(\mathrm{d}s' | s, a) \mathbb{E}_{\{P_t\}_{t=h+1}^H} \left[ \right. \right.$$

$$\left. \left. \sum_{t=h+1}^H \left( r_t(s_t, a_t) - \Omega_{s_t}^\eta(\pi_t) + \beta \langle \boldsymbol{\phi}(s_t, a_t), \mathbf{D}(\boldsymbol{\mu}_t \| \boldsymbol{\mu}_t^o) \rangle \right) \Big| s_{h+1} = s', \pi_{h+1:H} \right] \right\}$$

$$\leq r_h(s,a) + \inf_{\substack{\boldsymbol{\mu}_h \in \Delta(S)^d \\ P_h = \langle \boldsymbol{\phi}, \boldsymbol{\mu}_h \rangle}} \left\{ \beta \langle \boldsymbol{\phi}(s_h, a_h), \mathbf{D}(\boldsymbol{\mu}_h \| \boldsymbol{\mu}_h^o) \rangle + \int_{\mathcal{S}} P_h(\mathrm{d}s' | s, a) \mathbb{E}_{\{\widetilde{P}_t\}_{t=h+1}^H} \left[ \right. \right.$$

$$\left. \left. \sum_{t=h+1}^H \left( r_t(s_t, a_t) - \Omega_{s_t}^\eta(\pi_t) + \beta \langle \boldsymbol{\phi}(s_t, a_t), \mathbf{D}(\widetilde{\boldsymbol{\mu}}_t \| \boldsymbol{\mu}_t^o) \rangle \right) \Big| s_{h+1} = s', \pi_{h+1:H} \right] \right\} \tag{E.3}$$

$$= r_h(s,a) + \inf_{\substack{\boldsymbol{\mu}_h \in \Delta(S)^d \\ P_h = \langle \boldsymbol{\phi}, \boldsymbol{\mu}_h \rangle}} \left\{ \beta \langle \boldsymbol{\phi}(s_h, a_h), \mathbf{D}(\boldsymbol{\mu}_h \| \boldsymbol{\mu}_h^o) \rangle + \mathbb{E}_{s' \sim P_h^o(\cdot | s, a)} \left[ V_{h+1}^{\pi,\beta,\eta}(s') \right] \right\}, \tag{E.4}$$

where (E.3) is because the definition of the infimum operator and (E.4) is from (E.2).

On the other hand, we can lower bound $Q_h^{\pi,\beta,\eta}(s,a)$ as

$$Q_h^{\pi,\beta,\eta}(s,a)$$

$$= \inf_{\substack{\boldsymbol{\mu}_t \in \Delta(S)^d \\ P_t = \langle \boldsymbol{\phi}, \boldsymbol{\mu}_t \rangle}} \mathbb{E}_{\{P_t\}_{t=h}^H} \left[ \sum_{t=h}^H \left( r_t(s_t, a_t) + \beta \langle \boldsymbol{\phi}(s_t, a_t), \mathbf{D}(\boldsymbol{\mu}_t \| \boldsymbol{\mu}_t^o) \rangle \right) - \sum_{t=h+1}^H \Omega_{s_t}^\eta(\pi_t) \Big| s_h = s, a_h = a, \pi_{h+1:H} \right]$$

$$= r_h(s,a) + \inf_{\substack{\boldsymbol{\mu}_t \in \Delta(S)^d \\ P_t = \langle \boldsymbol{\phi}, \boldsymbol{\mu}_t \rangle}} \left\{ \beta \langle \boldsymbol{\phi}(s_h, a_h), \mathbf{D}(\boldsymbol{\mu}_h \| \boldsymbol{\mu}_h^o) \rangle + \int_{\mathcal{S}} P_h(\mathrm{d}s' | s, a) \mathbb{E}_{\{P_t\}_{t=h+1}^H} \left[ \right. \right.$$

$$\left. \left. \sum_{t=h+1}^H \left( r_t(s_t, a_t) - \Omega_{s_t}^\eta(\pi_t) + \beta \langle \boldsymbol{\phi}(s_t, a_t), \mathbf{D}(\boldsymbol{\mu}_t \| \boldsymbol{\mu}_t^o) \rangle \right) \Big| s_{h+1} = s', \pi_{h+1:H} \right] \right\}$$

$$\geq r_h(s,a) + \inf_{\substack{\boldsymbol{\mu}_h \in \Delta(S)^d \\ P_h = \langle \boldsymbol{\phi}, \boldsymbol{\mu}_h \rangle}} \left\{ \beta \langle \boldsymbol{\phi}(s_h, a_h), \mathbf{D}(\boldsymbol{\mu}_h \| \boldsymbol{\mu}_h^o) \rangle + \int_{\mathcal{S}} P_h(\mathrm{d}s' | s, a) \inf_{\substack{\boldsymbol{\mu}_t \in \Delta(S)^d \\ P_t = \langle \boldsymbol{\phi}, \boldsymbol{\mu}_t \rangle}} \mathbb{E}_{\{P_t\}_{t=h+1}^H} \left[ \right. \right.$$

$$\left. \left. \sum_{t=h+1}^H \left( r_t(s_t, a_t) - \Omega_{s_t}^\eta(\pi_t) + \beta \langle \boldsymbol{\phi}(s_t, a_t), \mathbf{D}(\boldsymbol{\mu}_t \| \boldsymbol{\mu}_t^o) \rangle \right) \Big| s_{h+1} = s', \pi_{h+1:H} \right] \right\} \tag{E.5}$$

$$= r_h(s,a) + \inf_{\substack{\boldsymbol{\mu}_h \in \Delta(S)^d \\ P_h = \langle \boldsymbol{\phi}, \boldsymbol{\mu}_h \rangle}} \left\{ \beta \langle \boldsymbol{\phi}(s_h, a_h), \mathbf{D}(\boldsymbol{\mu}_h \| \boldsymbol{\mu}_h^o) \rangle + \mathbb{E}_{s' \sim P_h^o(\cdot | s, a)} \left[ V_{h+1}^{\pi,\beta,\eta}(s') \right] \right\}, \tag{E.6}$$

where (E.5) is because $\inf \int f \geq \int \inf f$ and (E.6) is from the definition of $V_{h+1}^{\pi,\beta,\eta}(s')$.

By combining (E.4) and (E.6), we have

$$Q_h^{\pi,\beta,\eta}(s,a) = r_h(s,a) + \inf_{\substack{\boldsymbol{\mu}_h \in \Delta(S)^d \\ P_h = \langle \boldsymbol{\phi}, \boldsymbol{\mu}_h \rangle}} \left\{ \mathbb{E}_{s' \sim P_h^o(\cdot | s, a)} \left[ V_{h+1}^{\pi,\beta,\eta}(s') \right] + \beta \langle \boldsymbol{\phi}(s,a), \mathbf{D}(\boldsymbol{\mu}_h \| \boldsymbol{\mu}_h^o) \rangle \right\},$$

which implies (6.1).

Since $\Delta(\mathcal{S})$ is compact, there exists $\widetilde{\boldsymbol{\mu}}_h$ together with $\widetilde{P}_h = \langle \boldsymbol{\phi}, \widetilde{\boldsymbol{\mu}}_h \rangle$, such that

$$Q_h^{\pi,\beta,\eta}(s,a) = r_h(s,a) + \mathbb{E}_{s' \sim \widetilde{P}_h(\cdot | s, a)} \left[ V_{h+1}^{\pi,\beta,\eta}(s') \right] + \beta \langle \boldsymbol{\phi}(s,a), \mathbf{D}(\widetilde{\boldsymbol{\mu}}_h \| \boldsymbol{\mu}_h^o) \rangle. \tag{E.7}$$

This finishes the proof of (E.1).

For $V_h^{\pi,\beta,\eta}(s)$, on the one hand, we upper bound it as

$$
V_h^{\pi,\beta,\eta}(s)
$$

$$
= \inf_{\substack{\boldsymbol{\mu}_t \in \Delta(S)^d \\ P_t = \langle \boldsymbol{\phi}, \boldsymbol{\mu}_t \rangle}} \mathbb{E}_{\{P_t\}_{t=h}^H} \left[ \sum_{t=h}^H \left( r_t(s_t, a_t) - \Omega_{s_t}^\eta(\pi_t) + \beta \langle \boldsymbol{\phi}(s_t, a_t), \mathbf{D}(\boldsymbol{\mu}_t \| \boldsymbol{\mu}_t^o) \rangle \right) \Big| s_h = s, \pi_{h:H} \right]
$$

$$
= \inf_{\substack{\boldsymbol{\mu}_t \in \Delta(S)^d \\ P_t = \langle \boldsymbol{\phi}, \boldsymbol{\mu}_t \rangle}} \left\{ -\Omega_s^\eta(\pi_h) + \sum_{a \in \mathcal{A}} \pi_h(a|s) \mathbb{E}_{\{P_t\}_{t=h}^H} \left[ \right. \right.
$$

$$
\left. \left. \sum_{t=h}^H \left( r_t(s_t, a_t) + \beta \langle \boldsymbol{\phi}(s_t, a_t), \mathbf{D}(\boldsymbol{\mu}_t \| \boldsymbol{\mu}_t^o) \rangle \right) - \sum_{t=h+1}^H \Omega_{s_t}^\eta(\pi_t) \Big| s_h = s, a_h = a, \pi_{h+1:H} \right] \right\}
$$

$$
\leq -\Omega_s^\eta(\pi_h) + \sum_{a \in \mathcal{A}} \pi_h(a|s) \mathbb{E}_{\{\widetilde{P}_t\}_{t=h}^H} \left[ \right.
$$

$$
\left. \sum_{t=h}^H \left( r_t(s_t, a_t) + \beta \langle \boldsymbol{\phi}(s_t, a_t), \mathbf{D}(\widetilde{\boldsymbol{\mu}}_t \| \boldsymbol{\mu}_t^o) \rangle \right) - \sum_{t=h+1}^H \Omega_{s_t}^\eta(\pi_t) \Big| s_h = s, a_h = a, \pi_{h+1:H} \right] \tag{E.8}
$$

$$
= -\Omega_s^\eta(\pi_h) + \sum_{a \in \mathcal{A}} \pi_h(a|s) Q_h^{\pi,\beta,\eta}(s, a), \tag{E.9}
$$

where (E.8) is because the definition of the infimum operator and (E.9) is from (E.7) and (E.2).

On the other hand, we can lower bound $V_h^{\pi,\beta,\eta}(s)$ as

$$
V_h^{\pi,\beta,\eta}(s)
$$

$$
= \inf_{\substack{\boldsymbol{\mu}_t \in \Delta(S)^d \\ P_t = \langle \boldsymbol{\phi}, \boldsymbol{\mu}_t \rangle}} \mathbb{E}_{\{P_t\}_{t=h}^H} \left[ \sum_{t=h}^H \left( r_t(s_t, a_t) - \Omega_{s_t}^\eta(\pi_t) + \beta \langle \boldsymbol{\phi}(s_t, a_t), \mathbf{D}(\boldsymbol{\mu}_t \| \boldsymbol{\mu}_t^o) \rangle \right) \Big| s_h = s, \pi_{h:H} \right]
$$

$$
= \inf_{\substack{\boldsymbol{\mu}_t \in \Delta(S)^d \\ P_t = \langle \boldsymbol{\phi}, \boldsymbol{\mu}_t \rangle}} \left\{ -\Omega_s^\eta(\pi_h) + \sum_{a \in \mathcal{A}} \pi_h(a|s) \mathbb{E}_{\{P_t\}_{t=h}^H} \left[ \right. \right.
$$

$$
\left. \left. \sum_{t=h}^H \left( r_t(s_t, a_t) + \beta \langle \boldsymbol{\phi}(s_t, a_t), \mathbf{D}(\boldsymbol{\mu}_t \| \boldsymbol{\mu}_t^o) \rangle \right) - \sum_{t=h+1}^H \Omega_{s_t}^\eta(\pi_t) \Big| s_h = s, a_h = a, \pi_{h+1:H} \right] \right\}
$$

$$
\geq -\Omega_s^\eta(\pi_h) + \sum_{a \in \mathcal{A}} \pi_h(a|s) \inf_{\substack{\boldsymbol{\mu}_t \in \Delta(S)^d \\ P_t = \langle \boldsymbol{\phi}, \boldsymbol{\mu}_t \rangle}} \mathbb{E}_{\{P_t\}_{t=h}^H} \left[ \right.
$$

$$
\left. \sum_{t=h}^H \left( r_t(s_t, a_t) + \beta \langle \boldsymbol{\phi}(s_t, a_t), \mathbf{D}(\boldsymbol{\mu}_t \| \boldsymbol{\mu}_t^o) \rangle \right) - \sum_{t=h+1}^H \Omega_{s_t}^\eta(\pi_t) \Big| s_h = s, a_h = a, \pi_{h+1:H} \right] \tag{E.10}
$$

$$
= -\Omega_s^\eta(\pi_h) + \sum_{a \in \mathcal{A}} \pi_h(a|s) Q_h^{\pi,\beta,\eta}(s, a), \tag{E.11}
$$

where (E.10) is because $\inf \sum f \geq \sum \inf f$ and (E.11) is from the definition of $Q_h^{\pi,\beta,\eta}(s, a)$.

By combining (E.9) and (E.11), we have

$$
V_h^{\pi,\beta,\eta}(s) = -\Omega_s^\eta(\pi_h) + \sum_{a \in \mathcal{A}} \pi_h(a|s) Q_h^{\pi,\beta,\eta}(s, a),
$$

which implies (6.2). $\qquad \square$

### E.2. Proofs of the Main Results

To establish Theorems 6.6 and 6.7, we first introduce Lemma E.1, which bounds the regret by the sum of bonus terms under the expectation defined by the worst-case transition.

**Lemma E.1.** Under Assumption 6.5 and Assumption 6.1, for each $f$-divergence setting and any $\delta \in (0, 1/2)$, if we choose $\lambda = 1$ and set $c$ according to Theorem 6.6 in Algorithm 2, then with probability at least $1 - 2\delta$, the regret of Algorithm 2 satisfies

$$\text{Regret}(K) \leq 2c \sum_{k=1}^{K} \sum_{h=1}^{H} \sum_{i=1}^{d} \mathbb{E}_{P^{w,k}, \pi^k} \left[ \left\| \phi_i(s, a) \mathbb{1}_i \right\|_{(\Lambda_h^k)^{-1}} \right]. \tag{E.12}$$

*Proof of Theorem 6.6.* Recall that

$$\Gamma_h^k(s, a) = c \sum_{i=1}^{d} \left\| \phi_i(s, a) \mathbb{1}_i \right\|_{(\Lambda_h^k)^{-1}}$$

is the bonus assigned to the state-action pair $(s, a)$ at episode $k$ and step $h$. Due to the truncation in (6.3), we may assume without loss of generality that $\Gamma_h^k(s, a) \leq H$ for all $(k, h, s, a) \in [K] \times [H] \times \mathcal{S} \times \mathcal{A}$. Since otherwise we can simply truncate $\Gamma_h^k(s, a)$ at $H$, which does not affect the results.

Therefore, by Azuma-Hoeffding inequality, with probability at least $1 - \delta$, we have

$$\left| \sum_{k=1}^{K} \sum_{h=1}^{H} \mathbb{E}_{P^o, \pi^k} \left[ \Gamma_h^k(s, a) \right] - \sum_{k=1}^{K} \sum_{h=1}^{H} \Gamma_h^k(s_h^k, a_h^k) \right| \leq \sqrt{2KH^3 \ln(2/\delta)}.$$

Now we further analyze (E.12) as

$$\begin{aligned}
\text{Regret}(K) &\leq 2 \sum_{k=1}^{K} \sum_{h=1}^{H} \mathbb{E}_{P^{w,k}, \pi^k} \left[ \Gamma_h^k(s, a) \right] \\
&\leq 2 C_{vr} \sum_{k=1}^{K} \sum_{h=1}^{H} \mathbb{E}_{P^o, \pi^k} \left[ \Gamma_h^k(s, a) \right] \\
&\leq 2 C_{vr} \left( \sqrt{2KH^3 \ln(2/\delta)} + \sum_{k=1}^{K} \sum_{h=1}^{H} \Gamma_h^k(s_h^k, a_h^k) \right) \\
&= 2 C_{vr} \left( \sqrt{2KH^3 \ln(2/\delta)} + c \sum_{k=1}^{K} \sum_{h=1}^{H} \sum_{i=1}^{d} \left\| \phi_i(s_h^k, a_h^k) \mathbb{1}_i \right\|_{(\Lambda_h^k)^{-1}} \right),
\end{aligned} \tag{E.13}$$

where (E.13) follows from Assumption 6.5. This finishes the proof. $\square$

*Proof of Theorem 6.7.* Now we further analyze (E.12) as

$$\begin{aligned}
\text{Regret}(K) &\leq 2c \sum_{k=1}^{K} \sum_{h=1}^{H} \sum_{i=1}^{d} \mathbb{E}_{P^{w,k}, \pi^k} \left[ \left\| \phi_i(s, a) \mathbb{1}_i \right\|_{(\Lambda_h^k)^{-1}} \right] \\
&\leq 2c \sum_{k=1}^{K} \sum_{h=1}^{H} \mathbb{E}_{P^{w,k}, \pi^k} \left[ \sum_{i=1}^{d} \phi_i(s, a) \sqrt{\lambda_{\max}\left((\Lambda_h^k)^{-1}\right)} \right] \\
&= 2c \sum_{k=1}^{K} \sum_{h=1}^{H} \mathbb{E}_{P^{w,k}, \pi^k} \left[ \sqrt{\lambda_{\max}\left((\Lambda_h^k)^{-1}\right)} \right]
\end{aligned} \tag{E.14}$$

$$\leq 2c \sum_{k=1}^{K} \sum_{h=1}^{H} \sqrt{\mathbb{E}_{P^{w,k}, \pi^k} \left[ \lambda_{\max} \left( (\Lambda_h^k)^{-1} \right) \right]} \tag{E.15}$$

$$\leq 2c\sqrt{K} \sum_{h=1}^{H} \sqrt{\sum_{k=1}^{K} \mathbb{E}_{P^{w,k}, \pi^k} \left[ \lambda_{\max} \left( (\Lambda_h^k)^{-1} \right) \right]}, \tag{E.16}$$

where (E.14) is because $\|x\|_A \leq \sqrt{\lambda_{\max}(A)} \|x\|_2$, (E.15) is from Jensen's inequality, (E.16) is from Cauchy-Schwarz inequality.

Using Lemma F.8, with probability at least $1 - \delta$, we have

$$\sum_{k=1}^{K} \mathbb{E}_{P^{w,k}, \pi^k} \left[ \lambda_{\max} \left( (\Lambda_h^k)^{-1} \right) \right]$$

$$= \sum_{k=1}^{K} \mathbb{E}_{P^{w,k}, \pi^k} \left[ \frac{1}{\lambda_{\min}(\Lambda_h^k)} \right]$$

$$\leq \sum_{k=1}^{K} \mathbb{E}_{P^{w,k}, \pi^k} \left[ \frac{1}{\max \left\{ \alpha(k-1) + \lambda - \sqrt{32k \log(dKH/\delta)}, \lambda \right\}} \right]$$

$$= \sum_{k=1}^{K} \frac{1}{\max \left\{ \alpha(k-1) + \lambda - \sqrt{32k \log(dKH/\delta)}, \lambda \right\}}$$

$$\leq \frac{128}{\alpha^2} \log \left( \frac{dHK}{\delta} \right) + \frac{2}{\alpha} \log(K). \tag{E.17}$$

Combining (E.16) and (E.17), we finish the proof. $\qquad \square$

### E.3. Proofs of the Technical Lemmas

Before proving Lemma E.1 in different settings, we state a technical lemma about the covering number.

**Lemma E.2.** For any $h \in [H]$, let $\mathcal{V}_h$ denote a class of functions mapping from $\mathcal{S}$ to $\mathbb{R}$ with the following form

$$V_h(s; w, \beta, \Lambda) = \left[ \max_{\pi \in \Pi} \left\{ \left\langle \pi(s, a), \phi(s, a)^\top w + \beta \sum_{i=1}^{d} \| \phi_i(s, a) \mathbb{1}_i \|_{\Lambda^{-1}} \right\rangle_{\mathcal{A}} - \sigma_{\mathcal{R}_h} \left( -\pi_h(\cdot | s) \right) \right\} \right]_{[0, H-h+1]},$$

where the parameters $(w, \beta, \Lambda)$ satisfy $\|w\| \leq L$, $\beta \in [0, B]$, $\lambda_{\min}(\Lambda) \geq \lambda$. Let $\mathcal{N}_h(\epsilon; \mathcal{V}_h)$ be the $\epsilon$-covering number of $\mathcal{V}_h$ with respect to the distance $\text{dist}(V_1, V_2) = \sup_{s \in \mathcal{S}} |V_1(s) - V_2(s)|$. Then

$$\log \mathcal{N}_h(\epsilon) \leq d \log(1 + 4L|\mathcal{A}|/\epsilon) + d^2 \log \left( 1 + 8d^{1/2} B^2 |\mathcal{A}|^2 / (\lambda \epsilon^2) \right).$$

*Proof.* The argument is similar to that of Liu & Xu (2024a, Lemma D.3). Denoting $A = \beta^2 \Lambda^{-1}$, we have

$$V_h(s; w, \beta, \Lambda) = \left[ \max_{\pi \in \Pi} \left\{ \left\langle \pi(s, a), \phi(s, a)^\top w + \beta \sum_{i=1}^{d} \| \phi_i(s, a) \mathbb{1}_i \|_{\Lambda^{-1}} \right\rangle_{\mathcal{A}} - \sigma_{\mathcal{R}_h} \left( -\pi_h(\cdot | s) \right) \right\} \right]_{[0, H-h+1]},$$

for $\|w\| \leq L$ and $\|A\| \leq B^2 \lambda^{-1}$. For any two functions $V_1, V_2 \in \mathcal{V}_h$, let them take the form above with parameters $(\boldsymbol{w}_1, A_1)$ and $(\boldsymbol{w}_2, A_2)$ respectively. Since $\min\{\cdot, H - h + 1\}$ is a concentration map, we have

$$\text{dist}(V_1, V_2)$$

$$\leq \sup_{s, \pi} \left| \left\{ \left\langle \pi(s, a), \phi(s, a)^\top \boldsymbol{w}_1 + \sum_{i=1}^{d} \| \phi_i(s, a) \mathbb{1}_i \|_{A_1} \right\rangle_{\mathcal{A}} - \sigma_{\mathcal{R}_h} \left( -\pi_h(\cdot | s) \right) \right\} \right.$$

$$- \left\{ \left\langle \pi(s,a), \phi(s,a)^\top \boldsymbol{w}_2 + \sum_{i=1}^d \|\phi_i(s,a)\mathbb{1}_i\|_{A_2} \right\rangle_{\mathcal{A}} - \sigma_{\mathcal{R}_h}(-\pi_h(\cdot|s)) \right\} \right|$$

$$\leq \sup_{s,\pi} \left| \left\langle \pi(s,a), \left( \phi(s,a)^\top \boldsymbol{w}_1 + \sum_{i=1}^d \|\phi_i(s,a)\mathbb{1}_i\|_{A_1} \right) - \left( \phi(s,a)^\top \boldsymbol{w}_2 + \sum_{i=1}^d \|\phi_i(s,a)\mathbb{1}_i\|_{A_2} \right) \right\rangle_{\mathcal{A}} \right|$$

$$\leq \sup_{s,\pi} \left| \left\langle \pi(s,a), \phi(s,a)^\top(\boldsymbol{w}_1 - \boldsymbol{w}_2) \right\rangle \right| + \sup_{s,\pi} \left| \left\langle \pi(s,a), \sum_{i=1}^d \|\phi_i(s,a)\mathbb{1}_i\|_{A_1} - \sum_{i=1}^d \|\phi_i(s,a)\mathbb{1}_i\|_{A_2} \right\rangle_{\mathcal{A}} \right|$$

$$\leq \sup_{s,\pi} \left| \left\langle \pi(s,a), \phi(s,a)^\top(\boldsymbol{w}_1 - \boldsymbol{w}_2) \right\rangle \right| + \sup_{s,\pi} \left| \left\langle \pi(s,a), \sum_{i=1}^d \|\phi_i(s,a)\mathbb{1}_i\|_{A_1 - A_2} \right\rangle_{\mathcal{A}} \right| \qquad \text{(E.18)}$$

$$\leq |\mathcal{A}| \left( \|\boldsymbol{w}_1 - \boldsymbol{w}_2\| + \sqrt{\|A_1 - A_2\|_F} \right),$$

where (E.18) is because $|\sqrt{x} - \sqrt{y}| \leq \sqrt{|x-y|}$ for $x, y \geq 0$. For matrices, $\|\cdot\|$ and $\|\cdot\|_F$ denote the matrix operator norm and Frobenius norm respectively.

Let $\mathcal{C}_{\boldsymbol{w}}$ be an $\epsilon/(2|\mathcal{A}|)$-cover of $\{\boldsymbol{w} \in \mathbb{R}^d \mid \|\boldsymbol{w}\|_2 \leq L\}$ with respect to the 2-norm and $\mathcal{C}_A$ be an $\epsilon^2/(4|\mathcal{A}|^2)$-cover of $\{A \in \mathbb{R}^{d \times d} \mid \|A\|_F \leq d^{1/2}B^2\lambda^{-1}\}$ with respect to the Frobenius norm. By Lemma F.6, we have

$$\log \mathcal{N}_\epsilon \leq \log|\mathcal{C}_{\boldsymbol{w}}| + \log|\mathcal{C}_A| \leq d\log(1 + 4L|\mathcal{A}|/\epsilon) + d^2 \log\left(1 + 8d^{1/2}B^2|\mathcal{A}|^2/(\lambda\epsilon^2)\right).$$

This finishes the proof. $\qquad\square$

Next, we present the proof of Lemma E.1 is various $f$-divergence settings. For convenience, we also write $P^{w,k} := P^{w,\pi^k}$, $\mathrm{d}^k := \mathrm{d}^{\pi^k}$ and $\mathrm{q}^k := \mathrm{q}^{\pi^k}$.

### E.3.1. PROOF OF LEMMA E.1 IN TV SETTING

**Lemma E.3** (Dual formulation). (Tang et al., 2025, Proposition 4.2) For the optimization problem $\inf_{\boldsymbol{\mu} \in \Delta(\mathcal{S})} \left( \mathbb{E}_{s \sim \boldsymbol{\mu}}[V_{h+1}] + \beta \mathrm{TV}(\boldsymbol{\mu} \| \boldsymbol{\mu}^o) \right)$, we have its dual formulation as follows

$$\inf_{\boldsymbol{\mu} \in \Delta(\mathcal{S})} \left( \mathbb{E}_{s \sim \boldsymbol{\mu}}[V_{h+1}] + \beta \mathrm{TV}(\boldsymbol{\mu} \| \boldsymbol{\mu}^o) \right) = \mathbb{E}_{s \sim \boldsymbol{\mu}^o} \left[ V_{h+1}(s) \right]_{\min_{s' \in \mathcal{S}} V_{h+1}(s') + \beta}.$$

**Lemma E.4** (Optimism). If we set the bonus term as follows

$$\Gamma_h^k(s,a) = c_{\mathrm{TV}} \sum_{i=1}^d \|\phi_i(s,a)\mathbb{1}_i\|_{(\Lambda_h^k)^{-1}},$$

$$\text{where} \quad c_{\mathrm{TV}} = Hd \cdot \xi_{\mathrm{TV}},$$

$$\text{where} \quad \xi_{\mathrm{TV}} = 720 + 3\sqrt{40 \log\left(96K^{13/2}H|\mathcal{A}|^3/\delta\right)},$$

then for any policy $\pi$ and any $(k, h, s, a) \in [K] \times [H] \times \mathcal{S} \times \mathcal{A}$, with probability at least $1 - \delta$, we have $Q_h^{k,\beta,\eta}(s,a) \geq Q_h^{\pi,\beta,\eta}(s,a)$. Specially, by setting $\pi = \pi^*$, we have $Q_h^{k,\beta,\eta}(s,a) \geq Q_h^{\pi^*,\beta,\eta}(s,a)$.

*Proof.* We prove this by induction. First, $h = H + 1$ holds trivially since $Q_{H+1}^{k,\beta,\eta}(s,a) = 0 = Q_{H+1}^{\pi,\beta,\eta}(s,a)$.

Assume $Q_{h+1}^{k,\beta,\eta}(s,a) \geq Q_{h+1}^{\pi,\beta,\eta}(s,a)$ holds, due to the choice of $\pi_{h+1}^k$ in (4.3), we have

$$V_{h+1}^{k,\beta,\eta}(s) = \left\langle Q_{h+1}^{k,\beta,\eta}(s,\cdot), \pi_{h+1}^k(\cdot|s) \right\rangle - \Omega_s^\eta(\pi_{h+1}^k)$$

$$\geq \left\langle Q_{h+1}^{k,\beta,\eta}(s,\cdot), \pi_{h+1}(\cdot|s) \right\rangle - \Omega_s^\eta(\pi_{h+1})$$

$$\geq \left\langle Q_{h+1}^{\pi,\beta,\eta}(s,\cdot), \pi_{h+1}(\cdot|s) \right\rangle - \Omega_s^\eta(\pi_{h+1}) = V_{h+1}^{\pi,\beta,\eta}(s).$$

Recall that we denote $Q_h^{k,\beta,\eta}$ as the optimistic estimation in $k$-th episode, that is,

$$Q_h^{k,\beta,\eta}(s,a) \leftarrow \min\left\{ \phi(s,a)^\top \left( \theta_h^k + w_h^k \right) + \Gamma_h^k(s,a), H - h + 1 \right\}. \tag{E.19}$$

If $Q_h^{k,\beta,\eta}(s,a) = H - h + 1$, then it follows immediately that

$$Q_h^{k,\beta,\eta}(s,a) = H - h + 1 \geq Q_h^{\pi,\beta,\eta}(s,a)$$

by the definition of $Q_h^{\pi,\beta,\eta}(s,a)$. Otherwise, we can infer that

$$Q_h^{k,\beta,\eta} - Q_h^{\pi,\beta,\eta}$$

$$= \Gamma_h^k + \phi^\top \left( \theta_h^k + w_h^k \right) - \phi^\top \left[ \theta_h + \inf_{\mu \in \Delta(\mathcal{S})} \left( \mathbb{E}_{s\sim\mu}[V_{h+1}^{\pi,\beta,\eta}] + \beta \mathrm{TV}(\mu\|\mu^o) \right) \right]$$

$$\geq \Gamma_h^k + \phi^\top \left( \theta_h^k + w_h^k \right) - \phi^\top \left[ \theta_h + \inf_{\mu \in \Delta(\mathcal{S})} \left( \mathbb{E}_{s\sim\mu}[V_{h+1}^{k,\beta,\eta}] + \beta \mathrm{TV}(\mu\|\mu^o) \right) \right] \tag{E.20}$$

$$= \Gamma_h^k + \left\langle \phi, \theta_h^k + \widehat{\mathbb{E}}_{s\sim\mu^o}\left[V_{h+1}^{k,\beta,\eta}\right]_{\min_{s'} V_{h+1}^{k,\beta,\eta}(s')+\beta} - \theta_h - \mathbb{E}_{s\sim\mu^o}\left[V_{h+1}^{k,\beta,\eta}\right]_{\min_{s'} V_{h+1}^{k,\beta,\eta}(s')+\beta} \right\rangle \tag{E.21}$$

$$\geq \Gamma_h^k - \left\langle \phi, |\theta_h^k - \theta_h| + \left| \widehat{\mathbb{E}}_{s\sim\mu^o}\left[V_{h+1}^{k,\beta,\eta}\right]_{\min_{s'} V_{h+1}^{k,\beta,\eta}(s')+\beta} - \mathbb{E}_{s\sim\mu^o}\left[V_{h+1}^{k,\beta,\eta}\right]_{\min_{s'} V_{h+1}^{k,\beta,\eta}(s')+\beta} \right| \right\rangle$$

$$= \Gamma_h^k - \sum_{i=1}^d \underbrace{\phi_i \mathbb{1}_i^\top |\theta_h^k - \theta_h|}_{(i)} - \sum_{i=1}^d \underbrace{\phi_i \mathbb{1}_i^\top \left| \widehat{\mathbb{E}}_{s\sim\mu^o}\left[V_{h+1}^{k,\beta,\eta}\right]_{\min_{s'} V_{h+1}^{k,\beta,\eta}(s')+\beta} - \mathbb{E}_{s\sim\mu^o}\left[V_{h+1}^{k,\beta,\eta}\right]_{\min_{s'} V_{h+1}^{k,\beta,\eta}(s')+\beta} \right|}_{(ii)}, \tag{E.22}$$

where (E.20) follows from the induction assumption, and (E.21) is obtained from the algorithm update formulation and the dual formulation.

We introduce the following notation for the sake of convenience

$$\mathrm{err}_h^\tau(f) = \mathbb{E}_{s'\sim\mathbb{P}_h^o(\cdot|s_h^\tau,a_h^\tau)}[f(s')] - f(s_{h+1}^\tau). \tag{E.23}$$

For term (i) in (E.22), we have

$$\left| \phi_i \mathbb{1}_i^\top \left( \theta_h^k - \theta_h \right) \right|$$

$$= \left| \phi_i \mathbb{1}_i^\top \left( \Lambda_h^k \right)^{-1} \left( \sum_{\tau=1}^{k-1} \phi(s_h^\tau, a_h^\tau) \cdot r_h^\tau \right) \right.$$

$$\left. - \phi_i \mathbb{1}_i^\top \left( \Lambda_h^k \right)^{-1} \left( \sum_{\tau=1}^{k-1} \phi(s_h^\tau, a_h^\tau)\phi(s_h^\tau, a_h^\tau)^\top + \lambda \cdot I \right) \cdot \theta_h \right|$$

$$= \lambda \left| \phi_i \mathbb{1}_i^\top \left( \Lambda_h^k \right)^{-1} \cdot \theta_h \right| \tag{E.24}$$

$$\leq \lambda \| \phi_i \mathbb{1}_i \|_{(\Lambda_h^k)^{-1}} \cdot \| \theta_h \|_{(\Lambda_h^k)^{-1}} \tag{E.25}$$

$$\leq \lambda \| \phi_i \mathbb{1}_i \|_{(\Lambda_h^k)^{-1}} \cdot \lambda_{\max}\left( (\Lambda_h^k)^{-1} \right)^{\frac{1}{2}} \| \theta_h \|_2 \tag{E.26}$$

$$\leq \lambda^{\frac{1}{2}} \sqrt{d} \cdot \| \phi_i \mathbb{1}_i \|_{(\Lambda_h^k)^{-1}},$$

where (E.24) is obtained from the algorithm update formulation and the definition of $\Lambda_h^k$, (E.25) follows from the Cauchy-Schwartz inequality, (E.26) is because $\|x\|_A \leq \sqrt{\lambda_{\max}(A)}\|x\|_2$.

For term (ii) in (E.22), we decompose it as follows.

$$\left| \phi_i \mathbb{1}_i^\top \left( \widehat{\mathbb{E}}_{s\sim\mu^o}\left[V_{h+1}^{k,\beta,\eta}\right]_{\min_{s'} V_{h+1}^{k,\beta,\eta}(s')+\beta} - \mathbb{E}_{s\sim\mu^o}\left[V_{h+1}^{k,\beta,\eta}\right]_{\min_{s'} V_{h+1}^{k,\beta,\eta}(s')+\beta} \right) \right|$$

$$
= \left| \phi_i \mathbb{1}_i^\top (\Lambda_h^k)^{-1} \left( \sum_{\tau=1}^{k-1} \phi(s_h^\tau, a_h^\tau) \cdot [V_{h+1}^{k,\beta,\eta}(s_{h+1}^\tau)]_{\min_{s'} V_{h+1}^{k,\beta,\eta}(s')+\beta} \right) \right.
$$
$$
\left. - \phi_i \mathbb{1}_i^\top (\Lambda_h^k)^{-1} \left( \sum_{\tau=1}^{k-1} \phi(s_h^\tau, a_h^\tau) \phi(s_h^\tau, a_h^\tau)^\top + \lambda \cdot I \right) \left( \mathbb{E}_{s\sim\boldsymbol{\mu}^\circ} [V_{h+1}^{k,\beta,\eta}(s)]_{\min_{s'} V_{h+1}^{k,\beta,\eta}(s')+\beta} \right) \right| \tag{E.27}
$$

$$
= \left| \phi_i \mathbb{1}_i^\top (\Lambda_h^k)^{-1} \left( \sum_{\tau=1}^{k-1} \phi(s_h^\tau, a_h^\tau) \cdot \mathrm{err}_h^\tau \left( [V_{h+1}^{k,\beta,\eta}(s)]_{\min_{s'} V_{h+1}^{k,\beta,\eta}(s')+\beta} \right) \right) \right.
$$
$$
\left. + \lambda \phi_i \mathbb{1}_i^\top (\Lambda_h^k)^{-1} \left( \mathbb{E}_{s\sim\boldsymbol{\mu}^\circ} [V_{h+1}^{k,\beta,\eta}(s)]_{\min_{s'} V_{h+1}^{k,\beta,\eta}(s')+\beta} \right) \right| \tag{E.28}
$$

$$
\leq \left| \phi_i \mathbb{1}_i^\top (\Lambda_h^k)^{-1} \left( \sum_{\tau=1}^{k-1} \phi(s_h^\tau, a_h^\tau) \cdot \mathrm{err}_h^\tau \left( [V_{h+1}^{k,\beta,\eta}(s)]_{\min_{s'} V_{h+1}^{k,\beta,\eta}(s')+\beta} \right) \right) \right|
$$
$$
+ \lambda \left| \phi_i \mathbb{1}_i^\top (\Lambda_h^k)^{-1} \left( \mathbb{E}_{s\sim\boldsymbol{\mu}^\circ} [V_{h+1}^{k,\beta,\eta}(s)]_{\min_{s'} V_{h+1}^{k,\beta,\eta}(s')+\beta} \right) \right|
$$

$$
\leq \|\phi_i \mathbb{1}_i\|_{(\Lambda_h^k)^{-1}} \cdot \underbrace{\left\| \sum_{\tau=1}^{k-1} \phi(s_h^\tau, a_h^\tau) \cdot \mathrm{err}_h^\tau \left( [V_{h+1}^{k,\beta,\eta}(s)]_{\min_{s'} V_{h+1}^{k,\beta,\eta}(s')+\beta} \right) \right\|_{(\Lambda_h^k)^{-1}}}_{\text{(iii)}}
$$
$$
+ \lambda \|\phi_i \mathbb{1}_i\|_{(\Lambda_h^k)^{-1}} \cdot \underbrace{\left\| \mathbb{E}_{s\sim\boldsymbol{\mu}^\circ} [V_{h+1}^{k,\beta,\eta}(s)]_{\min_{s'} V_{h+1}^{k,\beta,\eta}(s')+\beta} \right\|_{(\Lambda_h^k)^{-1}}}_{\text{(iv)}}, \tag{E.29}
$$

where (E.27) is obtained from the algorithm update formulation and the definition of $\Lambda_h^k$, (E.28) is from the definition in (E.23), (E.29) follows from the Cauchy-Schwartz inequality.

For term (iv) in (E.29), since $V_{h+1}^{k,\beta,\eta} \leq H$, we have

$$
\left\| \mathbb{E}_{s\sim\boldsymbol{\mu}^\circ} [V_{h+1}^{k,\beta,\eta}]_{\min_{s'} V_{h+1}^{k,\beta,\eta}(s')+\beta} \right\|_{(\Lambda_h^k)^{-1}}
$$
$$
\leq \lambda_{\max}((\Lambda_h^k)^{-1})^{\frac{1}{2}} \left\| \mathbb{E}_{s\sim\boldsymbol{\mu}^\circ} [V_{h+1}^{k,\beta,\eta}]_{\min_{s'} V_{h+1}^{k,\beta,\eta}(s')+\beta} \right\|_2 \tag{E.30}
$$
$$
\leq \lambda^{-\frac{1}{2}} H \sqrt{d},
$$

where (E.30) is because $\|x\|_A \leq \sqrt{\lambda_{\max}(A)} \|x\|_2$.

Before we continue to bound term (iii), we prove a auxiliary result first. That is, $\|\boldsymbol{w}_h^k\|_2 \leq H\sqrt{kd/\lambda}$. From the algorithm update formulation, we have

$$
\|\boldsymbol{w}_h^k\|_2 = \left\| (\Lambda_h^k)^{-1} \left( \sum_{\tau=1}^{k-1} \phi(s_h^\tau, a_h^\tau) \cdot [V_{h+1}^{k,\beta,\eta}(s_{h+1}^\tau)]_{\min_{s'} V_{h+1}^{k,\beta,\eta}(s')+\beta} \right) \right\|_2
$$
$$
\leq H \sum_{\tau=1}^{k-1} \left\| (\Lambda_h^k)^{-1} \phi(s_h^\tau, a_h^\tau) \right\|_2
$$
$$
= H \sum_{\tau=1}^{k-1} \sqrt{\phi(s_h^\tau, a_h^\tau)^\top (\Lambda_h^k)^{-1/2} (\Lambda_h^k)^{-1} (\Lambda_h^k)^{-1/2} \phi(s_h^\tau, a_h^\tau)}
$$
$$
\leq \frac{H}{\sqrt{\lambda}} \sum_{\tau=1}^{k-1} \sqrt{\phi(s_h^\tau, a_h^\tau)^\top (\Lambda_h^k)^{-1} \phi(s_h^\tau, a_h^\tau)}
$$
$$
\leq \frac{H\sqrt{k}}{\sqrt{\lambda}} \sqrt{\sum_{\tau=1}^{k-1} \phi(s_h^\tau, a_h^\tau)^\top (\Lambda_h^k)^{-1} \phi(s_h^\tau, a_h^\tau)}
$$

$$= \frac{H\sqrt{k}}{\sqrt{\lambda}}\sqrt{\mathrm{tr}\big((\Lambda_h^k)^{-1}(\Lambda_h^k - \lambda I)\big)}$$

$$\leq \frac{H\sqrt{k}}{\sqrt{\lambda}}\sqrt{\mathrm{tr}(I)} = H\sqrt{kd/\lambda}.$$

Now we are ready to bound term (iii) in (E.29). Let $\mathcal{V}_h$ denote a class of functions mapping from $\mathcal{S}$ to $\mathbb{R}$ with the following from

$$V(\cdot) = \min\bigg\{\max_{a \in \mathcal{A}}\bigg\{\boldsymbol{\phi}(\cdot, a)^{\top}\boldsymbol{w} + \beta \sum_{i=1}^{d}\|\phi_i(\cdot, a)\mathbb{1}_i\|_{\Lambda^{-1}}\bigg\}, H - h + 1\bigg\},$$

where the parameters $(w, \beta, \Lambda)$ satisfy $\|w\| \leq L$, $\beta \in [0, B]$, $\lambda_{\min} \geq \lambda$. Here we set $L = H\sqrt{kd/\lambda}$ and $B = Hd \cdot \xi_{\mathrm{TV}}$, where $\xi_{\mathrm{TV}} = 720 + 3\sqrt{40\log\big(96K^{13/2}H|\mathcal{A}|^3/\delta\big)}$.

Let $\mathcal{N}_h(\epsilon; \mathcal{V}_h)$ be the minimum $\epsilon$-cover of $\mathcal{V}_h$, $\mathcal{N}_h(\epsilon; [0, H])$ be the minimum $\epsilon$-cover of $[0, H]$. Therefore, for any $V \in \mathcal{V}_h$ and $\alpha \in [0, H]$, there exists a function $V_\epsilon \in \mathcal{N}_h(\epsilon; \mathcal{V}_h)$ and $\alpha_\epsilon \in \mathcal{N}_h(\epsilon; [0, H])$ such that

$$\sup_{s \in \mathcal{S}}\big|V(s) - V_\epsilon(s)\big| \leq \epsilon, \ \Big|\min_{s'}V_{h+1}^{k,\beta,\eta}(s') + \beta - \alpha_\epsilon\Big| \leq \epsilon.$$

Then we have

$$(\mathrm{iii})^2 = \bigg\|\sum_{\tau=1}^{k-1}\boldsymbol{\phi}(s_h^\tau, a_h^\tau) \cdot \mathrm{err}_h^\tau\Big(\big[V_{h+1}^{k,\beta,\eta}\big]_{\min_{s'}V_{h+1}^{k,\beta,\eta}(s')+\beta}\Big)\bigg\|_{(\Lambda_h^k)^{-1}}^2$$

$$\leq 2\bigg\|\sum_{\tau=1}^{k-1}\boldsymbol{\phi}(s_h^\tau, a_h^\tau) \cdot \mathrm{err}_h^\tau\big(\big[V_{h+1}^{k,\beta,\eta}\big]_{\alpha_\epsilon}\big)\bigg\|_{(\Lambda_h^k)^{-1}}^2$$

$$+ 2\bigg\|\sum_{\tau=1}^{k-1}\boldsymbol{\phi}(s_h^\tau, a_h^\tau) \cdot \mathrm{err}_h^\tau\Big(\big[V_{h+1}^{k,\beta,\eta}\big]_{\min_{s'}V_{h+1}^{k,\beta,\eta}(s')+\beta} - \big[V_{h+1}^{k,\beta,\eta}\big]_{\alpha_\epsilon}\Big)\bigg\|_{(\Lambda_h^k)^{-1}}^2 \quad (\mathrm{E}.31)$$

$$\leq 4\bigg\|\sum_{\tau=1}^{k-1}\boldsymbol{\phi}(s_h^\tau, a_h^\tau) \cdot \mathrm{err}_h^\tau\big(\big[V_\epsilon\big]_{\alpha_\epsilon}\big)\bigg\|_{(\Lambda_h^k)^{-1}}^2 + 4\bigg\|\sum_{\tau=1}^{k-1}\boldsymbol{\phi}(s_h^\tau, a_h^\tau) \cdot \mathrm{err}_h^\tau\big(\big[V_{h+1}^{k,\beta,\eta}\big]_{\alpha_\epsilon} - \big[V_\epsilon\big]_{\alpha_\epsilon}\big)\bigg\|_{(\Lambda_h^k)^{-1}}^2$$

$$+ 2(2\epsilon)^2 k^2/\lambda \quad (\mathrm{E}.32)$$

$$\leq 4\bigg\|\sum_{\tau=1}^{k-1}\boldsymbol{\phi}(s_h^\tau, a_h^\tau) \cdot \mathrm{err}_h^\tau\big(\big[V_\epsilon\big]_{\alpha_\epsilon}\big)\bigg\|_{(\Lambda_h^k)^{-1}}^2 + 4(2\epsilon)^2 k^2/\lambda + 2(2\epsilon)^2 k^2/\lambda$$

$$= 4\bigg\|\sum_{\tau=1}^{k-1}\boldsymbol{\phi}(s_h^\tau, a_h^\tau) \cdot \mathrm{err}_h^\tau\big(\big[V_\epsilon\big]_{\alpha_\epsilon}\big)\bigg\|_{(\Lambda_h^k)^{-1}}^2 + 24\epsilon^2 k^2/\lambda$$

$$\leq 4H^2\bigg(d\log(1 + k/\lambda) + 2\log\frac{KH|\mathcal{N}_h(\epsilon; \mathcal{V}_h)| \cdot |\mathcal{N}_h(\epsilon; [0, H])|}{\delta}\bigg) + 24\epsilon^2 k^2/\lambda \quad (\mathrm{E}.33)$$

$$\leq 4H^2\bigg(d\log(1 + k/\lambda) + 2d\log(1 + 4L|\mathcal{A}|/\epsilon) + 2d^2\log\big(1 + 8d^{1/2}B^2|\mathcal{A}|^2/(\lambda\epsilon^2)\big)$$

$$+ 2\log(3H/\epsilon) + 2\log(KH/\delta)\bigg) + 24\epsilon^2 k^2/\lambda \quad (\mathrm{E}.34)$$

$$\leq 4H^2\bigg(2d\log(k) + 4d\log\big(4k^{3/2}|\mathcal{A}|/(d^{1/2})\big) + 4d^2\log\big(8k^2B^2|\mathcal{A}|^2/(H^2d^{3/2})\big)$$

$$+ 2\log(3k/d) + 2\log(KH/\delta)\bigg) + 24H^2d^2 \quad (\mathrm{E}.35)$$

$$\leq 16H^2d^2\log\big(96K^{13/2}B^2|\mathcal{A}|^3/(Hd^3\delta)\big) + 24H^2d^2,$$

where (E.31) and (E.32) is because $\|a + b\|_A^2 \leq 2\|a\|_A^2 + 2\|b\|_A^2$, (E.33) is from Lemma F.7 together with a union bound over all $(k, h) \in [K] \times [H]$, (E.34) makes use of Lemmas E.2 and F.5, we set $\epsilon = Hd/k$ and $\lambda = 1$ and apply the inequality $\log(1 + x) \leq 2\log x$ for $x \geq 2$ in (E.35).

Combining everything together, recall that we set $\lambda = 1$, we have

$$
\begin{aligned}
\text{(i)} + \text{(ii)} &\leq \left( \sqrt{d} + \sqrt{16H^2 d^2 \log\left(96 K^{13/2} B^2 |\mathcal{A}|^3 / (H d^3 \delta)\right) + 24 H^2 d^2} + H\sqrt{d} \right) \cdot \|\phi_i \mathbb{1}_i\|_{(\Lambda_h^k)^{-1}} \\
&\leq 3 H d \sqrt{40 \log\left(96 K^{13/2} B^2 |\mathcal{A}|^3 / (H d^3 \delta)\right)} \cdot \|\phi_i \mathbb{1}_i\|_{(\Lambda_h^k)^{-1}}.
\end{aligned}
$$

With our choice of $B$, and noting that $x \geq \max\left\{ (2B + \sqrt{A})^2, 1 \right\}$ guarantees $x \geq A + B \log x$, we conclude the proof. $\qquad\square$

**Lemma E.5.** *If we set the bonus term to be the same as in Lemma E.4, then for any $(k, s, a) \in [K] \times \mathcal{S} \times \mathcal{A}$, with probability at least $1 - \delta$, the sum of estimation errors can be bounded as*

$$
Q_1^{k, \beta, \eta}(s, a) - Q_1^{\pi^k, \beta, \eta}(s, a) \leq 2 \cdot \mathbb{E}_{P^{w,k}, \pi^k}\left[ \sum_{h=1}^{H} \Gamma_h^k(s_h^k, a_h^k) \right],
$$

*where $P_h^{w,k}$ is defined by $P_h^{w,k}(s, a) = \phi(s, a)^\top \cdot \boldsymbol{\mu}_h^{w,k}$.*

*Proof.* From the definition, We have that

$$
\begin{aligned}
&Q_h^{k, \beta, \eta} - Q_h^{\pi^k, \beta, \eta} \\
&= \Gamma_h^k + \phi^\top\left(\boldsymbol{\theta}_h^k + \boldsymbol{w}_h^k\right) - \phi^\top\left[ \boldsymbol{\theta}_h + \inf_{\boldsymbol{\mu}_h \in \Delta(\mathcal{S})}\left( \mathbb{E}_{s \sim \boldsymbol{\mu}_h}[V_{h+1}^{\pi^k, \beta, \eta}] + \beta \text{TV}(\boldsymbol{\mu}_h \| \boldsymbol{\mu}_h^o) \right) \right] \\
&= \Gamma_h^k + \phi^\top\left(\boldsymbol{\theta}_h^k - \boldsymbol{\theta}_h\right) + \phi^\top\left[ \boldsymbol{w}_h^k - \inf_{\boldsymbol{\mu}_h \in \Delta(\mathcal{S})}\left( \mathbb{E}_{s \sim \boldsymbol{\mu}_h}[V_{h+1}^{k, \beta, \eta}] + \beta \text{TV}(\boldsymbol{\mu}_h \| \boldsymbol{\mu}_h^o) \right) \right] \\
&\quad + \phi^\top\left[ \inf_{\boldsymbol{\mu}_h \in \Delta(\mathcal{S})}\left( \mathbb{E}_{s \sim \boldsymbol{\mu}_h}[V_{h+1}^{k, \beta, \eta}] + \beta \text{TV}(\boldsymbol{\mu}_h \| \boldsymbol{\mu}_h^o) \right) - \inf_{\boldsymbol{\mu}_h \in \Delta(\mathcal{S})}\left( \mathbb{E}_{s \sim \boldsymbol{\mu}_h}[V_{h+1}^{\pi^k, \beta, \eta}] + \beta \text{TV}(\boldsymbol{\mu}_h \| \boldsymbol{\mu}_h^o) \right) \right] \\
&\leq 2 \cdot \Gamma_h^k + \phi^\top\left[ \inf_{\boldsymbol{\mu}_h \in \Delta(\mathcal{S})}\left( \mathbb{E}_{s \sim \boldsymbol{\mu}_h}[V_{h+1}^{k, \beta, \eta}] + \beta \text{TV}(\boldsymbol{\mu}_h \| \boldsymbol{\mu}_h^o) \right) - \inf_{\boldsymbol{\mu}_h \in \Delta(\mathcal{S})}\left( \mathbb{E}_{s \sim \boldsymbol{\mu}_h}[V_{h+1}^{\pi^k, \beta, \eta}] + \beta \text{TV}(\boldsymbol{\mu}_h \| \boldsymbol{\mu}_h^o) \right) \right] \quad \text{(E.36)} \\
&\leq 2 \cdot \Gamma_h^k + \phi^\top \mathbb{E}_{s \sim \boldsymbol{\mu}_h^{w,k}}\left[ V_{h+1}^{k, \beta, \eta} - V_{h+1}^{\pi^k, \beta, \eta} \right] \quad\quad\quad\quad\quad\quad\quad\quad\quad\quad\quad\quad\quad\quad\quad\quad\quad\quad\quad\quad \text{(E.37)} \\
&\leq 2 \cdot \Gamma_h^k + \mathbb{E}_{s \sim P_h^w}\left[ V_{h+1}^{k, \beta, \eta} - V_{h+1}^{\pi^k, \beta, \eta} \right],
\end{aligned}
$$

where (E.36) can be derived from Lemma E.4, and (E.37) uses the definition of the worst-case transition. Apply the above inequality recursively, we have

$$
Q_1^{k, \beta, \eta} - Q_1^{\pi^k, \beta, \eta} \leq 2 \cdot \mathbb{E}_{P^{w,k}, \pi^k}\left[ \sum_{h=1}^{H} \Gamma_h^k \right].
$$

This finishes the proof. $\qquad\square$

*Proof of Lemma E.1 in TV Setting.* The result directly follows from combining Lemma E.4 and Lemma E.5, along with applying a union bound. $\qquad\square$

### E.3.2. PROOF OF LEMMA E.1 IN KL SETTING

**Lemma E.6** (Dual formulation). *(Tang et al., 2025, Proposition 4.5) For the optimization problem $\inf_{\boldsymbol{\mu} \in \Delta(\mathcal{S})}\left( \mathbb{E}_{s \sim \boldsymbol{\mu}}[V_{h+1}] + \beta \text{KL}(\boldsymbol{\mu} \| \boldsymbol{\mu}^o) \right)$, we have its dual formulation as follows*

$$
\inf_{\boldsymbol{\mu} \in \Delta(\mathcal{S})}\left( \mathbb{E}_{s \sim \boldsymbol{\mu}}[V_{h+1}] + \beta \text{KL}(\boldsymbol{\mu} \| \boldsymbol{\mu}^o) \right) = -\beta \log \mathbb{E}_{s \sim \boldsymbol{\mu}^o}\left[ e^{-V_{h+1}(s)/\beta} \right].
$$

**Lemma E.7** (Optimism). If we set the bonus term as follows

$$\Gamma_h^k(s,a) = c_{\mathrm{KL}} \sum_{i=1}^d \|\phi_i(s,a)\mathbb{1}_i\|_{(\Lambda_h^k)^{-1}},$$

$$\text{where} \quad c_{\mathrm{KL}} = (1 + 2\beta e^{\beta^{-1}H}) H d \cdot \xi_{\mathrm{KL}},$$

$$\text{where} \quad \xi_{\mathrm{KL}} = 80 + \sqrt{40\log\left(64\beta K^{11/2} H d^{1/2}(1 + 2\beta e^{\beta^{-1}H})^2 |\mathcal{A}|^3/\delta\right)},$$

then for any policy $\pi$ and any $(k,h,s,a) \in [K] \times [H] \times \mathcal{S} \times \mathcal{A}$, with probability at least $1 - \delta$, we have $Q_h^{k,\beta,\eta}(s,a) \geq Q_h^{\pi,\beta,\eta}(s,a)$. Specially, by setting $\pi = \pi^*$, we have $Q_h^{k,\beta,\eta}(s,a) \geq Q_h^{\pi^*,\beta,\eta}(s,a)$.

*Proof.* We prove this by induction. First, $h = H + 1$ holds trivially since $Q_{H+1}^{k,\beta,\eta}(s,a) = 0 = Q_{H+1}^{\pi,\beta,\eta}(s,a)$.

Assume $Q_{h+1}^{k,\beta,\eta}(s,a) \geq Q_{h+1}^{\pi,\beta,\eta}(s,a)$ holds, due to the choice of $\pi_{h+1}^k$ in (4.3), we have

$$\begin{aligned}
V_{h+1}^{k,\beta,\eta}(s) &= \langle Q_{h+1}^{k,\beta,\eta}(s,\cdot), \pi_{h+1}^k(\cdot|s)\rangle - \Omega_s^\eta(\pi_{h+1}^k) \\
&\geq \langle Q_{h+1}^{k,\beta,\eta}(s,\cdot), \pi_{h+1}(\cdot|s)\rangle - \Omega_s^\eta(\pi_{h+1}) \\
&\geq \langle Q_{h+1}^{\pi,\beta,\eta}(s,\cdot), \pi_{h+1}(\cdot|s)\rangle - \Omega_s^\eta(\pi_{h+1}) = V_{h+1}^{\pi,\beta,\eta}(s).
\end{aligned}$$

Recall that we denote $Q_h^{k,\beta,\eta}$ as the optimistic estimation in $k$-th episode, that is,

$$Q_h^{k,\beta,\eta}(s,a) \leftarrow \min\left\{\phi(s,a)^\top(\boldsymbol{\theta}_h^k + \boldsymbol{w}_h^k) + \Gamma_h^k(s,a), H - h + 1\right\}. \tag{E.38}$$

If $Q_h^{k,\beta,\eta}(s,a) = H - h + 1$, then it follows immediately that

$$Q_h^{k,\beta,\eta}(s,a) = H - h + 1 \geq Q_h^{\pi,\beta,\eta}(s,a)$$

by the definition of $Q_h^{\pi,\beta,\eta}(s,a)$. Otherwise, we can infer that

$$\begin{aligned}
&Q_h^{k,\beta,\eta} - Q_h^{\pi,\beta,\eta} \\
&= \Gamma_h^k + \phi^\top(\boldsymbol{\theta}_h^k + \boldsymbol{w}_h^k) - \phi^\top\left[\boldsymbol{\theta}_h + \inf_{\mu \in \Delta(\mathcal{S})}\left(\mathbb{E}_{s\sim\mu}[V_{h+1}^{\pi,\beta,\eta}] + \beta\mathrm{KL}(\mu\|\mu^o))\right)\right] \\
&\geq \Gamma_h^k + \phi^\top(\boldsymbol{\theta}_h^k + \boldsymbol{w}_h^k) - \phi^\top\left[\boldsymbol{\theta}_h + \inf_{\mu \in \Delta(\mathcal{S})}\left(\mathbb{E}_{s\sim\mu}[V_{h+1}^{k,\beta,\eta}] + \beta\mathrm{KL}(\mu\|\mu^o))\right)\right] \tag{E.39} \\
&= \Gamma_h^k + \left\langle\phi, \boldsymbol{\theta}_h^k - \beta\log\max\left\{\widehat{\mathbb{E}}_{s\sim\boldsymbol{\mu}^o}\left[e^{-V_{h+1}^{k,\beta,\eta}(s)/\beta}\right], e^{-H/\beta}\right\} - \boldsymbol{\theta}_h + \beta\log\mathbb{E}_{s\sim\boldsymbol{\mu}^o}\left[e^{-V_{h+1}^{k,\beta,\eta}(s)/\beta}\right]\right\rangle \tag{E.40} \\
&\geq \Gamma_h^k - \left\langle\phi, |\boldsymbol{\theta}_h^k - \boldsymbol{\theta}_h| + \beta e^{\beta^{-1}H}\left|\max\left\{\widehat{\mathbb{E}}_{s\sim\boldsymbol{\mu}^o}\left[e^{-V_{h+1}^{k,\beta,\eta}(s)/\beta}\right], e^{-H/\beta}\right\} - \mathbb{E}_{s\sim\boldsymbol{\mu}^o}\left[e^{-V_{h+1}^{k,\beta,\eta}(s)/\beta}\right]\right|\right\rangle \\
&\geq \Gamma_h^k - \left\langle\phi, |\boldsymbol{\theta}_h^k - \boldsymbol{\theta}_h| + \beta e^{\beta^{-1}H}\left|\widehat{\mathbb{E}}_{s\sim\boldsymbol{\mu}^o}\left[e^{-V_{h+1}^{k,\beta,\eta}(s)/\beta}\right] - \mathbb{E}_{s\sim\boldsymbol{\mu}^o}\left[e^{-V_{h+1}^{k,\beta,\eta}(s)/\beta}\right]\right|\right\rangle \\
&= \Gamma_h^k - \sum_{i=1}^d \underbrace{\phi_i\mathbb{1}_i^\top|\boldsymbol{\theta}_h^k - \boldsymbol{\theta}_h|}_{(i)} - \beta e^{\beta^{-1}H}\sum_{i=1}^d \underbrace{\phi_i\mathbb{1}_i^\top\left|\widehat{\mathbb{E}}_{s\sim\boldsymbol{\mu}^o}\left[e^{-V_{h+1}^{k,\beta,\eta}(s)/\beta}\right] - \mathbb{E}_{s\sim\boldsymbol{\mu}^o}\left[e^{-V_{h+1}^{k,\beta,\eta}(s)/\beta}\right]\right|}_{(ii)}, \tag{E.41}
\end{aligned}$$

where (E.39) follows from the induction assumption, and (E.40) is obtained from the algorithm update formulation and the dual formulation.

For term (i) in (E.41), we have

$$\begin{aligned}
&\left|\phi_i\mathbb{1}_i^\top(\boldsymbol{\theta}_h^k - \boldsymbol{\theta}_h)\right| \\
&= \left|\phi_i\mathbb{1}_i^\top(\Lambda_h^k)^{-1}\left(\sum_{\tau=1}^{k-1}\phi(s_h^\tau, a_h^\tau)\cdot r_h^\tau\right)\right|
\end{aligned}$$

$$- \phi_i \mathbb{1}_i^\top (\Lambda_h^k)^{-1} \left( \sum_{\tau=1}^{k-1} \phi(s_h^\tau, a_h^\tau) \phi(s_h^\tau, a_h^\tau)^\top + \lambda \cdot I \right) \cdot \boldsymbol{\theta}_h \Bigg|$$

$$= \lambda \left| \phi_i \mathbb{1}_i^\top (\Lambda_h^k)^{-1} \cdot \boldsymbol{\theta}_h \right| \tag{E.42}$$

$$\leq \lambda \| \phi_i \mathbb{1}_i \|_{(\Lambda_h^k)^{-1}} \cdot \| \boldsymbol{\theta}_h \|_{(\Lambda_h^k)^{-1}} \tag{E.43}$$

$$\leq \lambda \| \phi_i \mathbb{1}_i \|_{(\Lambda_h^k)^{-1}} \cdot \lambda_{\max}\big((\Lambda_h^k)^{-1}\big)^{\frac{1}{2}} \| \boldsymbol{\theta}_h \|_2 \tag{E.44}$$

$$\leq \lambda^{\frac{1}{2}} \sqrt{d} \cdot \| \phi_i \mathbb{1}_i \|_{(\Lambda_h^k)^{-1}},$$

where (E.42) is obtained from the algorithm update formulation and the definition of $\Lambda_h^k$, (E.43) follows from the Cauchy-Schwartz inequality, (E.44) is because $\|x\|_A \leq \sqrt{\lambda_{\max}(A)} \|x\|_2$.

For term (ii) in (E.41), we decompose it as follows.

$$\left| \phi_i \mathbb{1}_i^\top \left( \widehat{\mathbb{E}}_{s \sim \boldsymbol{\mu}^o} \big[ e^{-V_{h+1}^{k,\beta,\eta}(s)/\beta} \big] - \mathbb{E}_{s \sim \boldsymbol{\mu}^o} \big[ e^{-V_{h+1}^{k,\beta,\eta}(s)/\beta} \big] \right) \right|$$

$$= \Bigg| \phi_i \mathbb{1}_i^\top (\Lambda_h^k)^{-1} \left( \sum_{\tau=1}^{k-1} \phi(s_h^\tau, a_h^\tau) \cdot e^{-V_{h+1}^{k,\beta,\eta}(s_{h+1}^\tau)/\beta} \right)$$

$$- \phi_i \mathbb{1}_i^\top (\Lambda_h^k)^{-1} \left( \sum_{\tau=1}^{k-1} \phi(s_h^\tau, a_h^\tau) \phi(s_h^\tau, a_h^\tau)^\top + \lambda \cdot I \right) \mathbb{E}_{s \sim \boldsymbol{\mu}^o} \big[ e^{-V_{h+1}^{k,\beta,\eta}(s)/\beta} \big] \Bigg| \tag{E.45}$$

$$= \Bigg| \phi_i \mathbb{1}_i^\top (\Lambda_h^k)^{-1} \left( \sum_{\tau=1}^{k-1} \phi(s_h^\tau, a_h^\tau) \cdot \mathrm{err}_h^\tau \big( e^{-V_{h+1}^{k,\beta,\eta}(s)/\beta} \big) \right)$$

$$+ \lambda \phi_i \mathbb{1}_i^\top (\Lambda_h^k)^{-1} \mathbb{E}_{s \sim \boldsymbol{\mu}^o} \big[ e^{-V_{h+1}^{k,\beta,\eta}(s)/\beta} \big] \Bigg| \tag{E.46}$$

$$\leq \Bigg| \phi_i \mathbb{1}_i^\top (\Lambda_h^k)^{-1} \left( \sum_{\tau=1}^{k-1} \phi(s_h^\tau, a_h^\tau) \cdot \mathrm{err}_h^\tau \big( e^{-V_{h+1}^{k,\beta,\eta}(s)/\beta} \big) \right) \Bigg|$$

$$+ \lambda \left| \phi_i \mathbb{1}_i^\top (\Lambda_h^k)^{-1} \mathbb{E}_{s \sim \boldsymbol{\mu}^o} \big[ e^{-V_{h+1}^{k,\beta,\eta}(s)/\beta} \big] \right|$$

$$\leq \| \phi_i \mathbb{1}_i \|_{(\Lambda_h^k)^{-1}} \cdot \underbrace{\left\| \sum_{\tau=1}^{k-1} \phi(s_h^\tau, a_h^\tau) \cdot \mathrm{err}_h^\tau \big( e^{-V_{h+1}^{k,\beta,\eta}(s)/\beta} \big) \right\|_{(\Lambda_h^k)^{-1}}}_{\text{(iii)}}$$

$$+ \lambda \| \phi_i \mathbb{1}_i \|_{(\Lambda_h^k)^{-1}} \cdot \underbrace{\left\| \mathbb{E}_{s \sim \boldsymbol{\mu}^o} \big[ e^{-V_{h+1}^{k,\beta,\eta}(s)/\beta} \big] \right\|_{(\Lambda_h^k)^{-1}}}_{\text{(iv)}}, \tag{E.47}$$

where (E.45) is obtained from the algorithm update formulation and the definition of $\Lambda_h^k$, (E.46) is from the definition in (E.23), (E.47) follows from the Cauchy-Schwartz inequality.

For term (iv) in (E.47), since $e^{-V_{h+1}^{k,\beta,\eta}(s)/\beta} \leq 1$, we have

$$\left\| \mathbb{E}_{s \sim \boldsymbol{\mu}^o} \big[ e^{-V_{h+1}^{k,\beta,\eta}(s)/\beta} \big] \right\|_{(\Lambda_h^k)^{-1}}$$

$$\leq \lambda_{\max}\big((\Lambda_h^k)^{-1}\big)^{\frac{1}{2}} \left\| \mathbb{E}_{s \sim \boldsymbol{\mu}^o} \big[ e^{-V_{h+1}^{k,\beta,\eta}(s)/\beta} \big] \right\|_2 \tag{E.48}$$

$$\leq \lambda^{-\frac{1}{2}} \sqrt{d},$$

where (E.48) is because $\|x\|_A \leq \sqrt{\lambda_{\max}(A)} \|x\|_2$.

Before we continue to bound term (iii), we prove a auxiliary result first. That is, $\|\boldsymbol{w}_h^k\|_2 \leq \sqrt{kd/\lambda}$. From the algorithm

update formulation, we have

$$
\begin{aligned}
\left\| \widehat{\mathbb{E}}_{s\sim\boldsymbol{\mu}^\circ}\left[e^{-V_{h+1}^{k,\beta,\eta}(s)/\beta}\right] \right\|_2 &= \left\| \left(\Lambda_h^k\right)^{-1}\left(\sum_{\tau=1}^{k-1}\boldsymbol{\phi}(s_h^\tau,a_h^\tau)\cdot e^{-V_{h+1}^{k,\beta,\eta}(s_{h+1}^\tau)/\beta}\right) \right\|_2 \\
&\leq \sum_{\tau=1}^{k-1}\left\| \left(\Lambda_h^k\right)^{-1}\boldsymbol{\phi}(s_h^\tau,a_h^\tau) \right\|_2 \\
&= \sum_{\tau=1}^{k-1}\sqrt{\boldsymbol{\phi}(s_h^\tau,a_h^\tau)^\top\left(\Lambda_h^k\right)^{-1/2}\left(\Lambda_h^k\right)^{-1}\left(\Lambda_h^k\right)^{-1/2}\boldsymbol{\phi}(s_h^\tau,a_h^\tau)} \\
&\leq \frac{1}{\sqrt{\lambda}}\sum_{\tau=1}^{k-1}\sqrt{\boldsymbol{\phi}(s_h^\tau,a_h^\tau)^\top\left(\Lambda_h^k\right)^{-1}\boldsymbol{\phi}(s_h^\tau,a_h^\tau)} \\
&\leq \frac{\sqrt{k}}{\sqrt{\lambda}}\sqrt{\sum_{\tau=1}^{k-1}\boldsymbol{\phi}(s_h^\tau,a_h^\tau)^\top\left(\Lambda_h^k\right)^{-1}\boldsymbol{\phi}(s_h^\tau,a_h^\tau)} \\
&= \frac{\sqrt{k}}{\sqrt{\lambda}}\sqrt{\mathrm{tr}\left(\left(\Lambda_h^k\right)^{-1}\left(\Lambda_h^k-\lambda I\right)\right)} \\
&\leq \frac{\sqrt{k}}{\sqrt{\lambda}}\sqrt{\mathrm{tr}(I)} = \sqrt{kd/\lambda}.
\end{aligned}
$$

Therefore,

$$
\begin{aligned}
\|\boldsymbol{w}_h^k\|_2 &= \left\| \beta\log\max\left\{\widehat{\mathbb{E}}_{s\sim\boldsymbol{\mu}^\circ}\left[e^{-V_{h+1}^{k,\beta,\eta}(s)/\beta}\right], e^{-H/\beta}\right\} \right\|_2 \\
&\leq \beta\sqrt{d}\max\left\{\log\left(\left\|\widehat{\mathbb{E}}_{s\sim\boldsymbol{\mu}^\circ}\left[e^{-V_{h+1}^{k,\beta,\eta}(s)/\beta}\right]\right\|_2\right), H/\beta\right\} \\
&\leq \beta\sqrt{d}\max\left\{\log\left(\sqrt{kd/\lambda}\right), H/\beta\right\} \\
&\leq \beta\sqrt{d}\left(\sqrt{kd/\lambda}+H/\beta\right) \\
&\leq 2\beta Hd\sqrt{k}.
\end{aligned}
$$

Now we are ready to bound term (iii) in (E.47). Let $\mathcal{V}_h$ denote a class of functions mapping from $\mathcal{S}$ to $\mathbb{R}$ with the following from

$$
V(\cdot) = \min\left\{ \max_{a\in\mathcal{A}}\left\{\boldsymbol{\phi}(\cdot,a)^\top\boldsymbol{w}+\beta\sum_{i=1}^d\|\phi_i(\cdot,a)\mathbb{1}_i\|_{\Lambda^{-1}}\right\}, H-h+1\right\},
$$

where the parameters $(w,\beta,\Lambda)$ satisfy $\|w\|\leq L$, $\beta\in[0,B]$, $\lambda_{\min}\geq\lambda$. Here we set $L=2\beta Hd\sqrt{k}$ and $B=(1+2\beta e^{\beta^{-1}H})Hd\cdot\xi_{\mathrm{KL}}$, where $\xi_{\mathrm{KL}}=80+\sqrt{40\log\left(64\beta K^{11/2}Hd^{1/2}(1+2\beta e^{\beta^{-1}H})^2|\mathcal{A}|^3/\delta\right)}$.

Let $\mathcal{N}_h(\epsilon;\mathcal{V}_h)$ be the minimum $\epsilon$-cover of $\mathcal{V}_h$. Therefore, for any $V\in\mathcal{V}_h$ and $\alpha\in[0,H]$, there exists a function $V_\epsilon\in\mathcal{N}_h(\epsilon;\mathcal{V}_h)$ and $\alpha_\epsilon\in\mathcal{N}_h(\epsilon;[0,H])$ such that

$$
\sup_{s\in\mathcal{S}}\left|V(s)-V_\epsilon(s)\right|\leq\epsilon.
$$

Then we have

$$
\begin{aligned}
\text{(iii)}^2 &= \left\| \sum_{\tau=1}^{k-1}\boldsymbol{\phi}(s_h^\tau,a_h^\tau)\cdot\mathrm{err}_h^\tau\left(e^{-V_{h+1}^{k,\beta,\eta}(s)/\beta}\right) \right\|_{(\Lambda_h^k)^{-1}}^2 \\
&\leq 2\left\| \sum_{\tau=1}^{k-1}\boldsymbol{\phi}(s_h^\tau,a_h^\tau)\cdot\mathrm{err}_h^\tau\left(e^{-V_\epsilon(s)/\beta}\right) \right\|_{(\Lambda_h^k)^{-1}}^2
\end{aligned}
$$

$$+ 2 \left\| \sum_{\tau=1}^{k-1} \phi(s_h^\tau, a_h^\tau) \cdot \mathrm{err}_h^\tau \left( e^{-V_{h+1}^{k,\beta,\eta}(s)/\beta} - e^{-V_\epsilon(s)/\beta} \right) \right\|_{(\Lambda_h^k)^{-1}}^2 \tag{E.49}$$

$$\leq 2 \left\| \sum_{\tau=1}^{k-1} \phi(s_h^\tau, a_h^\tau) \cdot \mathrm{err}_h^\tau \left( e^{-V_\epsilon(s)/\beta} \right) \right\|_{(\Lambda_h^k)^{-1}}^2 + 2(4\epsilon)^2 k^2/\lambda \tag{E.50}$$

$$\leq 4 \left( d \log(1 + k/\lambda) + 2 \log \frac{KH|\mathcal{N}_h(\epsilon; \mathcal{V}_h)|}{\delta} \right) + 32\epsilon^2 k^2/\lambda \tag{E.51}$$

$$\leq 4H^2 \Big( d \log(1 + k/\lambda) + 2d \log(1 + 4L|\mathcal{A}|/\epsilon) + 2d^2 \log \left( 1 + 8d^{1/2} B^2 |\mathcal{A}|^2/(\lambda\epsilon^2) \right)$$

$$+ 2 \log(KH/\delta) \Big) + 32\epsilon^2 k^2/\lambda \tag{E.52}$$

$$\leq 4H^2 \Big( 2d \log(k) + 4d \log \left( 8\beta k^{3/2}|\mathcal{A}| \right) + 4d^2 \log \left( 8k^2 B^2 |\mathcal{A}|^2/(H^2 d^{3/2}) \right)$$

$$+ 2 \log(KH/\delta) \Big) + 32H^2 d^2 \tag{E.53}$$

$$\leq 16H^2 d^2 \log \left( 64\beta K^{11/2} B^2 |\mathcal{A}|^3/(Hd^{3/2}\delta) \right) + 32H^2 d^2,$$

where (E.49) is because $\|a + b\|_A^2 \leq 2\|a\|_A^2 + 2\|b\|_A^2$, (E.50) follows from the inequality $|e^x - e^y| \leq e^{|x-y|} - 1$ for $\max\{x, y\} \leq 0$ and $e^t \leq 1 + 2t$ for $t \in [0, 1]$, (E.51) is from Lemma F.7 together with a union bound over all $(k, h) \in [K] \times [H]$, (E.52) makes use of Lemma E.2, we set $\epsilon = Hd/k$ and $\lambda = 1$ and apply the inequality $\log(1 + x) \leq 2 \log x$ for $x \geq 2$ in (E.53).

Combining everything together, recall that we set $\lambda = 1$, we have

$$(\mathrm{i}) + \beta e^{\beta^{-1}H}(\mathrm{ii}) \leq \sqrt{d} \cdot \|\phi_i \mathbb{1}_i\|_{(\Lambda_h^k)^{-1}}$$

$$+ \beta e^{\beta^{-1}H} \left( \sqrt{16H^2 d^2 \log \left( 64\beta K^{11/2} B^2 |\mathcal{A}|^3/(Hd^{3/2}\delta) \right) + 32H^2 d^2} + \sqrt{d} \right) \cdot \|\phi_i \mathbb{1}_i\|_{(\Lambda_h^k)^{-1}}$$

$$\leq (1 + 2\beta e^{\beta^{-1}H}) Hd \sqrt{40 \log \left( 64\beta K^{11/2} B^2 |\mathcal{A}|^3/(Hd^{3/2}\delta) \right)} \cdot \|\phi_i \mathbb{1}_i\|_{(\Lambda_h^k)^{-1}}.$$

With our choice of $B$, and noting that $x \geq \max \left\{ (2B + \sqrt{A})^2, 1 \right\}$ guarantees $x \geq A + B \log x$, we conclude the proof. $\qquad \square$

**Lemma E.8.** If we set the bonus term to be the same as in Lemma E.7, then for any $(k, s, a) \in [K] \times \mathcal{S} \times \mathcal{A}$, with probability at least $1 - \delta$, the sum of estimation errors can be bounded as

$$Q_1^{k,\beta,\eta}(s, a) - Q_1^{\pi^k,\beta,\eta}(s, a) \leq 2 \cdot \mathbb{E}_{P^{w,k}, \pi^k} \left[ \sum_{h=1}^H \Gamma_h^k(s_h^k, a_h^k) \right],$$

where $P_h^{w,k}$ is defined by $P_h^{w,k}(s, a) = \phi(s, a)^\top \cdot \boldsymbol{\mu}_h^{w,k}$.

*Proof.* From the definition, We have that

$$Q_h^{k,\beta,\eta} - Q_h^{\pi^k,\beta,\eta}$$

$$= \Gamma_h^k + \phi^\top \left( \boldsymbol{\theta}_h^k + \boldsymbol{w}_h^k \right) - \phi^\top \left[ \boldsymbol{\theta}_h + \inf_{\boldsymbol{\mu}_h \in \Delta(\mathcal{S})} \left( \mathbb{E}_{s \sim \boldsymbol{\mu}_h}[V_{h+1}^{\pi^k,\beta,\eta}] + \beta \mathrm{KL}(\boldsymbol{\mu}_h \| \boldsymbol{\mu}_h^o) \right) \right]$$

$$= \Gamma_h^k + \phi^\top \left( \boldsymbol{\theta}_h^k - \boldsymbol{\theta}_h \right) + \phi^\top \left[ \boldsymbol{w}_h^k - \inf_{\boldsymbol{\mu}_h \in \Delta(\mathcal{S})} \left( \mathbb{E}_{s \sim \boldsymbol{\mu}_h}[V_{h+1}^{k,\beta,\eta}] + \beta \mathrm{KL}(\boldsymbol{\mu}_h \| \boldsymbol{\mu}_h^o) \right) \right]$$

$$+ \phi^\top \left[ \inf_{\boldsymbol{\mu}_h \in \Delta(\mathcal{S})} \left( \mathbb{E}_{s \sim \boldsymbol{\mu}_h}[V_{h+1}^{k,\beta,\eta}] + \beta \mathrm{KL}(\boldsymbol{\mu}_h \| \boldsymbol{\mu}_h^o) \right) - \inf_{\boldsymbol{\mu}_h \in \Delta(\mathcal{S})} \left( \mathbb{E}_{s \sim \boldsymbol{\mu}_h}[V_{h+1}^{\pi^k,\beta,\eta}] + \beta \mathrm{KL}(\boldsymbol{\mu}_h \| \boldsymbol{\mu}_h^o) \right) \right]$$

$$\leq 2 \cdot \Gamma_h^k + \phi^\top \left[ \inf_{\boldsymbol{\mu}_h \in \Delta(\mathcal{S})} \left( \mathbb{E}_{s \sim \boldsymbol{\mu}_h}[V_{h+1}^{k,\beta,\eta}] + \beta \mathrm{KL}(\boldsymbol{\mu}_h \| \boldsymbol{\mu}_h^o) \right) - \inf_{\boldsymbol{\mu}_h \in \Delta(\mathcal{S})} \left( \mathbb{E}_{s \sim \boldsymbol{\mu}_h}[V_{h+1}^{\pi^k,\beta,\eta}] + \beta \mathrm{KL}(\boldsymbol{\mu}_h \| \boldsymbol{\mu}_h^o) \right) \right] \tag{E.54}$$

$$\leq 2 \cdot \Gamma_h^k + \phi^\top \mathbb{E}_{s \sim \boldsymbol{\mu}_h^{w,k}} \left[ V_{h+1}^{k,\beta,\eta} - V_{h+1}^{\pi^k,\beta,\eta} \right] \tag{E.55}$$

$$\leq 2 \cdot \Gamma_h^k + \mathbb{E}_{s \sim P_h^w} \left[ V_{h+1}^{k,\beta,\eta} - V_{h+1}^{\pi^k,\beta,\eta} \right],$$

where (E.54) can be derived from Lemma E.7, and (E.55) uses the definition of the worst-case transition. Apply the above inequality recursively, we have

$$Q_1^{k,\beta,\eta} - Q_1^{\pi^k,\beta,\eta} \leq 2 \cdot \mathbb{E}_{P^{w,k},\pi^k} \left[ \sum_{h=1}^H \Gamma_h^k \right].$$

This finishes the proof. □

*Proof of Lemma E.1 in KL Setting.* The result directly follows from combining Lemma E.7 and Lemma E.8, along with applying a union bound. □

### E.3.3. PROOF OF LEMMA E.1 IN $\chi^2$ SETTING

**Lemma E.9** (Dual formulation). (Tang et al., 2025, Proposition 4.7) For the optimization problem $\inf_{\boldsymbol{\mu} \in \Delta(\mathcal{S})} \left( \mathbb{E}_{s \sim \boldsymbol{\mu}} [V_{h+1}] + \beta \chi^2(\boldsymbol{\mu} \| \boldsymbol{\mu}^o) \right)$, we have its dual formulation as follows

$$\inf_{\boldsymbol{\mu} \in \Delta(\mathcal{S})} \left( \mathbb{E}_{s \sim \boldsymbol{\mu}} [V_{h+1}] + \beta \chi^2(\boldsymbol{\mu} \| \boldsymbol{\mu}^o) \right) = \sup_{\alpha \in [V_{\min}, V_{\max}]} \left\{ \mathbb{E}_{s \sim \boldsymbol{\mu}^o} \left[ V_{h+1}(s) \right]_\alpha - \frac{1}{4\beta} \mathrm{Var}_{s \sim \boldsymbol{\mu}^o} \left[ V_{h+1}(s) \right]_\alpha \right\}.$$

**Lemma E.10** (Optimism). If we set the bonus term as follows

$$\Gamma_h^k(s,a) = c_{\chi^2} \sum_{i=1}^d \| \phi_i(s,a) \mathbb{1}_i \|_{(\Lambda_h^k)^{-1}},$$

$$\text{where} \quad c_{\chi^2} = \left( 1 + H/(2\beta) \right) H d \cdot \xi_{\chi^2},$$

$$\text{where} \quad \xi_{\chi^2} = 720 + 3\sqrt{40 \log \left( 96 K^6 H^5 (1 + H/(2\beta))^3 |\mathcal{A}|^3 / \delta \right)},$$

then for any policy $\pi$ and any $(k,h,s,a) \in [K] \times [H] \times \mathcal{S} \times \mathcal{A}$, with probability at least $1 - \delta$, we have $Q_h^{k,\beta,\eta}(s,a) \geq Q_h^{\pi,\beta,\eta}(s,a)$. Specially, by setting $\pi = \pi^*$, we have $Q_h^{k,\beta,\eta}(s,a) \geq Q_h^{\pi^*,\beta,\eta}(s,a)$.

*Proof.* We prove this by induction. First, $h = H + 1$ holds trivially since $Q_{H+1}^{k,\beta,\eta}(s,a) = 0 = Q_{H+1}^{\pi,\beta,\eta}(s,a)$.

Assume $Q_{h+1}^{k,\beta,\eta}(s,a) \geq Q_{h+1}^{\pi,\beta,\eta}(s,a)$ holds, due to the choice of $\pi_{h+1}^k$ in (4.3), we have

$$\begin{aligned} V_{h+1}^{k,\beta,\eta}(s) &= \langle Q_{h+1}^{k,\beta,\eta}(s,\cdot), \pi_{h+1}^k(\cdot|s) \rangle - \Omega_s^\eta(\pi_{h+1}^k) \\ &\geq \langle Q_{h+1}^{k,\beta,\eta}(s,\cdot), \pi_{h+1}(\cdot|s) \rangle - \Omega_s^\eta(\pi_{h+1}) \\ &\geq \langle Q_{h+1}^{\pi,\beta,\eta}(s,\cdot), \pi_{h+1}(\cdot|s) \rangle - \Omega_s^\eta(\pi_{h+1}) = V_{h+1}^{\pi,\beta,\eta}(s). \end{aligned}$$

Recall that we denote $Q_h^{k,\beta,\eta}$ as the optimistic estimation in $k$-th episode, that is,

$$Q_h^{k,\beta,\eta}(s,a) \leftarrow \min \left\{ \phi(s,a)^\top \left( \boldsymbol{\theta}_h^k + \boldsymbol{w}_h^k \right) + \Gamma_h^k(s,a), H - h + 1 \right\}. \tag{E.56}$$

If $Q_h^{k,\beta,\eta}(s,a) = H - h + 1$, then it follows immediately that

$$Q_h^{k,\beta,\eta}(s,a) = H - h + 1 \geq Q_h^{\pi,\beta,\eta}(s,a)$$

by the definition of $Q_h^{\pi,\beta,\eta}(s,a)$. Otherwise, we introduce a few auxiliary notations

$$\widetilde{\mathbb{E}}_{s \sim \boldsymbol{\mu}^o} [V_{h+1}^{k,\beta,\eta}]_{\alpha_i} = \operatorname*{argmin}_{\boldsymbol{w} \in \mathbb{R}^d} \sum_{\tau=1}^k \left( [V_{h+1}^{k,\beta,\eta}(s_{h+1}^\tau)]_{\alpha_i} - \phi(s_h^\tau, a_h^\tau)^\top \boldsymbol{w} \right)^2 + \lambda \| \boldsymbol{w} \|_2^2,$$

$$\widetilde{\mathbb{E}}_{s\sim\boldsymbol{\mu}^o}[V_{h+1}^{k,\beta,\eta}]_{\alpha_i}^2 = \underset{\boldsymbol{w}\in\mathbb{R}^d}{\operatorname{argmin}} \sum_{\tau=1}^k \left([V_{h+1}^{k,\beta,\eta}(s_{h+1}^\tau)]_{\alpha_i}^2 - \boldsymbol{\phi}(s_h^\tau, a_h^\tau)^\top \boldsymbol{w}\right)^2 + \lambda\|\boldsymbol{w}\|_2^2,$$

and denote

$$\alpha_i = \underset{\alpha\in[0,H]}{\operatorname{argmax}} \left\{ \mathbb{E}_{s\sim\boldsymbol{\mu}^o}[V_{h+1}^{k,\beta,\eta}]_\alpha + \frac{1}{4\beta}\left(\mathbb{E}_{s\sim\boldsymbol{\mu}^o}[V_{h+1}^{k,\beta,\eta}]_\alpha\right)^2 - \frac{1}{4\beta}\mathbb{E}_{s\sim\boldsymbol{\mu}^o}[V_{h+1}^{k,\beta,\eta}]_\alpha^2 \right\}.$$

Then

$$Q_h^{k,\beta,\eta} - Q_h^{\pi,\beta,\eta}$$
$$= \Gamma_h^k + \boldsymbol{\phi}^\top\left(\boldsymbol{\theta}_h^k + \boldsymbol{w}_h^k\right) - \boldsymbol{\phi}^\top\left[\boldsymbol{\theta}_h + \inf_{\mu\in\Delta(\mathcal{S})}\left(\mathbb{E}_{s\sim\mu}[V_{h+1}^{\pi,\beta,\eta}] + \beta\chi^2(\mu\|\mu^o)\right)\right]$$
$$\geq \Gamma_h^k + \boldsymbol{\phi}^\top\left(\boldsymbol{\theta}_h^k + \boldsymbol{w}_h^k\right) - \boldsymbol{\phi}^\top\left[\boldsymbol{\theta}_h + \inf_{\mu\in\Delta(\mathcal{S})}\left(\mathbb{E}_{s\sim\mu}[V_{h+1}^{k,\beta,\eta}] + \beta\chi^2(\mu\|\mu^o)\right)\right] \tag{E.57}$$
$$= \Gamma_h^k + \left\langle \boldsymbol{\phi}, \boldsymbol{\theta}_h^k + \sup_{\alpha\in[0,H]}\left\{\widehat{\mathbb{E}}_{s\sim\boldsymbol{\mu}^o}[V_{h+1}^{k,\beta,\eta}]_\alpha + \frac{1}{4\beta}\left(\widehat{\mathbb{E}}_{s\sim\boldsymbol{\mu}^o}[V_{h+1}^{k,\beta,\eta}]_\alpha\right)^2 - \frac{1}{4\beta}\widehat{\mathbb{E}}_{s\sim\boldsymbol{\mu}^o}[V_{h+1}^{k,\beta,\eta}]_\alpha^2\right\}\right.$$
$$\left. - \boldsymbol{\theta}_h - \sup_{\alpha\in[0,H]}\left\{\mathbb{E}_{s\sim\boldsymbol{\mu}^o}[V_{h+1}^{k,\beta,\eta}]_\alpha + \frac{1}{4\beta}\left(\mathbb{E}_{s\sim\boldsymbol{\mu}^o}[V_{h+1}^{k,\beta,\eta}]_\alpha\right)^2 - \frac{1}{4\beta}\mathbb{E}_{s\sim\boldsymbol{\mu}^o}[V_{h+1}^{k,\beta,\eta}]_\alpha^2\right\}\right\rangle \tag{E.58}$$
$$\geq \Gamma_h^k + \left\langle \boldsymbol{\phi}, \boldsymbol{\theta}_h^k + \left\{\widehat{\mathbb{E}}_{s\sim\boldsymbol{\mu}^o}[V_{h+1}^{k,\beta,\eta}]_{\alpha_i} + \frac{1}{4\beta}\left(\widehat{\mathbb{E}}_{s\sim\boldsymbol{\mu}^o}[V_{h+1}^{k,\beta,\eta}]_{\alpha_i}\right)^2 - \frac{1}{4\beta}\widehat{\mathbb{E}}_{s\sim\boldsymbol{\mu}^o}[V_{h+1}^{k,\beta,\eta}]_{\alpha_i}^2\right\}\right.$$
$$\left. - \boldsymbol{\theta}_h - \left\{\mathbb{E}_{s\sim\boldsymbol{\mu}^o}[V_{h+1}^{k,\beta,\eta}]_{\alpha_i} + \frac{1}{4\beta}\left(\mathbb{E}_{s\sim\boldsymbol{\mu}^o}[V_{h+1}^{k,\beta,\eta}]_{\alpha_i}\right)^2 - \frac{1}{4\beta}\mathbb{E}_{s\sim\boldsymbol{\mu}^o}[V_{h+1}^{k,\beta,\eta}]_{\alpha_i}^2\right\}\right\rangle \tag{E.59}$$
$$\geq \Gamma_h^k - \left\langle \boldsymbol{\phi}, \left|\boldsymbol{\theta}_h^k - \boldsymbol{\theta}_h\right| + \left|\frac{1}{4\beta}\widehat{\mathbb{E}}_{s\sim\boldsymbol{\mu}^o}[V_{h+1}^{k,\beta,\eta}]_{\alpha_i}^2 - \frac{1}{4\beta}\mathbb{E}_{s\sim\boldsymbol{\mu}^o}[V_{h+1}^{k,\beta,\eta}]_{\alpha_i}^2\right|\right.$$
$$\left. + \left|\widehat{\mathbb{E}}_{s\sim\boldsymbol{\mu}^o}[V_{h+1}^{k,\beta,\eta}]_{\alpha_i} + \frac{1}{4\beta}\left(\widehat{\mathbb{E}}_{s\sim\boldsymbol{\mu}^o}[V_{h+1}^{k,\beta,\eta}]_{\alpha_i}\right)^2 - \mathbb{E}_{s\sim\boldsymbol{\mu}^o}[V_{h+1}^{k,\beta,\eta}]_{\alpha_i} - \frac{1}{4\beta}\left(\mathbb{E}_{s\sim\boldsymbol{\mu}^o}[V_{h+1}^{k,\beta,\eta}]_{\alpha_i}\right)^2\right|\right\rangle$$
$$= \Gamma_h^k - \sum_{i=1}^d \phi_i\mathbb{1}_i^\top\left|\boldsymbol{\theta}_h^k - \boldsymbol{\theta}_h\right| - \frac{1}{4\beta}\sum_{i=1}^d \phi_i\mathbb{1}_i^\top\left|\widehat{\mathbb{E}}_{s\sim\boldsymbol{\mu}^o}[V_{h+1}^{k,\beta,\eta}]_{\alpha_i}^2 - \mathbb{E}_{s\sim\boldsymbol{\mu}^o}[V_{h+1}^{k,\beta,\eta}]_{\alpha_i}^2\right|$$
$$- \sum_{i=1}^d \phi_i\mathbb{1}_i^\top\left|\widehat{\mathbb{E}}_{s\sim\boldsymbol{\mu}^o}[V_{h+1}^{k,\beta,\eta}]_{\alpha_i} - \mathbb{E}_{s\sim\boldsymbol{\mu}^o}[V_{h+1}^{k,\beta,\eta}]_{\alpha_i}\right| \cdot \left|\frac{1}{4\beta}\left(\widehat{\mathbb{E}}_{s\sim\boldsymbol{\mu}^o}[V_{h+1}^{k,\beta,\eta}]_{\alpha_i} + \mathbb{E}_{s\sim\boldsymbol{\mu}^o}[V_{h+1}^{k,\beta,\eta}]_{\alpha_i}\right) + 1\right|$$
$$= \Gamma_h^k - \sum_{i=1}^d \phi_i\mathbb{1}_i^\top\left|\boldsymbol{\theta}_h^k - \boldsymbol{\theta}_h\right| - \frac{1}{4\beta}\sum_{i=1}^d \phi_i\mathbb{1}_i^\top\left|\widetilde{\mathbb{E}}_{s\sim\boldsymbol{\mu}^o}[V_{h+1}^{k,\beta,\eta}]_{\alpha_i}^2 - \mathbb{E}_{s\sim\boldsymbol{\mu}^o}[V_{h+1}^{k,\beta,\eta}]_{\alpha_i}^2\right|$$
$$\times \underbrace{\left|\frac{\widehat{\mathbb{E}}_{s\sim\boldsymbol{\mu}^o}[V_{h+1}^{k,\beta,\eta}]_{\alpha_i}^2 - \mathbb{E}_{s\sim\boldsymbol{\mu}^o}[V_{h+1}^{k,\beta,\eta}]_{\alpha_i}^2}{\widetilde{\mathbb{E}}_{s\sim\boldsymbol{\mu}^o}[V_{h+1}^{k,\beta,\eta}]_{\alpha_i}^2 - \mathbb{E}_{s\sim\boldsymbol{\mu}^o}[V_{h+1}^{k,\beta,\eta}]_{\alpha_i}^2}\right|}_{(i)} - \sum_{i=1}^d \phi_i\mathbb{1}_i^\top\left|\widetilde{\mathbb{E}}_{s\sim\boldsymbol{\mu}^o}[V_{h+1}^{k,\beta,\eta}]_{\alpha_i} - \mathbb{E}_{s\sim\boldsymbol{\mu}^o}[V_{h+1}^{k,\beta,\eta}]_{\alpha_i}\right|$$
$$\times \underbrace{\left|\left[\frac{1}{4\beta}\left(\widehat{\mathbb{E}}_{s\sim\boldsymbol{\mu}^o}[V_{h+1}^{k,\beta,\eta}]_{\alpha_i} + \mathbb{E}_{s\sim\boldsymbol{\mu}^o}[V_{h+1}^{k,\beta,\eta}]_{\alpha_i}\right) + 1\right] \cdot \frac{\widehat{\mathbb{E}}_{s\sim\boldsymbol{\mu}^o}[V_{h+1}^{k,\beta,\eta}]_{\alpha_i} - \mathbb{E}_{s\sim\boldsymbol{\mu}^o}[V_{h+1}^{k,\beta,\eta}]_{\alpha_i}}{\widetilde{\mathbb{E}}_{s\sim\boldsymbol{\mu}^o}[V_{h+1}^{k,\beta,\eta}]_{\alpha_i} - \mathbb{E}_{s\sim\boldsymbol{\mu}^o}[V_{h+1}^{k,\beta,\eta}]_{\alpha_i}}\right|}_{(ii)} \tag{E.60}$$
$$\geq \Gamma_h^k - \sum_{i=1}^d \underbrace{\phi_i\mathbb{1}_i^\top\left|\boldsymbol{\theta}_h^k - \boldsymbol{\theta}_h\right|}_{(iii)} - \frac{1}{4\beta}\sum_{i=1}^d \phi_i\mathbb{1}_i^\top\underbrace{\left|\widetilde{\mathbb{E}}_{s\sim\boldsymbol{\mu}^o}[V_{h+1}^{k,\beta,\eta}]_{\alpha_i}^2 - \mathbb{E}_{s\sim\boldsymbol{\mu}^o}[V_{h+1}^{k,\beta,\eta}]_{\alpha_i}^2\right|}_{(v)}$$
$$- \left(1 + \frac{H}{2\beta}\right)\sum_{i=1}^d \phi_i\mathbb{1}_i^\top\underbrace{\left|\widetilde{\mathbb{E}}_{s\sim\boldsymbol{\mu}^o}[V_{h+1}^{k,\beta,\eta}]_{\alpha_i} - \mathbb{E}_{s\sim\boldsymbol{\mu}^o}[V_{h+1}^{k,\beta,\eta}]_{\alpha_i}\right|}_{(iv)}, \tag{E.61}$$

where (E.57) follows from the induction assumption, (E.58) is obtained from the algorithm update formulation and the dual formulation, and (E.59) comes from our choice of $\alpha_i$. To establish in (E.60) that (i) $\leq 1$ and (ii) $\leq 1 + H/(2\beta)$ and hence that (E.61) holds, we analyze the value of $\widetilde{\mathbb{E}}_{s\sim\boldsymbol{\mu}^\circ}[V_{h+1}^{k,\beta,\eta}]_{\alpha_i}$ or $\widetilde{\mathbb{E}}_{s\sim\boldsymbol{\mu}^\circ}[V_{h+1}^{k,\beta,\eta}]_{\alpha_i}^2$ under three cases:

$$\begin{cases} \text{(i)} \leq 1 & \widetilde{\mathbb{E}}_{s\sim\boldsymbol{\mu}^\circ}[V_{h+1}^{k,\beta,\eta}]_{\alpha_i}^2 < 0 \\ \text{(i)} \leq 1 & \widetilde{\mathbb{E}}_{s\sim\boldsymbol{\mu}^\circ}[V_{h+1}^{k,\beta,\eta}]_{\alpha_i}^2 \in [0, H] \\ \text{(i)} \leq 1 & \widetilde{\mathbb{E}}_{s\sim\boldsymbol{\mu}^\circ}[V_{h+1}^{k,\beta,\eta}]_{\alpha_i}^2 > H, \end{cases} \qquad \begin{cases} \text{(ii)} \leq 1 + H/(4\beta) & \widetilde{\mathbb{E}}_{s\sim\boldsymbol{\mu}^\circ}[V_{h+1}^{k,\beta,\eta}]_{\alpha_i} < 0 \\ \text{(ii)} \leq 1 + H/(2\beta) & \widetilde{\mathbb{E}}_{s\sim\boldsymbol{\mu}^\circ}[V_{h+1}^{k,\beta,\eta}]_{\alpha_i} \in [0, H] \\ \text{(ii)} \leq 1 + H/(2\beta) & \widetilde{\mathbb{E}}_{s\sim\boldsymbol{\mu}^\circ}[V_{h+1}^{k,\beta,\eta}]_{\alpha_i} > H. \end{cases}$$

For term (iii) in (E.61), we have

$$\left| \phi_i \mathbb{1}_i^\top (\boldsymbol{\theta}_h^k - \boldsymbol{\theta}_h) \right|$$

$$= \left| \phi_i \mathbb{1}_i^\top (\Lambda_h^k)^{-1} \left( \sum_{\tau=1}^{k-1} \phi(s_h^\tau, a_h^\tau) \cdot r_h^\tau \right) \right.$$

$$\left. - \phi_i \mathbb{1}_i^\top (\Lambda_h^k)^{-1} \left( \sum_{\tau=1}^{k-1} \phi(s_h^\tau, a_h^\tau)\phi(s_h^\tau, a_h^\tau)^\top + \lambda \cdot I \right) \cdot \boldsymbol{\theta}_h \right|$$

$$= \lambda \left| \phi_i \mathbb{1}_i^\top (\Lambda_h^k)^{-1} \cdot \boldsymbol{\theta}_h \right| \tag{E.62}$$

$$\leq \lambda \|\phi_i \mathbb{1}_i\|_{(\Lambda_h^k)^{-1}} \cdot \|\boldsymbol{\theta}_h\|_{(\Lambda_h^k)^{-1}} \tag{E.63}$$

$$\leq \lambda \|\phi_i \mathbb{1}_i\|_{(\Lambda_h^k)^{-1}} \cdot \lambda_{\max}\left( (\Lambda_h^k)^{-1} \right)^{\frac{1}{2}} \|\boldsymbol{\theta}_h\|_2 \tag{E.64}$$

$$\leq \lambda^{\frac{1}{2}} \sqrt{d} \cdot \|\phi_i \mathbb{1}_i\|_{(\Lambda_h^k)^{-1}},$$

where (E.62) is obtained from the algorithm update formulation and the definition of $\Lambda_h^k$, (E.63) follows from the Cauchy-Schwartz inequality, (E.64) is because $\|x\|_A \leq \sqrt{\lambda_{\max}(A)}\|x\|_2$.

For term (iv) in (E.61), we decompose it as follows.

$$\left| \phi_i \mathbb{1}_i^\top \left( \widetilde{\mathbb{E}}_{s\sim\boldsymbol{\mu}^\circ}[V_{h+1}^{k,\beta,\eta}]_{\alpha_i} - \mathbb{E}_{s\sim\boldsymbol{\mu}^\circ}[V_{h+1}^{k,\beta,\eta}]_{\alpha_i} \right) \right|$$

$$= \left| \phi_i \mathbb{1}_i^\top (\Lambda_h^k)^{-1} \left( \sum_{\tau=1}^{k-1} \phi(s_h^\tau, a_h^\tau) \cdot [V_{h+1}^{k,\beta,\eta}(s_{h+1}^\tau)]_{\alpha_i} \right) \right.$$

$$\left. - \phi_i \mathbb{1}_i^\top (\Lambda_h^k)^{-1} \left( \sum_{\tau=1}^{k-1} \phi(s_h^\tau, a_h^\tau)\phi(s_h^\tau, a_h^\tau)^\top + \lambda \cdot I \right) \cdot \mathbb{E}_{s\sim\boldsymbol{\mu}^\circ}[V_{h+1}^{k,\beta,\eta}(s)]_{\alpha_i} \right| \tag{E.65}$$

$$= \left| \phi_i \mathbb{1}_i^\top (\Lambda_h^k)^{-1} \left( \sum_{\tau=1}^{k-1} \phi(s_h^\tau, a_h^\tau) \cdot \text{err}_h^\tau\left( [V_{h+1}^{k,\beta,\eta}(s)]_{\alpha_i} \right) \right) \right.$$

$$\left. + \lambda \phi_i \mathbb{1}_i^\top (\Lambda_h^k)^{-1} \mathbb{E}_{s\sim\boldsymbol{\mu}^\circ}[V_{h+1}^{k,\beta,\eta}(s)]_{\alpha_i} \right| \tag{E.66}$$

$$\leq \left| \phi_i \mathbb{1}_i^\top (\Lambda_h^k)^{-1} \left( \sum_{\tau=1}^{k-1} \phi(s_h^\tau, a_h^\tau) \cdot \text{err}_h^\tau\left( [V_{h+1}^{k,\beta,\eta}(s)]_{\alpha_i} \right) \right) \right|$$

$$+ \lambda \left| \phi_i \mathbb{1}_i^\top (\Lambda_h^k)^{-1} \mathbb{E}_{s\sim\boldsymbol{\mu}^\circ}[V_{h+1}^{k,\beta,\eta}(s)]_{\alpha_i} \right|$$

$$\leq \|\phi_i \mathbb{1}_i\|_{(\Lambda_h^k)^{-1}} \cdot \underbrace{\left\| \sum_{\tau=1}^{k-1} \phi(s_h^\tau, a_h^\tau) \cdot \text{err}_h^\tau\left( [V_{h+1}^{k,\beta,\eta}(s)]_{\alpha_i} \right) \right\|_{(\Lambda_h^k)^{-1}}}_{\text{(vi)}}$$

$$+ \lambda \|\phi_i \mathbb{1}_i\|_{(\Lambda_h^k)^{-1}} \cdot \underbrace{\left\| \mathbb{E}_{s\sim\boldsymbol{\mu}^\circ}[V_{h+1}^{k,\beta,\eta}(s)]_{\alpha_i} \right\|_{(\Lambda_h^k)^{-1}}}_{\text{(vii)}}, \tag{E.67}$$

where (E.65) is obtained from the algorithm update formulation and the definition of $\Lambda_h^k$, (E.66) is from the definition in (E.23), (E.67) follows from the Cauchy-Schwartz inequality.

For term (v) in (E.61), we decompose it as follows.

$$
\left| \phi_i \mathbb{1}_i^\top \left( \widetilde{\mathbb{E}}_{s\sim\boldsymbol{\mu}^o}[V_{h+1}^{k,\beta,\eta}]_{\alpha_i}^2 - \mathbb{E}_{s\sim\boldsymbol{\mu}^o}[V_{h+1}^{k,\beta,\eta}]_{\alpha_i}^2 \right) \right|
$$

$$
= \left| \phi_i \mathbb{1}_i^\top \left( \Lambda_h^k \right)^{-1} \left( \sum_{\tau=1}^{k-1} \phi(s_h^\tau, a_h^\tau) \cdot [V_{h+1}^{k,\beta,\eta}(s_{h+1}^\tau)]_{\alpha_i}^2 \right) \right.
$$
$$
\left. - \phi_i \mathbb{1}_i^\top \left( \Lambda_h^k \right)^{-1} \left( \sum_{\tau=1}^{k-1} \phi(s_h^\tau, a_h^\tau) \phi(s_h^\tau, a_h^\tau)^\top + \lambda \cdot I \right) \cdot \mathbb{E}_{s\sim\boldsymbol{\mu}^o}[V_{h+1}^{k,\beta,\eta}(s)]_{\alpha_i}^2 \right| \tag{E.68}
$$

$$
= \left| \phi_i \mathbb{1}_i^\top \left( \Lambda_h^k \right)^{-1} \left( \sum_{\tau=1}^{k-1} \phi(s_h^\tau, a_h^\tau) \cdot \mathrm{err}_h^\tau \left( [V_{h+1}^{k,\beta,\eta}(s)]_{\alpha_i}^2 \right) \right) \right.
$$
$$
\left. + \lambda \phi_i \mathbb{1}_i^\top \left( \Lambda_h^k \right)^{-1} \mathbb{E}_{s\sim\boldsymbol{\mu}^o}[V_{h+1}^{k,\beta,\eta}(s)]_{\alpha_i}^2 \right| \tag{E.69}
$$

$$
\leq \left| \phi_i \mathbb{1}_i^\top \left( \Lambda_h^k \right)^{-1} \left( \sum_{\tau=1}^{k-1} \phi(s_h^\tau, a_h^\tau) \cdot \mathrm{err}_h^\tau \left( [V_{h+1}^{k,\beta,\eta}(s)]_{\alpha_i}^2 \right) \right) \right|
$$
$$
+ \lambda \left| \phi_i \mathbb{1}_i^\top \left( \Lambda_h^k \right)^{-1} \mathbb{E}_{s\sim\boldsymbol{\mu}^o}[V_{h+1}^{k,\beta,\eta}(s)]_{\alpha_i}^2 \right|
$$

$$
\leq \| \phi_i \mathbb{1}_i \|_{(\Lambda_h^k)^{-1}} \cdot \underbrace{\left\| \sum_{\tau=1}^{k-1} \phi(s_h^\tau, a_h^\tau) \cdot \mathrm{err}_h^\tau \left( [V_{h+1}^{k,\beta,\eta}(s)]_{\alpha_i}^2 \right) \right\|_{(\Lambda_h^k)^{-1}}}_{\text{(viii)}}
$$
$$
+ \lambda \| \phi_i \mathbb{1}_i \|_{(\Lambda_h^k)^{-1}} \cdot \underbrace{\left\| \mathbb{E}_{s\sim\boldsymbol{\mu}^o}[V_{h+1}^{k,\beta,\eta}(s)]_{\alpha_i}^2 \right\|_{(\Lambda_h^k)^{-1}}}_{\text{(ix)}}, \tag{E.70}
$$

where (E.68) is obtained from the algorithm update formulation and the definition of $\Lambda_h^k$, (E.69) is from the definition in (E.23), (E.70) follows from the Cauchy-Schwartz inequality.

For term (vii) in (E.67) and For term (ix) in (E.70), since $V_{h+1}^{k,\beta,\eta} \leq H$, we have

$$
\left\| \mathbb{E}_{s\sim\boldsymbol{\mu}^o}[V_{h+1}^{k,\beta,\eta}]_{\alpha_i} \right\|_{(\Lambda_h^k)^{-1}} \leq \lambda_{\max}\left( (\Lambda_h^k)^{-1} \right)^{\frac{1}{2}} \left\| \mathbb{E}_{s\sim\boldsymbol{\mu}^o}[V_{h+1}^{k,\beta,\eta}]_{\alpha_i} \right\|_2 \leq \lambda^{-\frac{1}{2}} H \sqrt{d},
$$
$$
\left\| \mathbb{E}_{s\sim\boldsymbol{\mu}^o}[V_{h+1}^{k,\beta,\eta}]_{\alpha_i}^2 \right\|_{(\Lambda_h^k)^{-1}} \leq \lambda_{\max}\left( (\Lambda_h^k)^{-1} \right)^{\frac{1}{2}} \left\| \mathbb{E}_{s\sim\boldsymbol{\mu}^o}[V_{h+1}^{k,\beta,\eta}]_{\alpha_i}^2 \right\|_2 \leq \lambda^{-\frac{1}{2}} H^2 \sqrt{d}.
$$

Before we continue to bound term (vi) and (viii), we prove a auxiliary result first. That is, $\| \boldsymbol{w}_h^k \|_2 \leq \left( H + H^2/(2\beta) \right) \sqrt{d}$. From the algorithm update formulation, we have

$$
\| \boldsymbol{w}_h^k \|_2 = \left\| \sup_{\alpha\in[0,H]} \left\{ \widehat{\mathbb{E}}_{s\sim\boldsymbol{\mu}^o}[V_{h+1}^{k,\beta,\eta}]_\alpha + \frac{1}{4\beta} \left( \widehat{\mathbb{E}}_{s\sim\boldsymbol{\mu}^o}[V_{h+1}^{k,\beta,\eta}]_\alpha \right)^2 - \frac{1}{4\beta} \widehat{\mathbb{E}}_{s\sim\boldsymbol{\mu}^o}[V_{h+1}^{k,\beta,\eta}]_\alpha^2 \right\} \right\|_2
$$
$$
\leq \left\| \mathbb{1}\left( H + \frac{H^2}{2\beta} \right) \right\|_2 = \left( H + \frac{H^2}{2\beta} \right) \sqrt{d}.
$$

Now we are ready to bound term (vi) and (viii). Let $\mathcal{V}_h$ denote a class of functions mapping from $\mathcal{S}$ to $\mathbb{R}$ with the following from

$$
V(\cdot) = \min \left\{ \max_{a\in\mathcal{A}} \left\{ \phi(\cdot, a)^\top \boldsymbol{w} + \beta \sum_{i=1}^d \| \phi_i(\cdot, a) \mathbb{1}_i \|_{\Lambda^{-1}} \right\}, H - h + 1 \right\},
$$

where the parameters $(w, \beta, \Lambda)$ satisfy $\|w\| \le L$, $\beta \in [0, B]$, $\lambda_{\min} \ge \lambda$. Here we set $L = \left(H + H^2/(2\beta)\right)\sqrt{d}$ and $B = \left(1 + H/(2\beta)\right)Hd \cdot \xi_{\chi^2}$, where $\xi_{\chi^2} = 720 + 3\sqrt{40 \log\left(96 K^6 H^5 (1 + H/(2\beta))^3 |\mathcal{A}|^3/\delta\right)}$.

Let $\mathcal{N}_h(\epsilon; \mathcal{V}_h)$ be the minimum $\epsilon$-cover of $\mathcal{V}_h$, $\mathcal{N}_h(\epsilon; [0, H])$ be the minimum $\epsilon$-cover of $[0, H]$. Therefore, for any $V \in \mathcal{V}_h$ and $\alpha \in [0, H]$, there exists a function $V_\epsilon \in \mathcal{N}_h(\epsilon; \mathcal{V}_h)$ and $\alpha_\epsilon \in \mathcal{N}_h(\epsilon; [0, H])$ such that

$$\sup_{s \in \mathcal{S}} \left| V(s) - V_\epsilon(s) \right| \le \epsilon, \quad \left| \min_{s'} V_{h+1}^{k,\beta,\eta}(s') + \beta - \alpha_\epsilon \right| \le \epsilon.$$

Then we have

$$
\begin{aligned}
(\text{vi})^2 &= \left\| \sum_{\tau=1}^{k-1} \phi(s_h^\tau, a_h^\tau) \cdot \text{err}_h^\tau\left(\left[V_{h+1}^{k,\beta,\eta}\right]_{\alpha_i}\right) \right\|_{(\Lambda_h^k)^{-1}}^2 \\
&\le 2 \left\| \sum_{\tau=1}^{k-1} \phi(s_h^\tau, a_h^\tau) \cdot \text{err}_h^\tau\left(\left[V_{h+1}^{k,\beta,\eta}\right]_{\alpha_\epsilon}\right) \right\|_{(\Lambda_h^k)^{-1}}^2 \\
&\quad + 2 \left\| \sum_{\tau=1}^{k-1} \phi(s_h^\tau, a_h^\tau) \cdot \text{err}_h^\tau\left(\left[V_{h+1}^{k,\beta,\eta}\right]_{\alpha_i} - \left[V_{h+1}^{k,\beta,\eta}\right]_{\alpha_\epsilon}\right) \right\|_{(\Lambda_h^k)^{-1}}^2 \qquad (\text{E.71}) \\
&\le 4 \left\| \sum_{\tau=1}^{k-1} \phi(s_h^\tau, a_h^\tau) \cdot \text{err}_h^\tau\left(\left[V_\epsilon\right]_{\alpha_\epsilon}\right) \right\|_{(\Lambda_h^k)^{-1}}^2 + 4 \left\| \sum_{\tau=1}^{k-1} \phi(s_h^\tau, a_h^\tau) \cdot \text{err}_h^\tau\left(\left[V_{h+1}^{k,\beta,\eta}\right]_{\alpha_\epsilon} - \left[V_\epsilon\right]_{\alpha_\epsilon}\right) \right\|_{(\Lambda_h^k)^{-1}}^2 \\
&\quad + 2(2\epsilon)^2 k^2/\lambda \qquad (\text{E.72}) \\
&\le 4 \left\| \sum_{\tau=1}^{k-1} \phi(s_h^\tau, a_h^\tau) \cdot \text{err}_h^\tau\left(\left[V_\epsilon\right]_{\alpha_\epsilon}\right) \right\|_{(\Lambda_h^k)^{-1}}^2 + 4(2\epsilon)^2 k^2/\lambda + 2(2\epsilon)^2 k^2/\lambda \\
&\le 4 \left\| \sum_{\tau=1}^{k-1} \phi(s_h^\tau, a_h^\tau) \cdot \text{err}_h^\tau\left(\left[V_\epsilon\right]_{\alpha_\epsilon}\right) \right\|_{(\Lambda_h^k)^{-1}}^2 + 24\epsilon^2 k^2/\lambda \\
&\le 4H^2 \left( d\log(1 + k/\lambda) + 2\log \frac{KH|\mathcal{N}_h(\epsilon; \mathcal{V}_h)| \cdot |\mathcal{N}_h(\epsilon; [0, H])|}{\delta} \right) + 24\epsilon^2 k^2/\lambda \qquad (\text{E.73}) \\
&\le 4H^2 \left( d\log(1 + k/\lambda) + 2d\log(1 + 4L|\mathcal{A}|/\epsilon) + 2d^2 \log\left(1 + 8d^{1/2}B^2|\mathcal{A}|^2/(\lambda\epsilon^2)\right) \right. \\
&\quad \left. + 2\log(3H/\epsilon) + 2\log(KH/\delta) \right) + 24\epsilon^2 k^2/\lambda \qquad (\text{E.74}) \\
&\le 4H^2 \left( 2d\log(k) + 4d\log\left(4k(1 + H/(2\beta))|\mathcal{A}|/(d^{1/2})\right) + 4d^2 \log\left(8k^2 B^2|\mathcal{A}|^2/(H^2 d^{3/2})\right) \right. \\
&\quad \left. + 2\log(3k/d) + 2\log(KH/\delta) \right) + 24H^2 d^2 \qquad (\text{E.75}) \\
&\le 16H^2 d^2 \log\left(96 K^6 B^2(1 + H/(2\beta))|\mathcal{A}|^3/(H d^3 \delta)\right) + 24H^2 d^2,
\end{aligned}
$$

where (E.71) and (E.72) is because $\|a + b\|_A^2 \le 2\|a\|_A^2 + 2\|b\|_A^2$, (E.73) is from Lemma F.7 together with a union bound over all $(k, h) \in [K] \times [H]$, (E.74) makes use of Lemmas E.2 and F.5, we set $\epsilon = Hd/k$ and $\lambda = 1$ and apply the inequality $\log(1 + x) \le 2\log x$ for $x \ge 2$ in (E.75).

And

$$
\begin{aligned}
(\text{viii})^2 &= \left\| \sum_{\tau=1}^{k-1} \phi(s_h^\tau, a_h^\tau) \cdot \text{err}_h^\tau\left(\left[V_{h+1}^{k,\beta,\eta}\right]_{\alpha_i}^2\right) \right\|_{(\Lambda_h^k)^{-1}}^2 \\
&\le 2 \left\| \sum_{\tau=1}^{k-1} \phi(s_h^\tau, a_h^\tau) \cdot \text{err}_h^\tau\left(\left[V_{h+1}^{k,\beta,\eta}\right]_{\alpha_\epsilon}^2\right) \right\|_{(\Lambda_h^k)^{-1}}^2 \\
&\quad + 2 \left\| \sum_{\tau=1}^{k-1} \phi(s_h^\tau, a_h^\tau) \cdot \text{err}_h^\tau\left(\left[V_{h+1}^{k,\beta,\eta}\right]_{\alpha_i}^2 - \left[V_{h+1}^{k,\beta,\eta}\right]_{\alpha_\epsilon}^2\right) \right\|_{(\Lambda_h^k)^{-1}}^2 \qquad (\text{E.76})
\end{aligned}
$$

$$
\leq 4\left\|\sum_{\tau=1}^{k-1} \phi(s_h^\tau, a_h^\tau) \cdot \mathrm{err}_h^\tau\left([V_\epsilon]_{\alpha_\epsilon}^2\right)\right\|_{(\Lambda_h^k)^{-1}}^2 + 4\left\|\sum_{\tau=1}^{k-1} \phi(s_h^\tau, a_h^\tau) \cdot \mathrm{err}_h^\tau\left([V_{h+1}^{k,\beta,\eta}]_{\alpha_\epsilon}^2 - [V_\epsilon]_{\alpha_\epsilon}^2\right)\right\|_{(\Lambda_h^k)^{-1}}^2
$$
$$
+ 2(2H\epsilon)^2 k^2/\lambda \tag{E.77}
$$

$$
\leq 4\left\|\sum_{\tau=1}^{k-1} \phi(s_h^\tau, a_h^\tau) \cdot \mathrm{err}_h^\tau\left([V_\epsilon]_{\alpha_\epsilon}^2\right)\right\|_{(\Lambda_h^k)^{-1}}^2 + 4(2H\epsilon)^2 k^2/\lambda + 2(2H\epsilon)^2 k^2/\lambda
$$

$$
\leq 4\left\|\sum_{\tau=1}^{k-1} \phi(s_h^\tau, a_h^\tau) \cdot \mathrm{err}_h^\tau\left([V_\epsilon]_{\alpha_\epsilon}^2\right)\right\|_{(\Lambda_h^k)^{-1}}^2 + 24H^2\epsilon^2 k^2/\lambda
$$

$$
\leq 4H^2\left(d\log(1+k/\lambda) + 2\log\frac{KH|\mathcal{N}_h(\epsilon;\mathcal{V}_h)|\cdot|\mathcal{N}_h(\epsilon;[0,H])|}{\delta}\right) + 24H^2\epsilon^2 k^2/\lambda \tag{E.78}
$$

$$
\leq 4H^2\Big(d\log(1+k/\lambda) + 2d\log(1+4L|\mathcal{A}|/\epsilon) + 2d^2\log\left(1+8d^{1/2}B^2|\mathcal{A}|^2/(\lambda\epsilon^2)\right)
$$
$$
+ 2\log(3H/\epsilon) + 2\log(KH/\delta)\Big) + 24H^2\epsilon^2 k^2/\lambda \tag{E.79}
$$

$$
\leq 4H^2\Big(2d\log(k) + 4d\log\left(4kH(1+H/(2\beta))|\mathcal{A}|/(d^{1/2})\right) + 4d^2\log\left(8k^2B^2|\mathcal{A}|^2/(d^{3/2})\right)
$$
$$
+ 2\log(3kH/d) + 2\log(KH/\delta)\Big) + 24H^2d^2 \tag{E.80}
$$

$$
\leq 16H^2d^2\log\left(96K^6H^3B^2(1+H/(2\beta))|\mathcal{A}|^3/(d^3\delta)\right) + 24H^2d^2,
$$

where (E.76) and (E.77) is because $\|a+b\|_A^2 \leq 2\|a\|_A^2 + 2\|b\|_A^2$, (E.78) is from Lemma F.7 together with a union bound over all $(k,h) \in [K] \times [H]$, (E.79) makes use of Lemmas E.2 and F.5, we set $\epsilon = d/k$ and $\lambda = 1$ and apply the inequality $\log(1+x) \leq 2\log x$ for $x \geq 2$ in (E.80).

Combining everything together, recall that we set $\lambda = 1$, we have

$$
\mathrm{(iii)} + \frac{1}{4\beta}\mathrm{(v)} + \left(1 + \frac{H}{2\beta}\right)\mathrm{(iv)}
$$
$$
\leq \sqrt{d}\cdot\|\phi_i\mathbb{1}_i\|_{(\Lambda_h^k)^{-1}}
$$
$$
+ \frac{1}{4\beta}\Big(\sqrt{16H^2d^2\log\left(96K^6H^3B^2(1+H/(2\beta))|\mathcal{A}|^3/(d^3\delta)\right) + 24H^2d^2} + H^2\sqrt{d}\Big)\cdot\|\phi_i\mathbb{1}_i\|_{(\Lambda_h^k)^{-1}}
$$
$$
+ \left(1+\frac{H}{2\beta}\right)\Big(\sqrt{16H^2d^2\log\left(96K^6B^2(1+H/(2\beta))|\mathcal{A}|^3/(Hd^3\delta)\right) + 24H^2d^2} + H\sqrt{d}\Big)\cdot\|\phi_i\mathbb{1}_i\|_{(\Lambda_h^k)^{-1}}
$$
$$
\leq 3\left(1+\frac{H}{2\beta}\right)Hd\sqrt{40\log\left(96K^6H^3B^2(1+H/(2\beta))|\mathcal{A}|^3/(d^3\delta)\right)}\cdot\|\phi_i\mathbb{1}_i\|_{(\Lambda_h^k)^{-1}}.
$$

With our choice of $B$, and noting that $x \geq \max\left\{(2B+\sqrt{A})^2, 1\right\}$ guarantees $x \geq A + B\log x$, we conclude the proof. $\square$

**Lemma E.11.** *If we set the bonus term to be the same as in Lemma E.10, then for any $(k,s,a) \in [K] \times \mathcal{S} \times \mathcal{A}$, with probability at least $1-\delta$, the sum of estimation errors can be bounded as*

$$
Q_1^{k,\beta,\eta}(s,a) - Q_1^{\pi^k,\beta,\eta}(s,a) \leq 2\cdot\mathbb{E}_{P^{w,k},\pi^k}\left[\sum_{h=1}^H \Gamma_h^k(s_h^k, a_h^k)\right],
$$

*where $P_h^{w,k}$ is defined by $P_h^{w,k}(s,a) = \phi(s,a)^\top\cdot\boldsymbol{\mu}_h^{w,k}$.*

*Proof.* From the definition, We have that

$$
Q_h^{k,\beta,\eta} - Q_h^{\pi^k,\beta,\eta}
$$
$$
= \Gamma_h^k + \phi^\top\left(\boldsymbol{\theta}_h^k + \boldsymbol{w}_h^k\right) - \phi^\top\left[\boldsymbol{\theta}_h + \inf_{\boldsymbol{\mu}_h\in\Delta(\mathcal{S})}\left(\mathbb{E}_{s\sim\boldsymbol{\mu}_h}[V_{h+1}^{\pi^k,\beta,\eta}] + \beta\chi^2(\boldsymbol{\mu}_h\|\boldsymbol{\mu}_h^o)\right)\right]
$$

$$= \Gamma_h^k + \phi^\top(\theta_h^k - \theta_h) + \phi^\top\Big[w_h^k - \inf_{\mu_h \in \Delta(\mathcal{S})} \big(\mathbb{E}_{s \sim \mu_h}[V_{h+1}^{k,\beta,\eta}] + \beta\chi^2(\mu_h \| \mu_h^o)\big)\Big]$$

$$+ \phi^\top\Big[\inf_{\mu_h \in \Delta(\mathcal{S})} \big(\mathbb{E}_{s \sim \mu_h}[V_{h+1}^{k,\beta,\eta}] + \beta\chi^2(\mu_h \| \mu_h^o)\big) - \inf_{\mu_h \in \Delta(\mathcal{S})} \big(\mathbb{E}_{s \sim \mu_h}[V_{h+1}^{\pi^k,\beta,\eta}] + \beta\chi^2(\mu_h \| \mu_h^o)\big)\Big]$$

$$\leq 2 \cdot \Gamma_h^k + \phi^\top\Big[\inf_{\mu_h \in \Delta(\mathcal{S})} \big(\mathbb{E}_{s \sim \mu_h}[V_{h+1}^{k,\beta,\eta}] + \beta\chi^2(\mu_h \| \mu_h^o)\big) - \inf_{\mu_h \in \Delta(\mathcal{S})} \big(\mathbb{E}_{s \sim \mu_h}[V_{h+1}^{\pi^k,\beta,\eta}] + \beta\chi^2(\mu_h \| \mu_h^o)\big)\Big] \quad \text{(E.81)}$$

$$\leq 2 \cdot \Gamma_h^k + \phi^\top \mathbb{E}_{s \sim \mu_h^{w,k}}\big[V_{h+1}^{k,\beta,\eta} - V_{h+1}^{\pi^k,\beta,\eta}\big] \quad \text{(E.82)}$$

$$\leq 2 \cdot \Gamma_h^k + \mathbb{E}_{s \sim P_h^w}\big[V_{h+1}^{k,\beta,\eta} - V_{h+1}^{\pi^k,\beta,\eta}\big],$$

where (E.81) can be derived from Lemma E.10, and (E.82) uses the definition of the worst-case transition. Apply the above inequality recursively, we have

$$Q_1^{k,\beta,\eta} - Q_1^{\pi^k,\beta,\eta} \leq 2 \cdot \mathbb{E}_{P^{w,k},\pi^k}\left[\sum_{h=1}^H \Gamma_h^k\right].$$

This finishes the proof. □

*Proof of Lemma E.1 in $\chi^2$ Setting.* The result directly follows from combining Lemma E.10 and Lemma E.11, along with applying a union bound. □

# F. Auxiliary Lemmas

Here, we present some auxiliary lemmas which are useful in the proof.

**Lemma F.1** (Hoeffding's inequality). (Vershynin, 2018, Theorem 2.2.6) Let $X_1, \cdots, X_T$ be independent random variables. Assume that $X_t \in [0, M]$ for every $t$ with $M > 0$. Let $S_T = \frac{1}{T}\sum_{t=1}^T X_t$, then for any $\epsilon > 0$, we have

$$\mathbb{P}\big(|S_T - \mathbb{E}[S_T]| \geq \epsilon\big) \leq 2\exp\left(-\frac{2T\epsilon^2}{M^2}\right).$$

**Lemma F.2** (Self-bounding variance inequality). (Maurer & Pontil, 2009, Theorem 10) Let $X_1, \cdots, X_T$ be independent and identically distributed random variables with finite variance. Assume that $X_t \in [0, M]$ for every $t$ with $M > 0$. Let $S_T^2 = \frac{1}{T}\sum_{t=1}^T X_t^2 - (\frac{1}{T}\sum_{t=1}^T X_t)^2$, then for any $\epsilon > 0$, we have

$$\mathbb{P}\big(|S_T - \sqrt{\text{Var}(X_1)}| \geq \epsilon\big) \leq 2\exp\left(-\frac{T\epsilon^2}{2M^2}\right).$$

**Lemma F.3.** (Weissman et al., 2003, Theorem 2.1) Let $P$ be a probability distribution over $\mathcal{S} = \{s_1, \cdots, s_S\}$, $X_1, \cdots, X_T$ be independent and identically distributed random variables distributed according to $P$. Let $\widehat{P}(s) = \frac{1}{T}\sum_{t=1}^T \mathbb{1}\{X_t = s\}$, then for any $\epsilon > 0$, we have

$$\mathbb{P}\big(\|P - \widehat{P}\|_1 \geq \epsilon\big) \leq 2^S \exp\left(-\frac{T\epsilon^2}{2}\right).$$

**Lemma F.4.** (Panaganti & Kalathil, 2022, Lemma 7) We define $\mathcal{V} = \{V \in \mathbb{R}^S : \|V\|_\infty \leq V_{\max}\}$. Let $\mathcal{N}_\mathcal{V}(\epsilon)$ be a minimal $\epsilon$-cover of $\mathcal{V}$ with respect to the distance metric $d(V, V') = \|V - V'\|_\infty$ for some fixed $\epsilon \in (0, 1)$. Then we have

$$\log|\mathcal{N}_\mathcal{V}(\epsilon)| \leq |\mathcal{S}| \cdot \log\left(\frac{3V_{\max}}{\epsilon}\right).$$

**Lemma F.5.** (Van Handel, 2014, Lemma 5.13) Denote the $\epsilon$-covering number of the closed interval $[a, b]$ for some real number $b > a$ with respect to the distance metric $\text{dist}(\alpha_1, \alpha_2) = |\alpha_1 - \alpha_2|$ as $\mathcal{N}_h(\epsilon; [a, b])$, then we have $\mathcal{N}_h(\epsilon; [a, b]) \leq 3(b - a)/\epsilon$.

**Lemma F.6.** (Jin et al., 2020, Lemma D.5) For any $\epsilon > 0$, the $\epsilon$-covering number of the Euclidean ball in $\mathbb{R}^d$ with radius $R > 0$ is upper bounded by $(1 + 2R/\epsilon)^d$.

**Lemma F.7.** (Abbasi-Yadkori et al., 2011, Theorem 1) Let $\{\epsilon_t\}_{t=1}^{\infty}$ be a real-valued stochastic process with corresponding filtration $\{\mathcal{F}_t\}_{t=0}^{\infty}$. Let $\epsilon_t | \mathcal{F}_{t-1}$ be mean-zero and $\sigma$-sub-Gaussian. Let $\{\phi_t\}_{t=1}^{\infty}$ be a $\mathbb{R}^d$-valued stochastic process where $\phi_t$ is $\mathcal{F}_{t-1}$ measurable. Assume $\Lambda_0$ is a $d \times d$ positive define matrix, and let $\Lambda_t = \Lambda_0 + \sum_{s=1}^{t} \phi_s \phi_s^\top$. Then for any $\delta > 0$, with probability at least $1 - \delta$, we have for all $t \geq 0$,

$$\left\| \sum_{s=1}^{t} \phi_s \epsilon_s \right\|_{\Lambda_t^{-1}}^2 \leq 2\sigma^2 \log \left[ \frac{\det(\Lambda_t)^{\frac{1}{2}} \det(\Lambda_0)^{-\frac{1}{2}}}{\delta} \right].$$

**Lemma F.8.** (Liu & Xu, 2024a, Corollary 5.3) For all $(\pi, h) \in \Pi \times [H]$, assume that

$$\mathbb{E}_\pi [\phi(s_h, a_h) \phi(s_h, a_h)^\top] \succeq \alpha I,$$

where $\alpha > 0$. Then with probability at least $1 - \delta$, we have for all $(k, h) \in [K] \times [H]$,

$$\lambda_{\min}(\Lambda_h^k) \geq \max \left\{ \alpha(k-1) + \lambda - \sqrt{32k \log(dKH/\delta)}, \lambda \right\}.$$

