# OpenReview forum: "Doubly Regularized Markov Decision Processes for Robust Reinforcement Learning"
_ICML.cc/2026/Conference — ICML 2026 regular_

### Official Review · Reviewer_RpcJ · 2026-02-24

**Soundness:** 2
**Presentation:** 3
**Significance:** 2
**Originality:** 3
**Overall Recommendation:** 3
**Confidence:** 4

**Summary:**

This paper proposed the doubly regularised MDP framework that incorporates both transition and policy regularisation. Under this framework, the paper proposed RSPVI algorithm for solving discrete MDPs and its extension for solving linear MDPs. The paper shows finite sample regret bound of RSPVI under different f-divergences. The bound crucially depends on the visitation measure ratio. Some neumerical experiments are provided to show the importance of the framework.

**Compliance With Llm Reviewing Policy:**

Affirmed.

**Final Justification:**

The theory extends some of the current regularised MDP from discrete setting to linear (continuous) setting, however, the theoretical insights, techniques and formulations lacks novelty. Meanwhile, the experiments are not strong, as only very simple problems can be solved and it currently cannot handle harder control problems.

**Key Questions For Authors:**

Instead of dropping the exploration term in Algorithm 2, can you use any approximation method to boost the computational efficiency?

What are the bottlenecks to extend the experiments to larger problems, such as half-cheetah, walker, ant?

What is the reference policy used in the KL divergence? I'd imagine different types of reference policies provide quite different solutions?

Does similar results hold for Squared Hellinger distance? Or does the square root makes the proof tricky?

**Limitations:**

The paper lacks a discussion on the limitation of their method. Beside implementation issues, what are the limitation exists? In particular, their are other frameworks that investigates two regularisations for different terms, such as R^2 MDPS by Derman et.al, 2023. With multiple different frameworks, it's hard to see what advantages and distanvantages each has.

**Strengths And Weaknesses:**

===================================
Soundness:
All theoretical results are backed up by proper mathematical analysis and proofs. The main concern I had is for the experiment section. The paper says computing the exploration term in Algorithm 2 incur substantial computation overheads, hence they removed it from the implementation. This term arises from regularsiations they imposed am I correct? Then dropping it means the empirical algorithm is no longer justifying the theory.

Furthermore, you also dropped the term in DR-LSVI-UCB, but by definition the algorithm is no longer DR-LSVI-UCB without the term.

Also, the experiments are only demonstrated on simple control problems (i.e. pendulum and cartpole), it's unclear the general applicability of the method to broarder (harder) problems.

Another minor point is that the contribution section of the paper keeps claiming (or at least provides the impression) that they focus on general f-divergences, which give the reader a feeling that the theory is applied to all f-divergences, whereas they only provided theoretical results for different cases of f-divergences.

=====================================
Presentation:
Overall the presentation of the paper is clear. Some minor issues with the clarity of the mathematical notations. When defining the robust Bellman estimator in page 3 and 4, it's hard to see where does each s and a goes. Is it correct that E_P[\cdot](s,a) means E_{P(\cdot\mid s,a)}[\cdot]? And the minimisation is only applied to the first V term inside the expectation, the second s come from the expectation? In this case, the second s should be s' and the expectation should be written as E_{s'\sim P(\cdot\mid s,a)}.

Similarly, in the last equation on page 17, the equality just before E.3, the inf term operates over P_t only. I don't say how this equality holds unless P_t here also applies to P_h?

On page 20, why suddenly the notation of the bouns term is change from b to bonus?

If changes are made according to above, please ensure consistencies to other sections too (i.e. KL case, Tsallis case etc).

=====================================
Significance:
The technical proof largely follows the one used in He et.al, 2025, but applied to a different setting (i.e. doubly regularised MDPs). The paper also do not extent their theoretical results for arbitary f-divergences. The empirical section is not strong enough to show the importance of the framework, I am not convinced that in general one has to apply regularisations to both policy and distributions to achieve robustness.

====================================
Originality:
The doubly regularised MDP framework is studied and multiple theoretical results justify its importance.

---

> ### Author Rebuttal · Authors · 2026-03-31
>
> Thank you for your time and effort. We hope our response fully addresses your questions.
>
> ---
>
> **Q1:** Concerns about dropping the bonus term
>
> **A1: Origin of bonus:** We employ policy regularization to enable softmax-based updates, making the algorithm suitable for continuous action spaces, where greedy updates in prior work ([1, 2, 3]) cannot handle infinitely many actions. Additionally, we incorporate transition regularization to learn a more robust policy. A bonus is introduced specifically to promote exploration, rather than from regularization; therefore, its removal does not compromise the robustness of decision-making. For the same reason, omitting the bonus term for DR-LSVI-UCB does not affect its robustness.
>
> **Theoretical gap:** Bonus terms are often difficult to design and compute for deep RL methods with function approximation. As a result, alternative techniques are commonly used to achieve similar exploration abilities. In this work, we adopt a warm-up phase of 2000 training steps using a random policy. We also employ policy regularization to maintain stochasticity in the learned policy. These methods ensure that the agent retains sufficient exploration capability, which is reflected by the fact that the agent learns an optimal policy in the absence of perturbations. We also implemented RSPVI with bonus terms in CartPole and refer the reviewer to A2 of our response to Reviewer 8XQo due to space constraints.
>
> ---
>
> **Q2:** Concerns about the importance of the empirical section
>
> **A2: Our contributions:**
>
> (1). **Tabular Setting**. Prior work ([2, 3]) is limited to discrete state and action spaces. In contrast, we employ neural networks as general function approximators and incorporate policy regularization for softmax updates to extend the algorithm to continuous domains. Furthermore, in the KL-divergence setting, the dual formulation of the robust Bellman equation results in a nonlinear expectation over the value function, leading to empirical estimations being biased. To address this, we propose an exponentiation transformation of the Q-function approximator (Appendix C.1).
>
> (2). **Linear Setting**. Prior work ([1]) assumes that the linear feature mapping $\phi$ is known, an assumption that rarely holds in practice. We instead propose a novel approach to learn $\phi$ using a discrete VAE and demonstrate its success in CartPole.
>
> **Bottlenecks to scale**: The applicability of our approach to more challenging problems remains uncertain. Extending it to complex environments introduces several difficulties, including higher-dimensional state and action spaces, increased dynamical complexity and computational cost, as well as challenges in designing meaningful adversarial perturbations and assessing robustness.
>
> ---
>
> **Q3**: Comments on the writing
>
> **A3: Writing and notations**: This work considers only three f-divergences (TV, KL, $\chi^2$). By $\mathbb{E}\_{\widehat{P}\_h^k}\[V](s,a)$, we mean $\mathbb{E}_{s'\sim\widehat{P}_h\^k(s,a)}[V(s')]$. Your understanding about the minimization and expectation is correct. The infimum operator on page 17 also applies to $P_h$. We will revise these in the camera-ready version.
>
> **Reference policy**: The reference policy is trained using NS entropy policy regularization. Your understanding that different types of reference policies can lead to different solutions is correct. In practice, the reference policy serves as an anchor, ensuring that the learned policy does not deviate excessively from a chosen baseline.
>
> **Squared Hellinger distance**: Our TV, KL, and $\chi^2$ divergence proofs rely on the dual formulation of the robust Bellman operator, extending such a formulation to the Squared Hellinger distance remains an open problem.
>
> ---
>
> **Q4:** The limitations of our method and a comparison with [4]
>
> **A4: Limitations:** Our experiments primarily focus on classical control tasks. In addition, we study only the (s,a)-rectangularity and d-rectangularity settings; extending our algorithm to other robust MDP frameworks remains an open problem.
>
> **Comparison:** A detailed comparison between our work and that of [4] is provided in Appendix B. Importantly, the incorporation of policy regularization in our framework is intended to facilitate extension to continuous action spaces. Combined with the closed-form dual formulations under TV and KL divergence settings, this allows us to establish a practical and computationally efficient algorithm with the potential to scale to more complex tasks.
>
> ---
>
> **References:**
>
> [1] Distributionally Robust Off-Dynamics Reinforcement Learning: Provable Efficiency with Linear Function Approximation
>
> [2] Sample Complexity of Distributionally Robust Off-Dynamics Reinforcement Learning with Online Interaction
>
> [3] Distributionally Robust Reinforcement Learning with Interactive Data Collection: Fundamental Hardness and Near-Optimal Algorithms
>
> [4] Twice regularized mdps and the equivalence between robustness and regularization

---

> > ### Author Rebuttal · Reviewer_RpcJ · 2026-04-02
> >
> > Thanks for the detailed clarifications.
> >
> > What is the non-robust baseline in the experiment? Is it just SAC or some variant of it?
> >
> > It's interesting to see the linear MDP theory gets extended with regularisations. Transforming non-linear equation to linear ones using exponential trick is not new in general, though I do acknowledge using it under the double robust setting is new and allows one to work with continuous setting. Yet proof structure, as also mentioned by reviewer 8XQo, seems to follow the same structure as the discrete setting from previous works. For instance, if somehow the general theory of f-divergence can be properly studied, I would say the theory contribution is much stronger.
> >
> > Also, learning the linear feature $\phi$ has been extensively studied too. For instance, the general theory has been studied by [1] and [2]. Based on the theory, many practicle work show that $\phi$ can be obtained for free in Gaussian setting [3], learnt via NCE [4], and score matching [5]. Although I do acknowledge that they study the usual MDP setting without regularisation, their empirical experiments are quite strong and can handle many control tasks from gymnasium, DMC, and metaworld with visual inputs.
> >
> > Overall, I do acknowledge some of the new insights brought by the work, but the theory and empirical results are not strong enough.
> >
> > ---
> > [1] Agarwal, A., Kakade, S., Krishnamurthy, A. and Sun, W., 2020. Flambe: Structural complexity and representation learning of low rank mdps. Advances in neural information processing systems, 33, pp.20095-20107.
> >
> > [2] Uehara, M., Zhang, X. and Sun, W., 2021. Representation learning for online and offline rl in low-rank mdps. arXiv preprint arXiv:2110.04652.
> >
> > [3] Ren, T., Zhang, T., Szepesvári, C. and Dai, B., 2022, August. A free lunch from the noise: Provable and practical exploration for representation learning. In Uncertainty in Artificial Intelligence (pp. 1686-1696). PMLR.
> >
> > [4] Zhang, T., Ren, T., Yang, M., Gonzalez, J., Schuurmans, D. and Dai, B., 2022, June. Making linear mdps practical via contrastive representation learning. In International Conference on Machine Learning (pp. 26447-26466). PMLR.
> >
> > [5] Shribak, D., Gao, C.X., Li, Y., Xiao, C. and Dai, B., 2024. Diffusion spectral representation for reinforcement learning. Advances in Neural Information Processing Systems, 37, pp.110028-110056.

---

> > > ### Author Response · Authors · 2026-04-07
> > >
> > > Thank you for the detailed feedback and for the thoughtful discussion of both the theoretical and empirical aspects of our work. We appreciate the opportunity to further clarify the scope and positioning of the paper. Prior to this work, RMDPs under $(s,a)$-rectangularity and $d$-rectangularity have primarily been studied as conceptual frameworks, with limited guidance on how to translate these formulations into scalable deep RL algorithms. In particular, there is a gap between the rich theoretical characterizations of robustness and the lack of practical methods that can leverage these insights in modern function approximation settings. Our work aims to bridge this gap by establishing a concrete pathway from robust MDP theory to implementable deep RL algorithms, through the introduction of a doubly regularized MDP framework that admits both theoretical analysis and scalable optimization.
> > >
> > > More broadly, the paper is driven by a unified objective: to understand how policy regularization interacts with robustness in RRMDPs, and to turn this understanding into algorithmic design principles. The contributions therefore operate across multiple levels but are tightly coupled. On the theoretical side, we systematically study policy regularization under both $(s,a)$-rectangularity and $d$-rectangularity—two canonical formulations of robustness—and characterize their behavior under three common $f$-divergences (TV, KL, and $\chi^2$). We place particular emphasis on TV and KL divergences, where the resulting robust Bellman operators admit tractable forms that directly inform algorithm design. While extending these results to broader classes of $f$-divergences is an interesting direction and has been partially explored in prior work, our focus here is to develop settings where the theory not only provides insight, but also leads to computationally viable and scalable methods.
> > >
> > > Building on this foundation, we develop a set of algorithmic components tailored to the doubly regularized framework proposed in this paper. These components differ from existing approaches and are designed to work coherently:
> > >
> > > - For linear $d$-rectangular RRMDPs, the feature $\phi$ is constrained to lie on a $d$-dimensional probability simplex. Despite existing papers learning the linear feature $\phi$, their approach does not apply to our setting (we would discuss these related works). To accommodate this structure, we propose to learn $\phi$ via a discrete VAE and incorporate additional entropy regularization to avoid degeneration. We further separate representation learning from policy learning through a warm-up phase, which improves stability. Given the learned feature, we estimate the robust $Q$-function via $d$ ridge regressions.
> > >
> > > - For $(s,a)$-rectangular KL-RRMDPs, we introduce a new neural parameterization for value estimation (see details in eq. C.3) based on the robust Bellman operator. Instead of directly estimating the robust value function, we target an exponential transformation. This new estimation framework is also accompanied by several stability-oriented design choices, including (1) a clipping to avoid numerical issues when calculating $\log f_{\theta}(s,a)$; (2) maintaining two Q-networks as in SAC, but instead of taking min of the two networks, we take max to avoid overestimation issues. This is based on the observation that the Q-network is estimating $\exp(-Q(s,a)/\beta)$; (3) maintaining a separate value-network $g_{\eta}$ to avoid large variance and bias in estimation. These design choices arise naturally from the structure of the robust objective.
> > >
> > > All components are implemented from scratch and integrated into a single pipeline for deep robust RL. This integration requires careful coordination between representation learning, value estimation, and policy optimization, and is an essential part of making the approach work in practice. Empirically, we evaluate the method on classic control tasks such as CartPole and Inverted Pendulum variants. While these environments are relatively standard, they capture key challenges in robust RL, including continuous state-action spaces, adversarial perturbations in dynamics, and numerical instability in value estimation. The results show that the proposed pipeline can effectively address these challenges. An additional observation is that the learned robust policies achieve comparable or sometimes improved performance even in the nominal environment, suggesting that robustness does not necessarily degrade nominal performance. We believe this is an interesting phenomenon that may merit further investigation.
> > >
> > > Overall, the goal of this work is to connect theory and practice by developing a principled and implementable approach to robust deep RL. We hope this clarification better conveys the scope of our contributions, and would greatly appreciate it if the reviewer could consider raising the score in light of our clarifications.

---

### Official Review · Reviewer_xQ2f · 2026-03-05

**Soundness:** 3
**Presentation:** 3
**Significance:** 3
**Originality:** 3
**Overall Recommendation:** 5
**Confidence:** 4

**Summary:**

The paper proposes a doubly regularized MDP framework that simultaneously penalizes deviations of the transition model from a nominal one via general f-divergences and regularizes the policy. The authors design an optimism-based online algorithm with explicit robust Bellman operators under TV/KL/χ² regularization and prove the first finite-sample regret bounds for this setting in both tabular and linear function approximation cases under a bounded visitation measure ratio assumption. Empirically, they implement practical variants and show improved robustness on control tasks relative to non-robust baselines.

**Compliance With Llm Reviewing Policy:**

Affirmed.

**Final Justification:**

I recommend accept. The paper is original and technically solid: it introduces a doubly regularized MDP framework that unifies transition and reward robustness, provides explicit operators and regret guarantees in both tabular and linear settings, and is supported by empirical robustness improvements. The rebuttal addressed my main concerns well—especially by clarifying the role of the visitation-ratio assumption, explaining why the regret bound is essentially independent of $\eta$ in order while $\eta$ still matters in practice, and being transparent about the experimental omission of bonuses in the linear setting—so overall it strengthened my confidence in the paper.

**Key Questions For Authors:**

In the linear setting, the bonus term is computationally expensive and was omitted in experiments. Are there approximate or amortized ways to compute/use bonuses that retain some theoretical guarantees or empirical benefits?

How does the regret depend on the policy regularization strength η? Is the bound independent of η (beyond constants), and if so, why? Are there stability or bias-variance trade-offs that suggest principled tuning of η?

Is the bounded visitation measure ratio assumption testable or at least diagnosable from data? Any insights into when it is likely to hold in practice?

**Limitations:**

yes

**Strengths And Weaknesses:**

Strengths :

This paper combines transition-regularization with policy regularization in an online learning setup; provides a unifying “double robustness” view connecting transition robustness and reward robustness through policy regularization.

The paper derives explicit robust Bellman operators for TV, KL, and χ² cases and soft-policy updates for several regularizers, enabling stochastic optimal policies in contrast to classical robust MDPs’ deterministic policies.

Empirical results indicate robustness improvements under injected perturbations and demonstrate feasibility at moderate scale.


Weaknesses:

The central theoretical results hinge on the bounded visitation measure ratio assumption , which can be strong and environment-dependent.

The dependence on policy regularization strength η is largely hidden in the statements; it would help to make explicit if/where η influences constants, stability, or bias-variance trade-offs in regret.

The linear experiment learns features via a VAE and omits the exploration bonus , weakening the connection to the proven algorithm and making the comparison to theory indirect.

---

> ### Author Rebuttal · Authors · 2026-03-31
>
> Thank you for the positive feedback on our work. We hope our response fully addresses your questions.
>
> ---
>
> **Q1:** Discussion of the bounded visitation measure ratio assumption
>
> **A1: Environment-dependent:** This assumption is indeed environment-dependent. When $C_{vr}$ is large, it indicates a significant discrepancy between the nominal and perturbed environments. Consequently, the information obtained through exploration in the nominal environment becomes less useful for decision-making in the perturbed environment, making the learning more challenging.
>
> **Not strong:** This assumption is indeed **standard and necessary** to guarantee learnability. In fact, [1] constructs a hard instance (Lem 5.14) demonstrating that online robust learning can become exponentially difficult without such assumption. Furthermore, [1] (Thm 5.16 and 5.18) shows that the regret upper bounds match the corresponding lower bounds in terms of their dependence on $C_{vr}$. This suggests that $C_{vr}$ serves as a tight measure of exploration difficulty in RMDPs and can therefore be considered an **intrinsic quantity** of the environment.
>
> ---
>
> **Q2:** The omission of bonus terms
>
> **A2:** Bonus terms are often difficult to design and compute for deep RL methods with function approximation. As a result, alternative techniques are commonly used to achieve similar exploration abilities. For example, [2] introduces an entropy term for policy regularization, encouraging the learned policy to be more stochastic by distributing probability mass across multiple high-value actions. Other techniques include $\epsilon$-greedy strategies and warm-up phases.
>
> In this work, we adopt a warm-up phase of 2000 training steps and employ policy regularization to maintain stochasticity in the learned policy. These methods ensure that the agent retains sufficient exploration capability, which is reflected by the fact that the agent learns an optimal policy in the absence of perturbations.
>
> We also implemented RSPVI with bonus terms in CartPole and refer the reviewer to A2 of our response to Reviewer 8XQo due to space constraints.
>
> ---
>
> **Q3:** Dependence of regret on policy regularization strength $\eta$ and the trade-off in tuning $\eta$
>
> **A3: Dependence**: The regret is defined as the cumulative suboptimality between the robust regularized value function under the optimal policy and that under the policy selected by the agent, and the robust regularized value function itself depends on $\eta$. In our regret bounds (Thm 4.4, 5.6, and 5.7), the results are independent of $\eta$. We discuss this in the paragraph beginning at Line 247. The key is that, under our policy update (Eq 3.3 and Line 1106), the regret is upper bounded by bonus terms that are independent of $\eta$, and therefore the final regret bound does not depend on $\eta$.
>
> **Trade-offs:** In practice, $\eta$ plays an important role in controlling exploration. A larger $\eta$ encourages exploration by making the learned policy closer to uniform, but may result in overly random behavior. Conversely, a smaller $\eta$ concentrates probability mass on actions with higher estimated Q-values, which can lead to insufficient exploration. Therefore, choosing a moderate value of $\eta$​ is crucial for achieving good empirical performance. This trade-off is also validated by our experiments in Appendix C.2.
>
> ---
>
> **Q4:** Discussion of the bounded visitation measure ratio assumption
>
> **A4: Test assumption:** $C_{vr}$ is defined as the supremum of the ratio between the visitation measure in the worst-case environment and that in the nominal environment, taken over all states, actions, and policies. However, because the policy space is typically infinite, computing this quantity exactly is generally intractable.
>
> But we can instead estimate it empirically. For instance, we may roll out trajectories in both the nominal environment and a simulator for the perturbed environment using random policies. By comparing the resulting state coverage, we can obtain an approximate sense of the order of this supremum ratio.
>
> **Hold in practice:** This quantity satisfies $C_{vr}=\sup\frac{q(s)}{d(s)}\leq\sup\frac{1}{d(s)}$. Thus, if the nominal environment is sufficiently exploratory, meaning that all states have reasonably large visitation probability under typical policies, then online robust learning becomes feasible. Such conditions are often met in classical control environments where the transition dynamics are relatively simple and states are broadly reachable.
>
> However, [1] construct a hard instance (Lem 5.14) showing that when the distribution shift is large, this ratio can grow significantly. We might deliberately collect data in regions of the state space that exhibit large distributional shifts in such cases.
>
> ---
>
> **References:**
>
> [1] Sample Complexity of Distributionally Robust Off-Dynamics Reinforcement Learning with Online Interaction
>
> [2] Soft Actor-Critic Algorithms and Applications

---

> > ### Author Rebuttal · Reviewer_xQ2f · 2026-04-02
> >
> > My initial concern is solved. I will keep my positive score.

---

> > > ### Author Response · Authors · 2026-04-07
> > >
> > > Dear Reviewer xQ2f,
> > >
> > > We are pleased that your previous concerns have been addressed. Thank you again for your support of our work.

---

### Official Review · Reviewer_5Lse · 2026-03-11

**Soundness:** 3
**Presentation:** 3
**Significance:** 3
**Originality:** 3
**Overall Recommendation:** 5
**Confidence:** 4

**Summary:**

This paper studies doubly regularized Markov Decision Processes, where robustness is enforced with respect to both reward perturbations and transition dynamics uncertainty. The resulting formulation introduces regularization at both the policy and dynamics levels, yielding a principled robust policy optimization framework.

The authors develop the theory in both tabular and linear function approximation settings. They derive finite-sample regret bounds and show that incorporating reward uncertainty does not worsen the sample complexity compared to dynamics-only robustness. Beyond theoretical guarantees, the proposed method admits soft policies and can scale to continuous state-action spaces. The use of dual formulations of f-divergences enables closed-form policy updates, which is both theoretically elegant and practically relevant.

Overall, the work combines careful theoretical development with algorithmic implications and empirical validation.

**Compliance With Llm Reviewing Policy:**

Affirmed.

**Key Questions For Authors:**

-- In Section 6 the term perturbation could be clearer, i.e. line 433-436: does that mean that every time the policy takes an action, with a fixed probability p this action is replaced by a randomly sampled action? Are the randomly sampled actions clipped or unbounded? And p in this case corresponds to „perturbation” on the x-axis? Here additional clarification would be appreciated. An additional baseline of a policy that always chooses a random action could also be helpful to put the results in the limit perturbation -> 1 into context.

-- Personally, I don’t understand Figures 1 (a) and (b) very well. The double inverted pendulum should be much more sensitive than the classical inverted pendulum. Nevertheless, looking at Figure 1 (b) reveals that up to perturbation $\approx$ 0.5 there is almost no loss of performance, while in the classical inverted pendulum the performance monotonically decreases with increasing level of perturbation. Here the interplay between the sensitivity of the environment and the robustness of the proposed method should be discussed in more detail.

**Limitations:**

Yes

**Strengths And Weaknesses:**

**Strengths:**

-- The paper is well written and carefully structured. The theoretical exposition is comprehensive and detailed, and the relationship to prior and concurrent work is clearly articulated.

-- The motivation for robustness with respect to both reward and transition perturbations is compelling. In particular, Proposition 2.2 provides a strong conceptual justification for the penalty-based formulation and clearly guides the subsequent development.

-- The theoretical results are substantial. The derivations are rigorous and extend beyond the tabular case under reasonable assumptions. The finite-sample guarantees are well developed and carefully proven.

-- The introduction of dual formulations of f-divergences to obtain closed-form policy updates is both theoretically sound and practically meaningful. This design choice strengthens the scalability of the method.

**Weaknesses:**

-- Double regularized MDPs (line 106 onwards) the authors introduce $r^0$ which is not occurring in the subsequent definitions. Additional clarification would be helpful.

-- Section 3.2 deals with the estimation of the Q-function. While the derivations in the appendix are very detailed and seem correct, the motivation for the choice of the bonus terms could be more detailed in the main part of the paper, i.e. 1-2 sentences on the fact that the bonus terms give an upper bound in probability wrt the optimal Q-function. This would also make the occurrence of $\delta$ clearer.


**Minor comments:**

-- It could be beneficial to give explicit references to the proofs in the appendix, so readers can jump easily back and forth between the main part and the appendix.

-- The Notation in Def. 4.2 and 5.4 is slightly confusing, it could be better to use $\mathrm{d}^{\pi, \diamond}_h(\cdot)$ with $\diamond \in \{o,w\}$ to unify the notation instead of using q and d.

-- Assumption 5.5 „the definition of” potential typo.

-- Thm. 5.6 „holds” -> „hold” typo

---

> ### Author Rebuttal · Authors · 2026-03-31
>
> Thank you for the positive feedback on our work. We hope our response fully addresses your questions.
>
> ---
>
> **Q1.** The authors introduce r^0 which is not occurring in the subsequent definitions
>
> **A1:** Here, $r^0$ should instead be $r$​, representing the reward function of the nominal environment. We will correct this notation in the camera-ready version.
>
> ---
>
> **Q2:** The motivation for the choice of the bonus terms could be more detailed in the main part of the paper
>
> **A2:** We will revise the writing in the camera-ready version. Bonus terms of this form are widely used in the online RL literature ([1, 2, 3, 4]). In such settings, a central challenge is balancing exploration and exploitation. The bonus term addresses this by encouraging the agent to explore less frequently visited state-action pairs. From a theoretical perspective, the bonus is designed so that the estimated Q-function is optimistic, that is, with high probability, it upper-bounds the optimal Q-function. In our work, Lemmas E.4, E.7, and E.10 formally establish that, with the proposed bonus, the resulting Q-function estimate indeed satisfies this optimism property.
>
> ---
>
> **Q3:** Comments on typos, notations, and references to the proofs
>
> **A3:** Thank you for your detailed feedback. We have corrected these typos and include references to the proofs.
>
> ---
>
> **Q4:** In Section 6 the term perturbation could be clearer
>
> **A4:** Thank you for your suggestion. To clarify, the perturbation level $t$ (represented on the x-axis) determines the probability of action perturbation. After the agent selects an action $a$, this action is replaced by a random action sampled uniformly from the action space with probability $t$. For example, if the action space is $[-3, 3]$​, the randomly sampled action will lie within this bounded range.
>
> ---
>
> **Q5:** An additional baseline of a policy that always chooses a random action could also be helpful
>
> **A5:** We evaluated the performance of using random policy across various tasks, as summarized below. These results can be interpreted as the limiting case where the perturbation rate approaches 1.
>
> | Task Name                | Mean  | Std   |
> | ------------------------ | ----- | ----- |
> | Cart Pole                | 22.98 | 11.18 |
> | Inverted Pendulum        | 4.48  | 2.65  |
> | Inverted Double Pendulum | 47.35 | 15.48 |
>
> ---
>
> **Q6:** Different performance drops under action perturbations in Pendulum and Double Pendulum environments
>
> **A6:** Thank you for your insightful question. We would like to highlight that the action and reward designs differ significantly between these two environments. (1) The action space of the Inverted Pendulum is $[-3, 3]$, whereas that of the Inverted Double Pendulum is $[-1, 1]$. As a result, when perturbing the learned policy with actions sampled uniformly from the respective action spaces, the magnitude of perturbation, and thus the difficulty of the task, can differ substantially. (2) The reward structures also differ. In the Inverted Pendulum task, the agent receives a constant reward of $1$ as long as the angle between the pole and the vertical axis remains within $0.2$ radians. In contrast, for the Inverted Double Pendulum task, termination occurs when the y-coordinate of the tip of the second pole is $\leq 1$​, and the reward additionally depends on the tip position and the agent’s velocity. Therefore, the overall sensitivity is strongly dependent on the environment design and may vary across tasks.
>
> ---
>
> **References:**
>
> [1] Finite-time analysis of the multiarmed bandit problem
>
> [2] Provably efficient reinforcement learning with linear function approximation
>
> [3] Distributionally Robust Off-Dynamics Reinforcement Learning: Provable Efficiency with Linear Function Approximation
>
> [4] Sample Complexity of Distributionally Robust Off-Dynamics Reinforcement Learning with Online Interaction

---

> > ### Author Rebuttal · Reviewer_5Lse · 2026-04-05
> >
> > I thank the authors for their clarifications and additional empirical evaluation.

---

> > > ### Author Response · Authors · 2026-04-07
> > >
> > > Dear Reviewer 5Lse,
> > >
> > > We are pleased that your previous concerns have been addressed. Thank you again for your support of our work.

---

### Official Review · Reviewer_8XQo · 2026-03-11

**Soundness:** 2
**Presentation:** 3
**Significance:** 3
**Originality:** 2
**Overall Recommendation:** 3
**Confidence:** 3

**Summary:**

This paper proposes a doubly regularized Markov decision process framework, whose core lies in unifying policy regularization and dynamics regularization to address uncertainties in both the reward and transition models.

**Compliance With Llm Reviewing Policy:**

Affirmed.

**Key Questions For Authors:**

1.	The abstract claims to provide finite-sample regret guarantees in the "rich-observation" setting, but the main theoretical sections only specify the tabular and linear settings. What exactly is meant by "rich-observation" here?
2.	The paper proposes a doubly regularized framework. However, related literature shows that its two core components, namely using policy regularization for reward robustness and using dynamics regularization for transition robustness, are already established ideas. Recent work has also explored combining them. Therefore, a key question is what fundamental and inseparable new insight this framework provides at the conceptual level, beyond technical integration, when compared to simply adding the two regularization terms together.
3.	In the CartPole experiment, the authors omitted the critical exploration bonus term from the algorithm, which is a prerequisite for achieving the optimistic exploration required by the theoretical regret bound. After this omission, does the algorithm that was actually run still satisfy all the conditions of the theoretical analysis? Is the experiment validating a variant of the algorithm that lacks theoretical guarantees?

**Limitations:**

yes

**Strengths And Weaknesses:**

1.	The experimental section only evaluates robustness by introducing action perturbations during testing. However, a core claimed contribution of the method is robustness against dynamics perturbations. The paper lacks experimental validation in environments where the dynamics model itself is perturbed, which makes the empirical support for robustness against dynamics perturbations insufficient.
2.	In the CartPole experiment described in Section 6, the authors note that the exploration bonus term from Algorithm 2 was entirely omitted due to computational cost. This bonus term is a key component for achieving optimistic exploration, which underpins the theoretical regret bound. Its omission means the algorithm being evaluated is not the one analyzed theoretically, which weakens the experimental support for the theoretical claims.
3.	The framework presented in this paper shows substantial overlap with the work of Gu et al. (2025), offering insufficient differentiation. Although the paper notes three technical distinctions, namely extending the results to the tabular setting, adopting the more general bounded visitation measure ratio assumption in place of the fail-state assumption, and proposing a practical method for function approximation, these are primarily improvements in the underlying assumptions rather than innovations to the core framework itself.

---

> ### Author Rebuttal · Authors · 2026-03-31
>
> Thank you for your time and effort. We hope our response fully addresses your questions.
>
> ---
>
> **Q1:** The experimental section only evaluates robustness by introducing action perturbations during testing.
>
> **A1:** CartPole is a simulation of a physical system, which makes it difficult to explicitly perturb its transition dynamics. Instead, we perturb the action. This is equivalent to perturbing the transition dynamics, since random actions correspond to transitioning to different states with associated probabilities, i.e., $\widetilde{P}(s'\vert s,a)=(1-q)\*P^o(s'\vert s,a)+q*P^o(s'\vert s,a_\text{rand})$, where $q$ is the perturbation level and $\widetilde{P}$ is the perturbed transition.
>
> To evaluate our algorithm's performance under transition perturbations, we conduct experiments on the simulated MDP proposed in [1] (Appendix A.1), with $\Vert\xi\Vert=0.2$. The results demonstrate the robustness of our algorithm.
>
> |Perturbation|0|0.1|0.2|0.3|0.4|0.5|0.6|0.7|0.8|0.9|
> |-|-|-|-|-|-|-|-|-|-|-|
> |Non-robust|1.60|1.55|1.41|1.25|1.20|1.08|0.97|0.93|0.82|0.75|
> |RSPVI-TV|1.34|1.30|1.28|1.24|1.22|1.18|1.22|1.20|1.18|1.15|
>
> ---
>
> **Q2**: The omission of bonus terms weakens the experimental support for the theoretical claims
>
> **A2:** Bonus terms are often difficult to design and compute for deep RL methods with function approximation. As a result, alternative techniques are commonly used to achieve similar exploration abilities. For example, [2] introduces an entropy term for policy regularization, encouraging the learned policy to be more stochastic. Other techniques include $\epsilon$-greedy strategies and warm-up phases.
>
> In this work, we adopt a warm-up phase of 2000 training steps and employ policy regularization to maintain stochasticity in the learned policy. These methods ensure that the agent retains sufficient exploration capability, which is reflected by the fact that the agent learns an optimal policy in the absence of perturbations.
>
> We also implemented RSPVI with bonus terms in CartPole, which achieves similar performance but is much slower.
>
> |Perturbation|0|0.1|0.2|0.3|0.4|0.5|0.6|0.7|0.8|0.9|
> |-|-|-|-|-|-|-|-|-|-|-|
> |TV (with bonus)|200|197|195|180|164|117|72|55|34|26|
> |KL (with bonus)|200|200|193|186|160|115|72|44|36|27|
>
> |Setting|Time (min)|
> |-|-|
> |TV (no bonus)|12|
> |KL (no bonus)|10|
> |TV (with bonus)|160|
> |KL (with bonus)|154|
>
> ---
>
> **Q3:** Insights beyond combined regularization and comparison with prior work
>
> **A3: Novelties:**
>
> (1). **Tabular Setting**. Prior work ([3, 4]) is limited to discrete state and action spaces. In contrast, we employ neural networks as general function approximators and incorporate policy regularization for softmax updates to extend the algorithm to continuous domains. Furthermore, in the KL-divergence setting, the dual formulation of the robust Bellman equation results in a nonlinear expectation over the value function, leading to empirical estimations being biased. To address this, we propose an exponentiation transformation of the Q-function approximator (Appendix C.1).
>
> (2). **Linear Setting**. Prior work ([1]) assumes that the linear feature mapping $\phi$ is known, an assumption that rarely holds in practice. We instead propose a novel approach to learn $\phi$ using a discrete VAE and demonstrate its success in Cart Pole.
>
> **Comparison with [5]**: our work is concurrent with theirs but provides significantly stronger experimental evaluation. We assess our algorithm on the Cart Pole, Inverted Pendulum, and Inverted Double Pendulum environments, while [5] evaluate their approach only in a simulated RMDP, where the dynamics are much simpler, the state and action spaces are discrete, and the linear feature mapping is known to the agent.
>
> Our work also differs substantially from [5] in theory. In [5], Theorem 5.2 requires both the fail-state and an additional exploration assumption (Eq 5.5 in our work). In contrast, our Theorem 5.7 requires only the exploration assumption, enabled by an improved proof technique: we bound regret via bonus terms under a worst-case expectation (Lemma F.1) and then apply Jensen's inequality to leverage the exploration assumption (Line 1676).
>
> ---
>
> **Q4:** Meaning of "rich-observation"
>
> **A4:** This term refers to settings where the state space is continuous and high-dimensional, thereby requiring function approximation ([6]).
>
> ---
>
> **References:**
>
> [1] Distributionally Robust Off-Dynamics Reinforcement Learning: Provable Efficiency with Linear Function Approximation
>
> [2] Soft Actor-Critic Algorithms and Applications
>
> [3] Distributionally Robust Reinforcement Learning with Interactive Data Collection: Fundamental Hardness and Near-Optimal Algorithms
>
> [4] Sample Complexity of Distributionally Robust Off-Dynamics Reinforcement Learning with Online Interaction
>
> [5] Policy Regularized Distributionally Robust Markov Decision Processes with Linear Function Approximation
>
> [6] Rich-observation reinforcement learning with continuous latent dynamics

---

> > ### Author Rebuttal · Reviewer_8XQo · 2026-04-02
> >
> > Thank you for your reply, which has addressed all my concerns. However, I am also concerned about whether the prerequisites of each auxiliary lemma match the proof background.

---

> > > ### Author Response · Authors · 2026-04-07
> > >
> > > Thank you again for the engaging discussion. We are pleased that your previous concerns have been addressed. Below, we further clarify how the prerequisites of each auxiliary lemma align with the overall proof framework. In addition, we have highlighted the core contributions of our work in our further response to Reviewer RpcJ. Due to space limitations, we do not repeat those points here. We would greatly appreciate it if the reviewer would consider raising the score in light of these clarifications.
> > >
> > > ---
> > >
> > > **Lemma G.1**: We use Lemma G.1 to bound the estimation error $|\widehat{r}_h^k(s,a)-r_h(s,a)|$, where $\widehat{r}_h^k(s,a)$ is the empirical estimate of $r_h(s,a)$ defined in Equation (3.1).
> > >
> > > Fix $(h,s,a)$, and let $\{i_1,\dots,i_T\}$ denote the set of episodes such that $s_h^{i_t}=s$ and $a_h^{i_t}=a$, where $T=n_h^{k-1}(s,a)$. Define $X_t=r_h^{i_t}(s,a)$ for $t=1,\dots,T$.
> > >
> > > Then $\{X_t\}_{t=1}^T$ are independent and identically distributed random variables bounded in $[0,1]$, due to the Markov property of the MDP and independence across episodes. Therefore, the conditions of Lemma G.1 are satisfied.
> > >
> > > ---
> > >
> > > **Lemma G.2**: This lemma is not used in the current version of the paper. We will remove it in the camera-ready version.
> > >
> > > ---
> > >
> > > **Lemma G.3**: In the proof of Lemma E.4, we need to bound $\Vert P_h^o(\cdot|s,a)-\widehat{P}_h^k(\cdot|s,a)\Vert_1$, where $P_h^o$ is the nominal transition kernel and $\widehat{P}_h^k$ is its empirical estimate defined in Equation (3.1).
> > >
> > > Fix $(h,s,a)$, and let $\{i_1,\dots,i_T\}$ denote the set of episodes such that $s_h^{i_t}=s$ and $a_h^{i_t}=a$, where $T=n_h^{k-1}(s,a)$. Define $X_t=s_{h+1}^{i_t}$ for $t=1,\dots,T$.
> > >
> > > Then $\{X_t\}_{t=1}^T$ are independent and identically distributed samples drawn from $P_h^o(\cdot|s,a)$, due to the Markov property and independence across episodes. Therefore, the conditions of Lemma G.3 are satisfied.
> > >
> > > ---
> > >
> > > **Lemma G.4**: In the proof of Lemma E.10, we define $\mathcal{V}=\\{V\in\mathbb{R}^S:\Vert V\Vert_\infty\leq H\\}$, which satisfies the conditions of this lemma. Applying the lemma yields the covering number bound $|\mathcal{N}_{\mathcal{V}}(\epsilon)|\leq(3H/\epsilon)^{|S|}$.
> > >
> > > ---
> > >
> > > **Lemma G.5**: In Equations (F.34), (F.74), and (F.79), we apply this lemma to the interval $[0,H]$, which is compact. The $\epsilon$-covering number is therefore bounded as $|\mathcal{N}_h(\epsilon;[0,H])|\leq3H/\epsilon$.
> > >
> > > ---
> > >
> > > **Lemma G.6**: In the proof of Lemma F.2, the set $\\{A\in\mathbb{R}^{d\times d}:\Vert A\Vert_F\leq d^{1/2}B^2\lambda^{-1}\\}$ is a Euclidean ball in $\mathbb{R}^{d^2}$. The $\epsilon$-covering number is therefore bounded as $|\mathcal{C}_A|\leq(1+2d^{1/2}B^2/(\lambda\epsilon)^{d^2}$. This is a standard argument, widely used in the literature (e.g., Lemma D.3 in [1] and Lemma D.6 in [2]).
> > >
> > > ---
> > >
> > > **Lemma G.7**: This is a standard concentration result for self-normalized processes. In our work, it is used to bound $\Vert\sum_{\tau=1}^{k-1}\mathbf{\phi}(s_h^\tau,a_h^\tau)\cdot\mathrm{err}\_h^\tau([V\_\epsilon]\_{\alpha\_\epsilon})\Vert\_{(\Lambda_h^k)^{-1}}^2$.
> > >
> > > Specifically, define $\epsilon\_\tau=[V\_\epsilon]\_{\alpha\_\epsilon}-\mathbb{E}[[V\_\epsilon]\_{\alpha\_\epsilon}\mid\mathcal{F}\_{\tau-1}]$. Then $\{\epsilon_\tau\}$ forms a martingale difference sequence with $\mathbb{E}[\epsilon\_\tau\mid\mathcal{F}\_{\tau-1}]=0$. Moreover, $\epsilon_\tau$ is sub-Gaussian since it is bounded by $H$. Therefore, the conditions of Lemma G.7 are satisfied.
> > >
> > > ---
> > >
> > > **Lemma G.8**: The conditions of this lemma are satisfied under Equation (5.5) in Theorem 5.7.
> > >
> > > ---
> > >
> > > **References:**
> > >
> > > \[1]: Distributionally Robust Off-Dynamics Reinforcement Learning: Provable Efficiency with Linear Function Approximation
> > >
> > > \[2]: Provably Efficient Reinforcement Learning with Linear Function Approximation

---

### Decision · Program_Chairs · 2026-04-30

**Decision:**

Accept (regular)

**Comment:**

The two reviewers voted for rejection are concerned about the experiment setting. However, given the strong theoretical contributions and their potential algorithmic implications, I believe the submission meets the bar of ICML and should be accepted.